# SuperActivators: Transformers Concentrate Concept Signals in Just a Handful of Tokens

## Abstract

Concept vectors aim to enhance model interpretability by linking internal representations with human-understandable semantics, but their utility is often limited by noisy and inconsistent activations. In this work, we uncover a clear pattern within this noise, which we term the **SuperActivator Mechanism**: while in-concept and out-of-concept activations overlap considerably, the token activations in the extreme high tail of the in-concept distribution provide a clear, reliable signal of concept presence. We demonstrate the generality of this mechanism by showing that SuperActivator tokens consistently outperform standard vector-based and prompting concept detection approaches—achieving up to a 14% higher F1 score—across diverse image and text modalities, model architectures, model layers, and concept extraction techniques. Finally, we leverage these SuperActivator tokens to improve feature attributions for concepts. [1]

## 1 Introduction

Modern transformer-based models, while increasingly powerful and ubiquitous (Minaee et al., 2025), remain opaque and can behave in ways that are unpredictable or harmful (Greenblatt et al., 2024; Roose). This opacity hinders our ability to identify and debug undesirable representations—such as spurious correlations (Zhou et al., 2024b), biases Yang et al. (2024), or fragile reasoning Berglund et al. (2024)—or to intervene when models produce undesirable outputs.

Concept vectors (Kim et al., 2018; Zhou et al., 2018), or semantically meaningful directions in a model's latent space, provide a lightweight tool for examining and influencing internal representations. They have been used to uncover hidden model failures (Abid et al., 2022; Yeh et al., 2020), and to steer model behavior away from hallucinations (Rimsky et al., 2023; Suresh et al., 2025), unsafe responses (Liu et al., 2023; Xu et al., 2024), and toxic language (Turner et al., 2024; Nejadgholi et al., 2022). Unsupervised concept extraction is especially powerful, since labeled data is costly and such methods have the potential to uncover and explain new model behaviors, contributing to scientific discoveries (Lindsey et al., 2025).

To analyze the presence of concepts within a sample, we typically rely on their activation scores—a measure of alignment between an input token's embedding and a concept vector. However, these scores are often noisy and unreliable, and as a result misrepresent true concept presence. For instance, prior works have found that concepts frequently activate on unintended semantics Olah et al. (2020); Bricken et al. (2023), generate overlapping signals for correlated concepts Goh et al. (2021); Olah et al. (2020), and exhibit unstable activation patterns across different model layers Nicolson et al. (2025). The example in Figure 1 provides an illustration of such activation ambiguity on an image of a dog in a car mirror. The activation heatmaps for both the *Animal* and *Person* concepts appear to highlight the same region, even though only the former is present. Moreover, it is evident that many tokens on the car itself fail to activate for the *Car* concept. Such noisy signals makes it difficult to reliably detect or localize concepts.

To better understand the source of this noise, we examined the global activation distributions of in-concept and out-of-concept tokens and found that while they overlap considerably, there is clear separation in the extreme tail of the in-concept distribution. Notably, these tail-end activations are well-distributed across true-concept samples, allowing them to reliably distinguish concept presence

---

[1] https://anonymous.4open.science/r/superactivator-E02D/

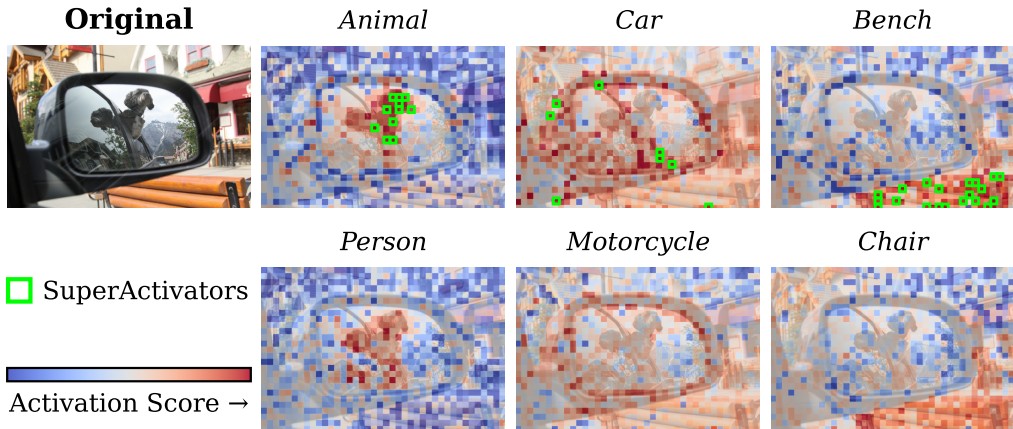

Figure 1: The SuperActivator Mechanism concentrates concept information into a sparse set of high-activation signals; by focusing on these signals, one can distinguish the true concepts in an image even when token activations are misleading, spuriously highlighting absent concepts and providing incomplete recall of the true ones. This example shows *LLaMA-3.2-11B-Vision-Instruct* linear separator concept activations on a COCO image; examples for a variety of image and text datasets are provided in Appendix A.

even when token activation maps are misleading or ambiguous (see Figure 1). We term this behavior the **SuperActivator Mechanism** and show it is a general property of how transformers encode semantics. Our analysis demonstrates that this mechanism more accurately detects concepts than standard concept-vector and prompting methods across various image and text modalities, model architectures, model layers, and concept extraction techniques. We also show that leveraging these localized signals leads to improved concept attributions.

Our key contributions are summarized as follows:

- **SuperActivator Mechanism:** By analyzing the global concept activation distributions, we discover that the highly activated tokens in the tail of the true-concept distribution are reliable indicators of concept presence. Using just a small set of these extreme activations, our method consistently outperforms standard vector and prompt-based concept detection methods, consistently yielding improved $F_1$ scores by up to 14%.
- **Broad Generality:** We show the SuperActivator Mechanism is a fundamental property of how transformers encode semantics, consistent across text and image modalities, model architectures, model layers, and both supervised and unsupervised concept extraction techniques.
- **Application for Improved Concept Attributions:** By localizing concept signals with the Super-Activator Mechanism, we obtain attribution maps with stronger alignment to ground-truth annotations and superior insertion/deletion performance relative to global concept-vector baselines.

## 2 CONCEPT VECTOR PRELIMINARIES

This section introduces the basic formalism for representing inputs, defining concept vectors, and computing activation scores; additional mathematical details are provided in Appendix D.

Let $f$ be a trained transformer model that processes an input sample $x \in \mathcal{X}$ (an image or a text sequence) through its layers. From any given layer of $f$, we can extract token-level embeddings $(z_1^{\text{tok}}(x), \ldots, z_{n(x)}^{\text{tok}}(x)) \in (\mathbb{R}^d)^{n(x)}$ and a sample-level embedding $z^{\text{cls}}(x) \in \mathbb{R}^d$. The number of tokens, $n(x)$, is sample dependent since it is influenced by text lengths and image sizes. For any semantic concept $c$, we associate a **concept vector** $v_c \in \mathbb{R}^d$, which represents a direction in the embedding space (see Section 4.1 for extraction methods). The concept activation score of an embedding $z$ with respect to concept $c$ is defined as $s_c(z) = \langle z, v_c \rangle$, where positive scores indicate alignment with the concept.

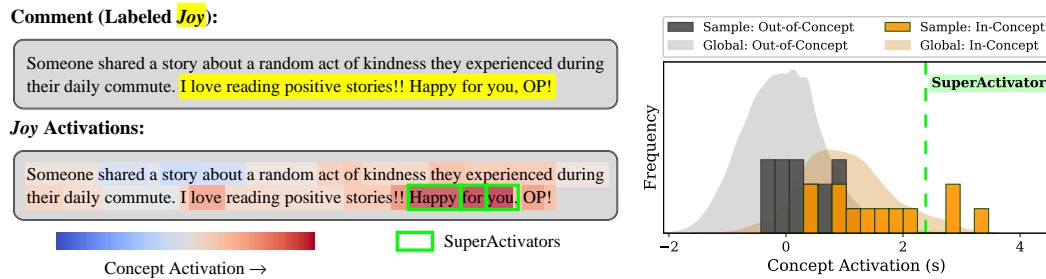

Figure 2: Transformers distribute concept signals unevenly across ground-truth regions, leading to substantial overlap between the concept-positive activation scores and $\mathrm{supp}(D_c^{\mathrm{out}})$. In this example from the *Augmented GoEmotions* dataset, the ground-truth span for *joy* is highlighted in a Reddit comment, with token-level activations from a Llama-Vision-Instruct model shown both as a heatmap over the text and as distributions. While a few true-concept tokens (separated by a blue dotted line) exhibit extremely high activations, most remain indistinguishable from non-concept tokens within the sample and across the global test set ($\mathrm{supp}(D_c^{\mathrm{out}})$).

We are interested in characterizing concept activation scores globally across many samples. Therefore, for each concept $c$ we define the *in-concept distribution* $D_c^{\mathrm{in}}$ as the collection of activation scores from tokens labeled concept-positive for $c$, and the *out-of-concept distribution* $D_c^{\mathrm{out}}$ as those from tokens labeled concept-negative. Formally, let $Z$ denote the set of tokens across samples and $S_c = \{\, s_c(z) : z \in Z \,\}$ the corresponding collection of activation scores. If $Z_c^{\mathrm{in}} \subseteq Z$ are the tokens containing $c$ and $Z_c^{\mathrm{out}} = Z \setminus Z_c^{\mathrm{in}}$, then

$$D_c^{\mathrm{in}} = \{\, s_c(z) : z \in Z_c^{\mathrm{in}} \,\}, \qquad D_c^{\mathrm{out}} = \{\, s_c(z) : z \in Z_c^{\mathrm{out}} \,\}.$$

Note that $Z_c^{\mathrm{out}}$ excludes *all* tokens from samples containing $c$, even those not labeled with the concept, in order to prevent self-attention from leaking concept information into the out-of-concept distribution.

The *support* of a distribution is the set of values with nonzero probability, and the *tail* refers to its extreme regions with small probability mass. To quantify how much $D_c^{\mathrm{in}}$ and $D_c^{\mathrm{out}}$ overlap, we use the *overlap coefficient (OVL)*, defined as the shared probability mass between the two distributions:

$$\mathrm{OVL}(D_c^{\mathrm{in}}, D_c^{\mathrm{out}}) = \int_{-\infty}^{\infty} \min\big(p^{\mathrm{in}}(s),\, p^{\mathrm{out}}(s)\big)\, ds,$$

where $p^{\mathrm{in}}$ and $p^{\mathrm{out}}$ are their densities. Large values of OVL indicate that most in-concept activations lie within the overlapping support $\mathrm{supp}(D_c^{\mathrm{in}}) \cap \mathrm{supp}(D_c^{\mathrm{out}})$ and are thus statistically indistinguishable from out-of-concept activations, whereas small values arise when only the high-activation tail of $D_c^{\mathrm{in}}$ extends beyond $D_c^{\mathrm{out}}$, yielding clearer separation.

One primary application of concept activation scores is **concept detection** (Wu et al., 2025; Rückert et al., 2023; Groza et al., 2024), which aims to determine whether a concept is present in a sample. Standard methods apply an aggregation operator $G : \mathbb{R}^{n(x)+1} \to \mathbb{R}$ to obtain a per-sample concept activation score:

$$s_c^{\mathrm{agg}}(x) = G\big(s_c(z_1^{\mathrm{tok}}(x)), \ldots, s_c(z_{n(x)}^{\mathrm{tok}}(x)), s_c(z^{\mathrm{cls}}(x))\big).$$

The concept is considered detected if $s_c^{\mathrm{agg}}(x)$ exceeds a threshold, typically obtained via calibration. There is no consensus on the best choice of aggregation operator $G$. Common strategies include using the score of the [CLS] token (Nejadgholi et al., 2022; Yu et al., 2024), applying mean (McKenzie et al., 2025; Benou & Riklin-Raviv, 2025) or max-pooling (Tillman & Mossing, 2025; Wu et al., 2025), or using the score of the last token (Chen et al., 2025; Tillman & Mossing, 2025).

Concept activations are also useful for **concept localization** (or attribution), which seeks to answer *where* a concept is located within a sample Santis et al. (2024). When evaluating concept localizations, we desire attribution maps that align with ground-truth annotations—segmentation masks for images or span-level labels for text. At the same time, attributions should be **faithful** (Zhang et al., 2023), meaning that they accurately reflect the features the model actually relies on.

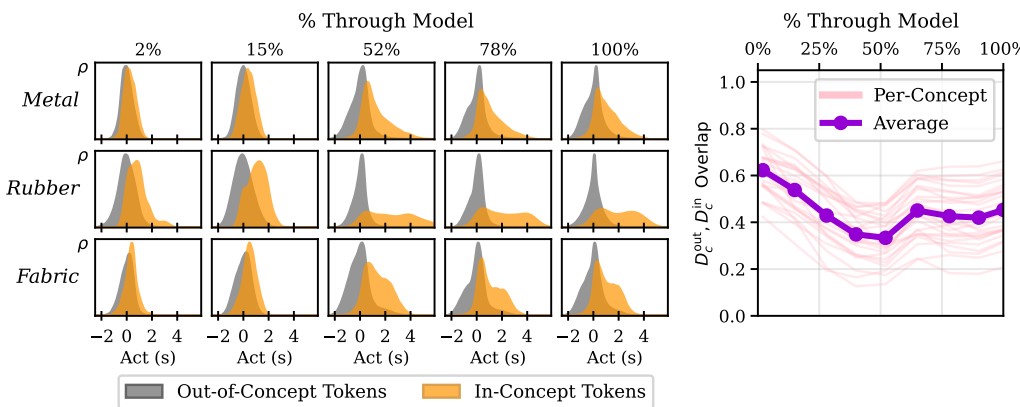

Figure 3: As concept signals evolve across transformer layers, $D_c^{\text{in}}$ and $D_c^{\text{out}}$ become more distinct with depth, though the separation is concentrated in a small subset of tokens in the tail of $D_c^{\text{in}}$. Shown here are activation distributions for linear separator concepts from LLAMA-3.2-11B-VISION-INSTRUCT on the *OpenSurfaces* dataset (*Metal*, *Rubber*, and *Fabric*); additional examples across datasets, models, and concept types are provided in Appendix B.

## 3 THE *SuperActivator* MECHANISM YIELDS CLEAR CONCEPT SIGNAL WITHIN NOISY CONCEPT ACTIVATIONS

### 3.1 CONCEPT ACTIVATIONS ARE NOISY AND INCONSISTENT

Concept vectors promise interpretability but they often deliver noisy activations that are hard to extract meaningful insights from. On the global image/sentence level, it is now well documented that concept vectors can encode spurious correlations and blur important context-specific distinctions (Abid et al., 2022; Zhou et al., 2021). These issues are further maintained at the local level of individual tokens leading to issues including entanglement (co-activation of related concepts) and polysemanticity (a single vector representing unrelated concepts) (Goh et al., 2021; Olah et al., 2020; Bricken et al., 2023).

We identify an additional challenge: transformers distribute concept signals non-uniformly across true-concept regions. This is illustrated in Figure 2 where a few tokens exhibit strong activations clearly aligned with the concept *Joy*, but many other positively labeled tokens have indistinguishable activations from those of non-concept tokens. As shown on the right of Figure 2, the true-concept token activations significantly overlap with non-concept activations, both within the sample and relative to the broader $\text{supp}(D_c^{\text{out}})$. Consequently, even if a few key tokens are correctly identified, a single global threshold cannot cleanly separate in-concept tokens from out-of-concept ones.

To understand how these noisy activations arise, we examine $D_c^{\text{in}}$ and $D_c^{\text{out}}$ across transformer layers. $D_c^{\text{out}}$ remains roughly normal and centered near zero, while early-layer $D_c^{\text{in}}$ overlaps heavily with it, yielding high $\text{OVL}(D_c^{\text{in}}, D_c^{\text{out}})$, as shown in Figure 3. With depth, overlap decreases and stabilizes in middle layers, consistent with prior findings that concept representations become more separable in intermediate layers and sometimes collapse in the final layer due to task-specific compression (Saglam et al., 2025; Yu et al., 2024; Dalvi et al., 2022).

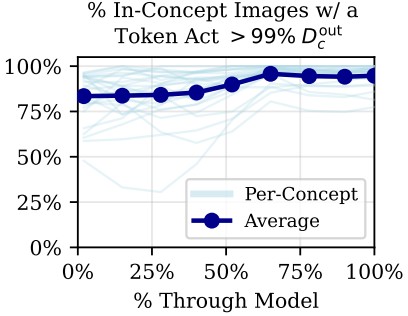

Figure 4: Most true-concept images in the *OpenSurfaces* dataset have at least one activation in the high-activation tail of $D_c^{\text{in}}$, well separated from $D_c^{\text{out}}$.

The separation between $D_c^{\text{in}}$ and $D_c^{\text{out}}$ does not arise from a uniform shift of all in-concept activations. Instead, while many scores remain overlapping with $\text{supp}(D_c^{\text{out}})$ and are thus indistinguishable from noise, $D_c^{\text{in}}$ develops a heavy tail as a small subset of extreme activations become increasingly separable with depth.

Notably, we find that the high-activation tail of $D_c^{\text{in}}$ provides good coverage: most true-concept samples contain at least one activation above this threshold. We define the tail as scores within $D_c^{\text{in}}$ that exceed the 99th percentile of $D_c^{\text{out}}$. This effect is shown for linear separator concepts on the *OpenSurfaces* dataset in Figure 4, and we show that it holds across datasets, models, and concept vector types in Appendix B.

## 3.2 INTRODUCING THE *SuperActivator* MECHANISM

A *reliable* concept signal should be *clear*, with activations that stand out from noise, and *accurate*, with high precision and broad coverage across true-concept samples. We find that such signals arise sparsely but consistently in the high-activation tail of $D_c^{\text{in}}$: they lie well outside $D_c^{\text{out}}$ (Figure 3) and appear in most concept-positive samples (Figure 4). These results hold cross modalities, architectures, and concept vector types, suggesting it is a general property of transformer representations.

We term this the **SuperActivator Mechanism**: a small set of extreme token activations carries the concept signal with both *clarity* (separation from $D_c^{\text{out}}$) and *coverage* (broad per-sample presence).

**Defining SuperActivators.** Let $\mathcal{S}_{\text{val},c}^{+} = \{\, s_c(z) : z \in Z_c^{\text{in}} \text{ from a validation set} \,\}$ be the empirical activation scores for concept $c$. For a sparsity level $N \in [1, 100]$, we define the *SuperActivator threshold* as

$$\tau_{c,N}^{\text{super}} \;=\; Q_{1-N/100}(\mathcal{S}_{\text{val},c}^{+}),$$

where $Q_q(S)$ denotes the $q$-quantile of a set of scores $S$. Tokens whose activations exceed this threshold form the set of SuperActivators,

$$\mathcal{T}_{c,N}^{\text{super}} \;=\; \{\, z \in Z_c^{\text{in}} : s_c(z) \geq \tau_{c,N}^{\text{super}} \,\}.$$

Intuitively, this means we are isolating the top $N\%$ of the in-concept distribution $D_c^{\text{in}}$, i.e. tokens in its high-activation tail.

**Leveraging SuperActivators for Concept Detection.** We develop a SuperActivator-based aggregator that predicts the presence of $c$ in a sample $x$ if it contains at least one SuperActivator for that concept. Concretely, we apply a max-pooling operator $G_{\max}$ over token activations, predicting concept presence if $G_{\max}(s_c(z_1^{\text{tok}}(x)), \ldots, s_c(z_{n(x)}^{\text{tok}}(x))) \;\geq\; \tau_{c,N}^{\text{super}}$.

This approach is closely related to the standard max aggregator (Wu et al., 2025; Xie et al., 2025), which we compare against in F. This design enables direct control over sparsity, letting us study how detection performance varies with $N$ (Appendix G). We find that SuperActivator detection is most effective at very low $N$, showing that the most reliable concept information is concentrated in a small high-activation tail of $D_c^{\text{in}}$. For final evaluation, we calibrate $N$ per-concept on the validation set to maximize detection $F_1$.

## 4 CONCEPT DETECTION AND LOCALIZATION WITH SUPERACTIVATORS

### 4.1 EXPERIMENTAL SETUP

We evaluate our framework across different modalities, models and concept types.

**Datasets.** Vision datasets include CLEVR (Johnson et al., 2017), COCO (Lin et al., 2014), and the PASCAL (Everingham et al., 2010) and OPENSURFACES (Bell et al., 2013) sections of the BRODEN dataset (Bau et al., 2020). For text, where token-level labels are scarce, we construct three datasets: SARCASM, AUGMENTED ISARCASM (Oprea & Magdy, 2020), and AUGMENTED GOEMOTIONS (Demszky et al., 2020). Full details are provided in Appendix C.3.

**Models.** For images, we extract both patch and CLS token embeddings from the CLIP ViT-L/14 (Radford et al., 2021) and LLAMA-3.2-11B-VISION-INSTRUCT (Meta, 2024). For text,

Table 1: **Our SuperActivator-based method outperforms concept vector-based and prompting baselines on concept detection $F_1$ scores**. The results shown here are for **linear separator concepts** using the *LLaMA-Vision-Instruct* model, where we improve performance by up to 14% over the best baseline. This trend generally holds across models and concept types, as detailed in Appendix E. **Bold** indicates the best score; underline marks the second best score.

| | Concept Detection Methods | | | | | |
| | RandTok | LastTok | MeanTok | CLS | Prompt | SuperAct |
| | | Chen et al. (2025) | (McKenzie et al., 2025) | (Yu et al., 2024) | (Wu et al., 2025) | (Ours) |
| CLEVR | 0.97 ± 0.09 | 0.88 ± 0.00 | 0.92 ± 0.00 | 0.96 ± 0.02 | 0.99 ± 0.01 | **1.00 ± 0.00** |
| COCO | 0.61 ± 0.01 | 0.68 ± 0.01 | 0.55 ± 0.01 | 0.57 ± 0.01 | 0.69 ± 0.05 | **0.83 ± 0.01** |
| Surfaces | 0.44 ± 0.01 | 0.41 ± 0.01 | 0.39 ± 0.01 | 0.46 ± 0.01 | 0.49 ± 0.06 | **0.56 ± 0.02** |
| Pascal | 0.66 ± 0.01 | 0.60 ± 0.01 | 0.59 ± 0.01 | 0.65 ± 0.01 | 0.68 ± 0.05 | **0.82 ± 0.01** |
| Sarcasm | 0.66 ± 0.06 | 0.68 ± 0.05 | 0.66 ± 0.06 | 0.74 ± 0.06 | 0.68 ± 0.07 | **0.87 ± 0.04** |
| iSarcasm | 0.89 ± 0.04 | 0.72 ± 0.03 | 0.79 ± 0.03 | 0.91 ± 0.03 | 0.79 ± 0.05 | **0.92 ± 0.03** |
| GoEmot | 0.37 ± 0.03 | 0.31 ± 0.03 | 0.19 ± 0.03 | 0.32 ± 0.03 | 0.25 ± 0.10 | **0.46 ± 0.03** |

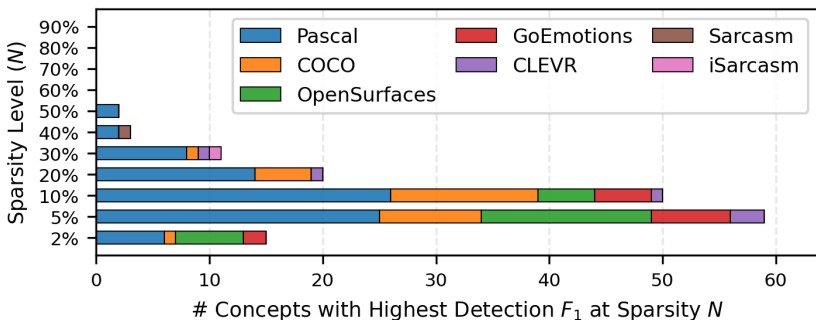

Figure 5: **SuperActivator-based concept detection is most effective when using only a small fraction of the most highly activated SuperActivators (5–10% of tokens).** Shown here are the numbers of **linear separator concepts** from *LLaMA-Vision-Instruct* across datasets that achieve their strongest $F_1$ scores at each sparsity level $N$; comprehensive results appear in Appendix E.

we use LLaMA-3.2-11B-VISION-INSTRUCT, GEMMA-2-9B (Team et al., 2024), and QWEN3-EMBEDDING-4B (Zhang et al., 2025). Since these models lack an explicit [CLS] token for text inputs, we approximate a [CLS]-style representation by averaging token embeddings, a strategy found to be quite effective (Choi et al., 2021; Tang & Yang, 2024; Dosovitskiy et al., 2020; Reimers & Gurevych, 2019).

**Concept Types.** We compute concepts at both the input token and [CLS]-level using the methods detailed in Appendix C.2: (1) mean prototypes (Zou et al., 2023), (2) labeled linear separators (Kim et al., 2018), (3) $k$-means (Ghorbani et al., 2019; Dalvi et al., 2022), (4) cluster-based separators (clusters as pseudo-labels), and (5) Sparse Autoencoders (Bricken et al., 2023). We incorporate the unsupervised concepts into our evaluation by matching each ground-truth concept with the discovered concept that is most reliable at detecting it. All methods in the following experiments make use of the same underlying concept vectors; detection strategies differ only in how activations are aggregated, while localization strategies generate attributions with respect to the same vectors.

### 4.2 SUPERACTIVATORS ARE RELIABLE INDICATORS OF CONCEPT PRESENCE

We now demonstrate that the SuperActivator tokens serve as more reliable indicators of concept presence than both concept-vector baselines and prompting methods.

We compare against several baseline aggregation strategies: $G_{\text{CLS}}$, which selects the [CLS] activation; $G_{\text{mean}}$, which averages input token activations; $G_{\text{last}}$, which selects the final input token activation; and $G_{\text{rand}}$, which selects a random token activation. We also include a prompting base-

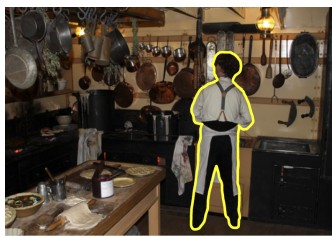 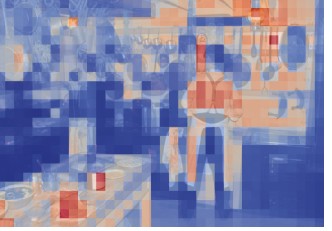 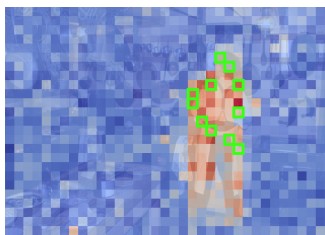

(a) Original Image    (b) Global Concept 'Person'    (c) SuperActivators for 'Person'

Figure 6: **SuperActivators produce attribution masks that align more closely with the ground truth concept region.** In (a), the yellow outline denotes the ground truth mask for the concept *person*. Compared to the Global Concept Objective (b), which yields noisier maps that miss parts of the person and highlight irrelevant regions, the SuperActivators Objective (c) provides local attribution. The green boxes in (c) mark the SuperActivators for the concept *person*, with their average embedding used for the objective. Results are shown for LIME-based attribution on the COCO dataset using the LLaMA model. Red indicates positive contributions and blue negative contributions.

line, where *LLaMA-Vision-Instruct* is directly queried about the presence of each concept, bypassing concept vectors altogether (Wu et al., 2025; Robicheaux et al., 2025; Tillman & Mossing, 2025).

For each concept, the model layer is calibrated on the validation set to maximize F1-score, and final results are averaged across concepts weighted by the number of test samples. This follows prior work showing that concepts become more or less distinguishable at different layers (Dorszewski et al., 2025; Alain & Bengio, 2018; Arps et al., 2022), so we select the best-performing layer per concept independently for all baselines (except prompting). To make this computationally feasible, calibration is performed over a fixed grid of layers (see Appendix C.1 for details).

As shown in Table 1, our SuperActivator method consistently outperforms all other detection strategies on linear separator concepts from the LLaMA-Vision-Instruct model. Prompting is typically the next strongest method, with CLS-token aggregators also showing competitive performance in certain settings.

Notably, Figure 5 shows that performance typically peaks when using only a very small fraction of the most activated tokens—2–10% for COCO, OPENSURFACES, and GOEMOTIONS, while IS-ARCASM peaks at a moderately higher 40%. This pattern highlights that only a sparse subset of tokens carry the strongest and most reliable concept information; including additional, weaker activations introduces noise from overlapping supports with $D_c^{\text{out}}$, which dilutes performance rather than improving it. We note one nuance with Sparse Autoencoder concepts, where peak performance occurs at higher $N$ levels, likely because SAEs already enforce sparsity during training. Detailed SAE-specific results and discussion are provided in Appendix N.1.

Tuning $N$ enabled us to experimentally validate that the most reliable concept signals lie in the extreme in-concept tail. Leveraging this insight, we evaluate a more practical detection procedure that fixes N at the tail in Appendix L. We simply set $N = 10\%$—a sparsity level that performs well across all concepts generally (see Appendix G)—and retain only the top-activated tokens per sample for each concept. Using *only* sample-level labels, we then train a threshold on these selected activations to separate those from in-concept and out-of-concept samples. This fixed-$N$ detector nearly matches the performance of the fully tuned SuperActivator method and outperforms all other baselines across datasets, providing a simple and effective way to leverage the highly informative tail for concept detection.

We perform several ablations to analyze how SuperActivator-based detection behaves across layers and sparsity levels. Appendix F shows heatmaps of average detection $F_1$ (weighted across concepts) for each model and dataset as a function of model depth, providing a global view of where concept signals are strongest. Appendix I summarizes the distribution of best-performing layers across concepts, revealing how different concepts peak at varying depths. To study sparsity, Appendix H reports histograms of optimal sparsity levels $\delta$ across model layers, while Appendix G plots $F_1$ as a function of $\delta$ at each concept's best-performing layer, showing how average Super-Activator detection performance varies with sparsity. Appendix J further analyzes the distribution

Table 2: **SuperActivators yield more accurate and faithful attributions than global concept vectors.** Accuracy is measured by attribution $F_1$ (alignment with ground-truth masks), while faithfulness is measured by insertion scores ($\uparrow$ is better) and deletion scores ($\downarrow$ is better). Results are shown for COCO (images) with CLIP and iSarcasm (text) with Gemma, comparing LinSep–Concept with SuperActivators. Similar patterns hold across other image datasets (CLEVR, OpenSurfaces, Pascal) and text datasets (Sarcasm, GoEmotions).

| Attribution Method | Dataset | Attribution $F_1$ ($\uparrow$ is better) | | Insertion Score ($\uparrow$ is better) | | Deletion Score ($\downarrow$ is better) | |
|---|---|---|---|---|---|---|---|
| | | Concept | Super Activators | Concept | Super Activators | Concept | Super Activators |
| LIME | COCO | 0.29±0.02 | **0.40±0.03** | 0.333±0.009 | **0.367±0.008** | 0.010±0.001 | **0.007±0.001** |
| (Ribeiro et al., 2016) | iSarcasm | 0.76±0.02 | **0.89±0.01** | 0.383±0.008 | **0.412±0.009** | 0.009±0.000 | **0.005±0.004** |
| SHAP | COCO | 0.35±0.01 | **0.37±0.02** | 0.334±0.004 | **0.365±0.004** | 0.010±0.001 | **0.008±0.002** |
| (Lundberg & Lee, 2017) | iSarcasm | 0.77±0.03 | **0.90±0.02** | 0.384±0.008 | **0.410±0.003** | 0.009±0.001 | **0.006±0.001** |
| RISE | COCO | 0.35±0.02 | **0.38±0.03** | 0.328±0.004 | **0.354±0.007** | 0.012±0.002 | **0.009±0.000** |
| (Petsiuk et al., 2018) | iSarcasm | 0.81±0.01 | **0.94±0.03** | 0.382±0.005 | **0.409±0.009** | 0.008±0.001 | **0.005±0.002** |
| SHAP IQ | COCO | 0.34±0.01 | **0.37±0.01** | 0.330±0.005 | **0.358±0.008** | 0.011±0.002 | **0.009±0.001** |
| (Fel et al., 2023) | iSarcasm | 0.79±0.02 | **0.92±0.01** | 0.379±0.004 | **0.407±0.004** | 0.009±0.001 | **0.006±0.001** |
| IntGrad | COCO | 0.28±0.00 | **0.35±0.04** | 0.326±0.003 | **0.359±0.005** | 0.013±0.003 | **0.010±0.003** |
| (Sundararajan et al., 2017) | iSarcasm | 0.72±0.02 | **0.84±0.01** | 0.375±0.004 | **0.405±0.009** | 0.011±0.001 | **0.008±0.003** |
| GradCAM | COCO | 0.37±0.01 | **0.38±0.02** | 0.329±0.005 | **0.352±0.004** | 0.012±0.003 | **0.010±0.001** |
| (Selvaraju et al., 2017) | iSarcasm | 0.74±0.02 | **0.87±0.03** | 0.377±0.004 | **0.403±0.008** | 0.010±0.001 | **0.007±0.001** |
| FullGrad | COCO | **0.43±0.01** | **0.43±0.00** | 0.331±0.006 | **0.357±0.010** | 0.011±0.001 | **0.009±0.002** |
| (Srinivas & Fleuret, 2019) | iSarcasm | 0.73±0.03 | **0.85±0.01** | 0.376±0.005 | **0.402±0.010** | 0.010±0.001 | **0.007±0.001** |
| CALM | COCO | **0.42±0.01** | **0.42±0.01** | 0.332±0.010 | **0.360±0.004** | 0.011±0.002 | **0.008±0.000** |
| (Mahajan et al., 2021) | iSarcasm | 0.78±0.01 | **0.91±0.02** | 0.380±0.007 | **0.408±0.004** | 0.009±0.001 | **0.006±0.001** |
| MFABA | COCO | 0.33±0.01 | **0.39±0.03** | 0.339±0.005 | **0.374±0.006** | 0.006±0.001 | **0.004±0.001** |
| (Srinivas & Fleuret, 2019) | iSarcasm | 0.77±0.02 | **0.90±0.03** | 0.391±0.002 | **0.420±0.009** | 0.006±0.001 | **0.003±0.001** |

of SuperActivators within each sample using cumulative distribution functions, showing that only a small fraction of in-concept tokens tend to be SuperActivators. Finally, Appendix K evaluates positional dependencies and shows that SuperActivators do not depend on token position.

Across image and text datasets, model architectures, and concept vector types, the same pattern emerges: the most reliable concept signals reside in the sparse, high-activation tail of $D_c^{\text{in}}$. The SuperActivator Mechanism thereby reflects a core principle of how transformers represent semantics.

### 4.3 SuperActivators Improve Attributions for Concepts

Standard concept attribution typically evaluates relevance with respect to a single global concept vector aggregated over many samples. While this captures broad concept information, it often blurs local context and introduces spurious correlations. In contrast, SuperActivators provide more consistent concept signals for detection (see Section 4.2), are tied to the specific local context of each sample, and avoid averaging across disparate occurrences. We hypothesize that using SuperActivators as the attribution objective improves attribution across three metrics: accuracy measuring average $F_1$ against ground truth, and insertion and deletion score based on the faithfulness metric.

To test this, we compare two attribution objectives: (1) the standard global concept vector and (2) our proposed method, which averages the embeddings of local SuperActivators within each instance.

We generate attribution maps following the standard procedures described in Appendix M.1, where attribution scores estimate each token's effect on changes in a given objective. Conventional concept attribution methods use the alignment between token embeddings and the global concept vector as this objective. We introduce one key modification: attribution is computed relative to the mean embedding of local SuperActivators. Each SuperActivator is defined using the sparsity level $\delta$ that achieves the highest detection $F_1$ score on the validation set. For each concept $c$, attribution scores are then binarized into $c$-positive or $c$-negative using the threshold that maximizes validation $F_1$. If a sample contains no SuperActivators associated with concept $c$, all tokens are assigned as $c$-negative.

This approach yields attributions more closely aligned with ground-truth segmentation masks than global concept vectors. Across datasets and attribution methods, local SuperActivators consistently

improve $F_1$, outperforming the global baseline on both COCO and ISARCASM (Table 2), with similar gains across four image and three text datasets (Tables 5–11). Figure 6 illustrates this advantage: SuperActivators for *person* provide more complete coverage of the target object while avoiding irrelevant regions incorrectly highlighted by the global vector. In addition, SuperActivators-based attributions consistently achieve higher insertion and lower deletion scores than global vectors, demonstrating improved attribution based on the faithfulness metric (Table 2).

These findings persist in the unsupervised setting, where clusters that best detect ground-truth concepts in the detection phase also produce higher attribution $F_1$ when explanations are generated using SuperActivators, with consistent improvements observed across all datasets (Tables 12–18).

## 5 RELATED WORK

**Concept-Based Interpretability:** Concept-based interpretability links model internals with human-understandable features. Approaches include defining concept vectors as linear separators (e.g., TCAV; (Kim et al., 2018)), or as centroid embeddings from labeled examples (Zou et al., 2023). Unsupervised discovery methods include ACE (Ghorbani et al., 2019), hierarchical clustering (Dalvi et al., 2022), matrix factorization approaches (Zhang & Zhang, 2017; Fel et al., 2022), and sparse autoencoders (Cunningham et al., 2023; Gao et al., 2024a). Across these works, concepts are assumed to be recoverable as structured vectors, clusters, or basis elements within representation space.

**Challenges in Concept Representations:** Many open questions remain concerning the structure of concept representations. The linearity hypothesis posits that concepts correspond to directions in activation space, linearly separable and recoverable with simple probes (Mikolov et al., 2013; Elhage et al., 2022). Empirically, however, activations are often *entangled*, firing on tokens or samples where the concept is absent or bleeding into related but unintended semantics (Goh et al., 2021; Olah et al., 2020), *polysemantic*, where a single neuron or direction encodes multiple features (Bricken et al., 2023; O'Mahony et al., 2023), and *unstable*, with concept signals shifting across layers, spatial locations, exemplar sets, and random seeds (Wu et al., 2025; Mahinpei et al., 2021; Nicolson et al., 2025; Mikriukov et al., 2023). These properties can amplify failure modes such as spurious correlations (Zhou et al., 2024b) and concept leakage (Parisini et al., 2025), undermining both detection and attribution. In response, some approaches enforce more interpretable or disentangled concept structures (Chen et al., 2020; Wang et al., 2024). Our work takes a different perspective: rather than redesigning representations, we identify a sparse and reliable signal that already exists within otherwise noisy activation distributions.

**Concept Detection:** Concept detection is a central task in concept-based interpretability (Wu et al., 2025), with practical importance wherever one needs to determine whether a given concept is present in a sample—for example, detecting clinical or radiological concepts in medical images and reports (Rückert et al., 2023; Groza et al., 2024) or identifying undesirable online behavior (Liu et al., 2023; Nejadgholi et al., 2022). Most approaches instantiate a concept as a vector (e.g., a prototype or separator) and then score a sample by its alignment to that vector. This can be done using a *global* representation—such as the [CLS] token or pooled embeddings—which can be effective but often dilute sparse, fine-grained signals (Choi et al., 2021; Tang & Yang, 2024). When token or patch embeddings are available, methods instead compute token-level activations and aggregate them into a single alignment score; common choices include [CLS]-based scoring (Nejadgholi et al., 2022; Stein et al., 2024; Yu et al., 2024; Behrendt et al., 2025), mean pooling (McKenzie et al., 2025; Benou & Riklin-Raviv, 2025; Suresh et al., 2025), max pooling (Tillman & Mossing, 2025; Wu et al., 2025; Lim et al., 2025; Xie et al., 2025), or last-token scoring (Chen et al., 2025; Tillman & Mossing, 2025; Tang & Yang, 2024). Beyond vector scoring, *concept bottleneck models* implicitly encode detection within a supervised concept layer designed for downstream tasks (Koh et al., 2020). More recently, high-performing vision–language models have enabled *zero-shot prompting* that bypasses explicit concept vectors altogether, with strong results from CLIP and newer multimodal LMs (e.g., GPT-4o-mini) (Wu et al., 2025; Robicheaux et al., 2025; Tillman & Mossing, 2025).

**Feature Attributions for Concepts:** Feature attributions for concept tell us *where* a concept is located within a sample Santis et al. (2024), which is useful for tasks such as debugging spurious correlations Wu et al. (2023). Traditional attribution methods such as Integrated Gradients (Sundararajan et al., 2017) and Grad-CAM (Selvaraju et al., 2017), along with concept-based adaptations (Kim et al., 2018; Santis et al., 2024; Yu et al., 2024; Fel et al., 2022), have been used to connect

predictions to concepts. Beyond these, various works generate localization maps via direct alignment with raw activation scores (Benou & Riklin-Raviv, 2025; Lim et al., 2025; Zhou et al., 2024a; Lim et al., 2025) and attention values (Gandelsman et al., 2023). Recent work extends CAVs to concept-level feature attribution, by producing sample-level localization maps (Shukla et al., 2023), and improving localization stability through cross-layer CAVs (He et al., 2025).

# 6 DISCUSSION AND FUTURE WORK

In this work, we introduced and characterized the SuperActivator Mechanism, demonstrating that transformers concentrate reliable concept evidence into a sparse set of highly activated tokens. Leveraging this property enabled us to cut through the noise of globally aggregated concept vector activations and uncover more reliable signals of concept presence, which in turn serve as a stronger basis for concept localization. In the future, investigating how SuperActivators arise during training may provide deeper insight into how this mechanism emerges. Moreover, applying these principles in real-world settings for improved concept detection and localization offers the potential to make model interpretability more actionable in practice.

## ACKNOWLEDGMENTS

We acknowledge the use of large language models (ChatGPT, Gemini, and Claude Code) to assist with text drafting and editing, as well as code generation and debugging. All content was reviewed and revised by the authors to ensure accuracy and clarity.

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

## A SuperActivator Visual Examples

This section presents visual examples of SuperActivators in test samples across multiple image and text datasets. The heatmaps illustrate the activation score between the token embeddings and the labeled concept vectors, where red indicates high alignment, blue indicates low alignment, and a green rectangle indicates SuperActivators. The concepts used in these visualizations are linear separators trained on *LLaMA-3.2-11B-Vision-Instruct* embeddings at the model depth that achieved the highest validation performance, with SuperActivators defined at the sparsity level $\delta$ that yielded the best validation $F_1$ for each concept.

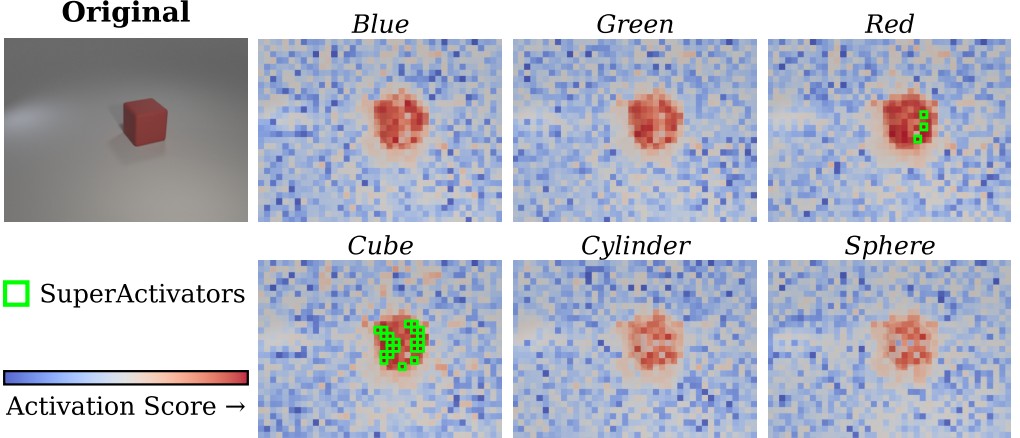

Figure 7: *CLEVR* – Visualization of Concept Activations and SuperActivators

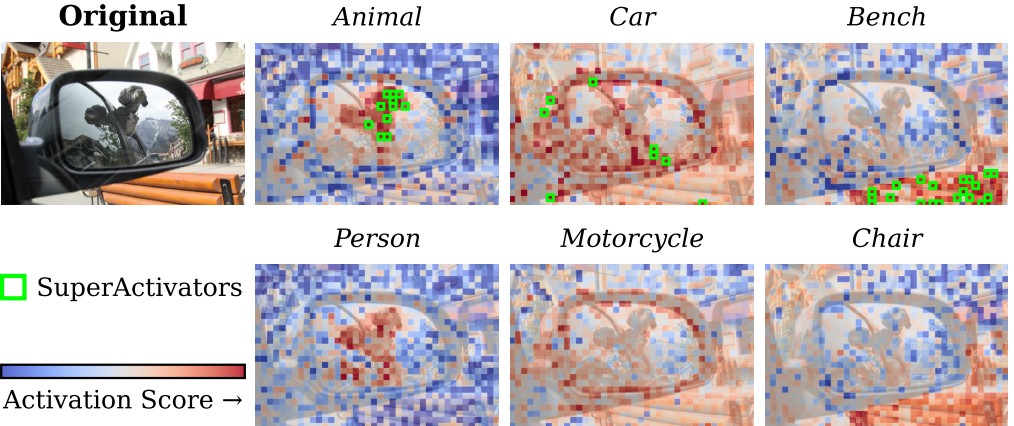

Figure 8: *COCO* – Visualization of Concept Activations and SuperActivators

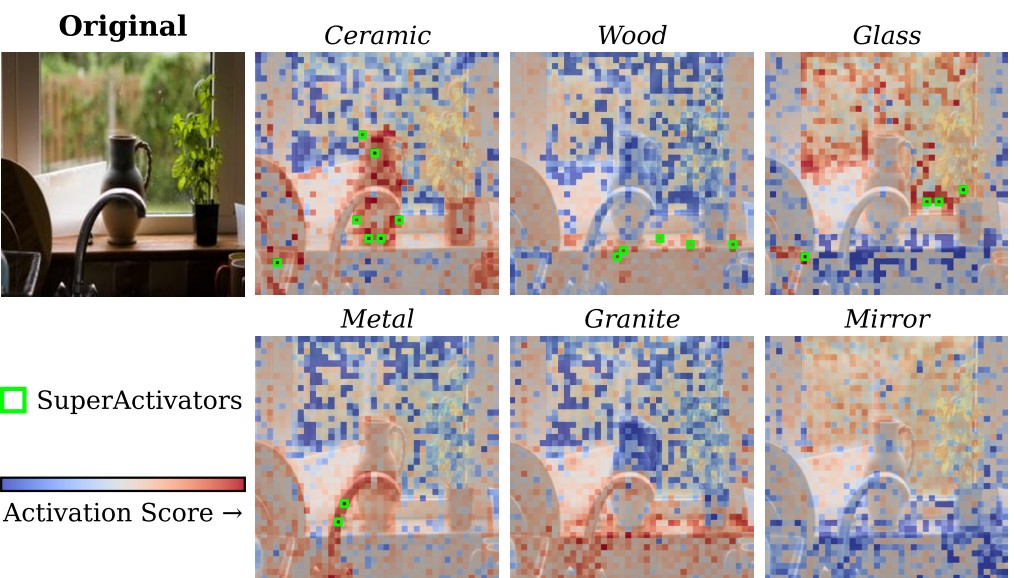

Figure 9: *OpenSurfaces* – Visualization of Concept Activations and SuperActivators

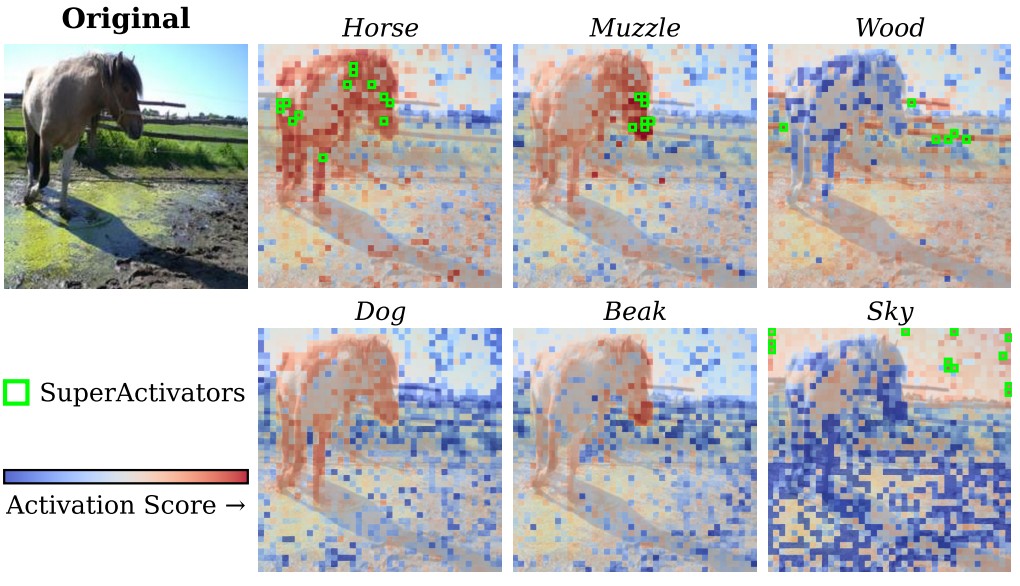

Figure 10: *Pascal* – Visualization of Concept Activations and SuperActivators

Original Text (No Labeled Concept):

Regrettably, my morning coffee spilled all over my fresh white shirt. I was running late for work and in my rush, I knocked my coffee mug right off the counter. Thankfully, I had a spare shirt in my car.

*Sarcasm Activations:*

Regrettably, my morning coffee spilled all over my fresh white shirt. I was running late for work and in my rush, I knocked my coffee mug right off the counter. Thankfully, I had a spare shirt in my car.

(a) Non-Sarcastic Version

Original Text (Sarcasm highlighted):

It's such a treat when my morning coffee decides to spill all over my fresh white shirt. I was running late for work and in my rush, I knocked my coffee mug right off the counter. Thankfully, I had a spare shirt in my car.

*Sarcasm Activations:*

It's such a treat when my morning coffee decides to spill all over my fresh white shirt. I was running late for work and in my rush, I knocked my coffee mug right off the counter. Thankfully, I had a spare shirt in my car.

Activation Score →  □ SuperActivators

(b) Sarcastic Version

Figure 11: *Sarcasm* – Visualization of Concept Activations and SuperActivators (sarcastic and non-sarcastic version of same sentiment)

Original Text (No Labeled Concept):

the worst way to wake up is when the alarm is too loud. it makes me feel really startled first thing in the morning. #NeedCoffee

*Sarcastic Activations:*

the worst way to wake up is when the alarm is too loud. it makes me feel really startled first thing in the morning. #NeedCoffee

(a) Non-Sarcastic Sample

Original Text (Sarcastic highlighted):

there's no better way to wake up than having one dog jump directly on your stomach and knock the wind out of you while the other drop a dead rodent on the end of the bed. i really need to start closing the bedroom door at night. #morningchaos

*Sarcastic Activations:*

there's no better way to wake up than having one dog jump directly on your stomach and knock the wind out of you while the other drop a dead rodent on the end of the bed. i really need to start closing the bedroom door at night. #morningchaos

Activation Score →  □ SuperActivators

(b) Sarcastic Sample

Figure 12: *Sarcasm* – Visualization of Concept Activations and SuperActivators (non-sarcastic and sarcastic text samples)

Original Text (*Anger* highlighted):

WHAT THE HELL! I opened up the new software update, and it seems like they've moved all the settings around again.

*Anger Activations:*

WHAT THE HELL! I opened up the new software update, and it seems like they've moved all the settings around again.

*Love Activations:*

WHAT THE HELL! I opened up the new software update, and it seems like they've moved all the settings around again.

*Gratitude Activations:*

WHAT THE HELL! I opened up the new software update, and it seems like they've moved all the settings around again.

Activation Score → □ SuperActivators

Figure 13: *Augmented GoEmotions* SuperActivator Example

## B  MOTIVATION FOR SUPERACTIVATOR

In this section, we motivate our focus on the highly-aligned activations in the tail of the in-concept activation distribution, $\mathcal{D}_c^{\text{in}}$. For this initial inquiry, we consider a token separable from the empirical out-of-concept activation distribution $D_c^{\text{out}}$ if its concept activation is greater than $99\%$ of the out-of-concept token activations, $q_{0.99}(D_c^{\text{out}})$. Then, for each dataset, on the left we plot the percent of in-concept token activations that are separable from out-of-concept activations (averaged across concepts) as a function of model depth. On the right, we plot the percentage of in-concept samples (images, comments, tweets, etc) that contain at least one token that is separable from the out-of-concept distribution as a function of model depth (again, averaged across concepts). In Figure 14, we report results across various datasets and models, as well as both average and linear separator concept vectors.

Generally, as shown in the leftmost plots, the percentage of well-separated in-concept token activations gradually increases throughout the model. However, the majority of the in-concept token activations typically do not exceed $q_{0.99}(D_c^{\text{out}})$ even at the most distinguishing layers, indicating a fundamental problem with separability. This problem is particularly severe for the text datasets. For the image concepts, most of the true-concept images have at least one well-separated token activation, and this separation generally also increases with model depth. In the text setting, while not all in-concept samples contain an activated patch, a substantial proportion do—indicating that some concept signal is present, albeit more diffuse. This likely reflects the specific text datasets used here, where concepts such as sarcasm and emotion are more subjective and nuanced than the object and texture annotations in image data. The main takeaway from these results is that across all image and text datasets, models, and concept types, there appears to be activations in the tail of $D_c^{\text{in}}$ that are well-separated from $D_c^{\text{in}}$ and carry signals of concept presence.

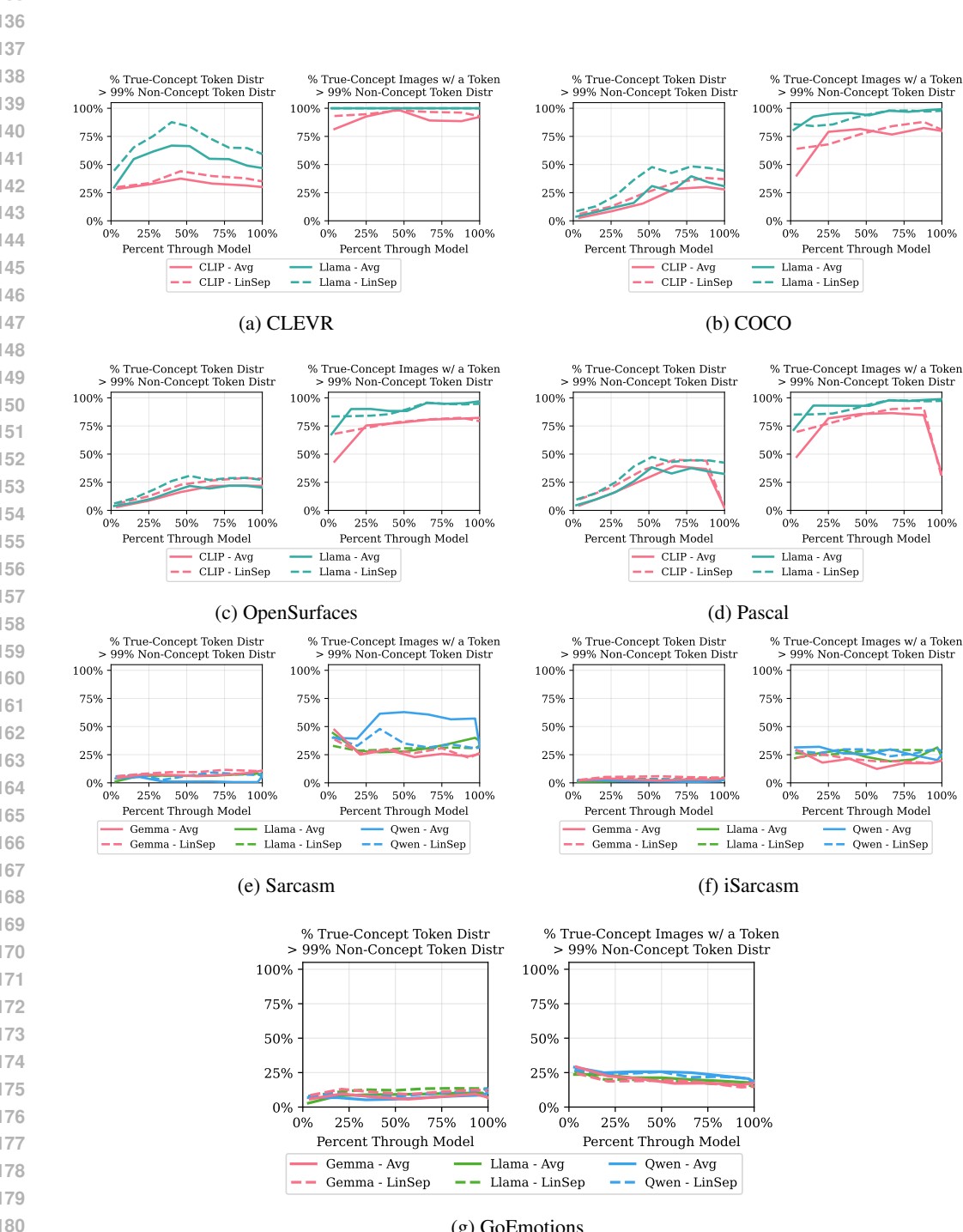

Figure 14: Across all image and text datasets, models, and concept types, there appears to be high magnitude in-concept activations that are well-separated from $D_c^{\text{in}}$ and carry signals of concept presence.

## C    EXPERIMENTAL CONFIGURATIONS

### C.1    EMBEDDING MODELS

We extract both input patch and [CLS] token embeddings from the CLIP ViT-L/14 (Radford et al., 2021) and LLAMA-3.2-11B-VISION-INSTRUCT (Meta, 2024). For text, we use LLAMA-3.2-11B-VISION-INSTRUCT, GEMMA-2-9B (Team et al., 2024), and QWEN3-EMBEDDING-4B (Zhang et al., 2025). Since these text models lack an explicit [CLS] token for text inputs, we approximate a [CLS]-style representation by averaging token embeddings (Choi et al., 2021; Tang & Yang, 2024; Dosovitskiy et al., 2020; Reimers & Gurevych, 2019). For each model, we obtain embeddings across multiple layers. To ensure comparability, we normalize and mean-center each layer's embeddings using statistics computed from the training set.

To make the computation feasible, we evaluate models at a fixed set of **default percentage depths through the network**, rather than at every layer. The chosen checkpoints are:

- **Vision Models:** CLIP: [4, 25, 46, 67, 88, 100]; LLaMA-Vision: [2, 15, 28, 40, 52, 65, 78, 90, 100]
- **Text Models:** LLaMA-Text: [3, 19, 34, 50, 66, 81, 97, 100]; Gemma: [4, 21, 39, 57, 75, 93, 100]; Qwen: [3, 19, 34, 50, 66, 81, 97, 100]

These default layer subsets balance coverage of early, middle, and late representations while avoiding the prohibitive costs of evaluating every model layer.

### C.2    CONCEPT EXTRACTION METHODS

Throughout, let $x$ denote a sample (image or text), and $z(x) \in \mathbb{R}^d$ its embedding obtained from the underlying model. For a ground-truth concept $c$, let $\mathcal{X}_c^+$ denote the set of samples labeled positive for $c$. We use $v_c \in \mathbb{R}^d$ to denote the concept vector associated with $c$, and $v_j$ to denote candidate concept vectors discovered by an unsupervised method. All concepts are constructed only using embeddings from the training set.

We extract concepts using supervised methods, unsupervised methods, and a prompting baseline. Concept representations are computed at both the token level, using embeddings from input tokens, and the [CLS] level, using embeddings from the [CLS] tokens, which lie in a distinct representational space optimized for sequence-level summarization.

**Supervised Methods:**

1. **Mean Prototypes** (Zou et al., 2023): Each concept vector is defined as the average embedding of all positive examples,

$$v_c = \frac{1}{|\mathcal{X}_c^+|} \sum_{x \in \mathcal{X}_c^+} z(x).$$

2. **Linear Separators (LinSep)** (Kim et al., 2018): For each concept $c$, we train a linear model (without bias) to distinguish positives from negatives. Training balances positive and negative samples and uses `BCEWithLogitsLoss` with the Adam optimizer (learning rate 0.01). We train for up to 100 epochs with a batch size of 32, apply weight decay of $1e{-}4$, and decay the learning rate by a factor of 0.5 every 10 epochs. Early stopping is used with a patience of 15 epochs and a tolerance of 3, which sets the minimum improvement required to continue training. The resulting normal vector of the separating hyperplane is used as the concept vector:
$$v_c = w_c.$$

**Unsupervised Methods:**

1. **K-Means Prototypes** (Ghorbani et al., 2019; Dalvi et al., 2022): We cluster embeddings using FAISS GPU (Johnson et al., 2019) with Euclidean distance, a maximum of 300 iterations, and $k{=}1000$ for token-level embeddings and $k{=}50$ for [CLS] embeddings. The

choice of $k$ was determined experimentally using an elbow curve. Token-level embeddings are finer-grained and therefore benefit from a larger number of clusters. Each cluster centroid is used as a concept vector:

$$v_j = \mu_j = \frac{1}{|\mathcal{C}_j|} \sum_{x \in \mathcal{C}_j} z(x).$$

2. **Cluster-Based Separators (K-LinSep)**: We first assign soft labels to embeddings based on their K-means cluster membership, then train linear separators with the same procedure described above to predict whether an embedding belongs to a given cluster. The normal vectors of these separators are treated as concept directions:

$$v_{ij} = w_{ij}.$$

3. **Sparse Autoencoders (SAEs)** (Bricken et al., 2023): SAEs learn a sparse reconstruction

$$z(x) \approx W h(x), \quad h(x) \in \mathbb{R}^m \text{ sparse}, \quad v_j = w_j,$$

where each column $w_j$ of $W$ corresponds to a candidate concept. Because SAE training is computationally expensive, we use pretrained SAEs; see Appendix N for architectural and implementation details.

To ensure we can evaluate against unsupervised methods, each ground-truth concept $c$ is matched to the unsupervised unit $v_j$ that achieves the highest validation $F_1$ score for detecting $c$:

$$v_c = \arg\max_{v_j} \ F_1^{\text{val}}(c, v_j).$$

**Prompt Baseline:** As a non-concept vector baseline, we query LLAMA-3.2-11B-VISION-INSTRUCT directly. For each sample $x$ and concept $c$, we prompt:

"Is the concept of $c$ present in the following? $x$".

Prior works have employed this baseline successfully (Wu et al., 2025; Robicheaux et al., 2025; Tillman & Mossing, 2025).

### C.3    DATASET OVERVIEW

**CLEVR (Single-Object) (Johnson et al., 2017):** A synthetic dataset of 1,000 images, each containing a red, green, or blue object with shape sphere, cylinder, or cube. Images and segmentation masks are generated programmatically, allowing fine-grained control over object properties and patch-level annotations.

**COCO (Lin et al., 2014):** We use the 2017 validation set, containing 5,500 images with everyday scenes involving people, objects, and natural contexts. Each image comes with human-annotated segmentations, providing dense labels for both object categories and broader supercategories.

**Broden–Pascal (Everingham et al., 2010) and Broden–OpenSurfaces (Bell et al., 2013):** We use 4,503 samples from Pascal and 3,578 samples from OpenSurfaces. These are subsets of the Broden dataset (Bau et al., 2020), which unifies multiple segmentation datasets into a single benchmark for concept-based interpretability research. Pascal primarily contains natural images with segmented objects from diverse categories such as animals, vehicles, and household items, while OpenSurfaces emphasizes fine-grained material and surface property annotations (e.g., wood, fabric, metal). These subsets focus on patch-level segmentation where concepts do not necessarily span the entire image.

**Sarcasm (Fully Synthetic):** We generate a dataset of 1,446 paragraphs, where roughly half contain exactly one sarcastic sentence surrounded by neutral sentences.

**iSarcasm (Augmented):** We adapt 1,734 samples from the original iSarcasm dataset (Oprea & Magdy, 2020), which provides sarcastic tweets alongside non-sarcastic rewrites conveying the same meaning (both provided by the original authors). We augment these by embedding sarcastic and non-sarcastic sentences into short paragraphs of neutral context, with sarcastic spans explicitly marked.

**GoEmotions (Augmented):** We use 5,427 samples from the GoEmotions dataset (Demszky et al., 2020), a human-annotated collection of Reddit comments labeled with 27 emotion categories. We augment selected samples by embedding emotional sentences within surrounding neutral context, tagging the emotional span while preserving natural paragraph flow.

### C.4 Text Augmentation Pipelines and Prompts

This section describes the augmentation pipelines used for generating and adapting text datasets, along with the exact prompts. Our goal was to create datasets with localized token-level concept spans, since most publicly available text datasets only provide unit-level (sentence, tweet, comment, etc) labels. Generation and augmentation are performed via controlled prompting of GPT-4o (OpenAI, 2024).

#### C.4.1 Sarcasm (Fully Synthetic)

**Pipeline:** We generate entirely new paragraphs containing exactly one sarcastic sentence. The sarcastic sentence is wrapped in <SARCASM> tags, while all other sentences are neutral. This ensures that each paragraph contains exactly one labeled sarcastic span, with natural context surrounding it. By constraining sarcastic content to a single line, we obtain a controlled setup where token-level supervision is precise and unambiguous.

**Prompt:**

```
Write 10 short paragraphs (4{8 sentences each). Each paragraph must include
**exactly one sarcastic sentence**, wrapped in <SARCASM> ... </SARCASM> tags.

Guidelines:
- The sarcastic sentence should be subtle, deadpan, or context-dependent.
- All other sentences must be sincere and literal.
- Vary topic, tone, and structure across paragraphs.

Only the sarcastic line may be wrapped in tags.

Return only the 10 numbered paragraphs.
```

**Example:** Jane always prided herself on her cooking abilities. <SARCASM>Indeed, the local fire department must have also appreciated her culinary exploits, given the number of times they've had to rush to her house.</SARCASM> Still, she was not deterred and continued to experiment in the kitchen, determined to perfect her skills. She understood that learning anything new involved a process of trial and error.

#### C.4.2 iSarcasm Augmentation

**Dataset Overview:** The original iSarcasm dataset contains sarcastic tweets paired with author-provided sincere rewrites conveying the same meaning. We extend this dataset synthetically by surrounding the sarcastic tweets with literal, neutral context, ensuring precise span-level supervision. Only sarcastic samples are selected for augmentation, and for each sarcastic input we generate both a sarcastic augmented post and a non-sarcastic rewrite.

**Augmentation Pipeline:** Each sarcastic input is expanded into casual, paragraph-like text using controlled prompting of GPT-4.0. To introduce variation, random structural features are applied:

- 20% chance of forcing a `[Sarcasm][Trigger]` structure.
- 15% chance of adding emojis or hashtags.
- Otherwise, a random choice among `[Sarcasm][Trigger]`, `[Trigger][Sarcasm]`, or `[Trigger][Sarcasm][Trigger]`.

**Sarcastic Augmentation Prompt:**

```
You are a data annotation machine. Your only goal is to produce perfectly literal
text that follows the rules. You must not be creative or clever. You must not
generate any figurative language outside of the provided tags.
```

```
Your Task:
You will be given a sarcastic tweet and its true meaning. Rewrite the tweet by
embedding it within a strictly literal train of thought that matches the original's
casual tone.

Structure: [Randomly choose or force specific structure]
[Optional emoji/hashtag instruction if selected]

Constraints Checklist:
- The tone is casual and informal.
- The added text is not redundant.
- Outside <SARCASM> tags is strictly literal and descriptive.
- The original sarcastic tweet is fully preserved within <SARCASM> tags.
- Output contains ONLY the final post.

Input Sarcastic Tweet: "{sarcastic_tweet}"
Sincere Meaning (for your context): "{rephrased_text}"

Your Output:
```

**Non-Sarcastic Augmentation Prompt.**

```
You are a data annotation machine. Your only goal is to produce perfectly literal
text that follows the rules. You must not be creative or clever. You must not
invent new details.

Your Task:
Take a sincere idea and expand it slightly into a personal, casual post,
remaining 100% faithful to the original meaning.

[Optional emoji/hashtag instruction if selected]

Constraints Checklist:
- The tone is casual and informal.
- The entire post is strictly literal and descriptive.
- No sarcasm, irony, overstatement, or rhetorical questions.
- The post must be 100% faithful to the meaning of the original idea.
- Output contains ONLY the final post.

Input Sincere Idea: "{rephrased_text}"

Your Output:
```

**Verification Process:**   Outputs are verified via flexible matching with progressively lenient checks: exact matching (case-insensitive), whitespace normalization, URL/punctuation removal, and word-overlap thresholds.  If all attempts fail, the original tweet is wrapped in <SARCASM> tags as a fallback.

**Example:**

> **Input sarcastic tweet:** "The only thing I got from college is a caffeine addiction."
> **Input sincere rephrase:** "College is really difficult, expensive, tiring, and I often question if a degree is worth the stress."
>
> **Sarcastic augmentation:**  "I just checked my calendar and saw how many assignments are due this week. ¡SARCASM¿the only thing i got from college is a caffeine addiction¡/SARCASM¿"
> **Non-sarcastic rewrite:** "college is really difficult. it's also expensive and tiring. sometimes i find myself questioning if getting a degree is worth all the stress."

### C.4.3 GoEmotions Augmentation

**Dataset Overview:** GoEmotions is a large-scale dataset of Reddit comments labeled with up to 27 fine-grained emotions. We extend it synthetically by surrounding the original emotional comment with strictly neutral filler context, ensuring the emotional span remains localized and clearly marked with <EMOTION> tags.

**Augmentation Pipeline:** Every comment in GoEmotions is augmented without filtering, following a two-step process:

1. **Step 1: Generation.** A "Neutral Filler Machine" prompt is used to generate five diverse neutral-context options embedding the original emotional comment.

2. **Step 2: Selection.** A "Grader" prompt evaluates the five drafts and selects the best single option according to neutrality and naturalness.

To increase variation, a random structure is sampled per comment:

- 50% chance: [Emotion][Context]
- 25% chance: [Context][Emotion]
- 25% chance: [Context][Emotion][Context]

**Step 1 — Neutral Filler Prompt:**

```
You are a Neutral Filler Machine. Your task is to generate neutral,
non-emotional text to surround a given Reddit comment.

Task:
- Preserve the original emotional comment exactly inside <EMOTION> tags.
- Generate five unique and diverse neutral contexts that flow naturally.
- All options must follow the required structure.

Constraints:
- Text outside <EMOTION> must be strictly neutral (no emotion leakage).
- Sound natural and casual like a Reddit post.
- No redundancy with the emotional comment.

Input Emotional Comment: "{emotional_comment}"
Primary Emotion(s): "{emotion_labels_str}"
Required Structure: "{structure_choice}"

Your Output: Five options, each in the correct structure.
```

**Step 2 — Selection Prompt.**

```
You are a data annotation quality assurance specialist.
Your task is to select the best draft among five options.

Checklist:
- Context must be strictly neutral (no emotions).
- Flow naturally as a Reddit comment.
- No contradiction or redundancy.
- Only output the single best final option.

Draft Options:
{draft_options}

Your Final, Best Output:
```

**Verification Process:** The augmented comments are verified using flexible string matching to ensure that the original text is preserved inside <EMOTION> tags. We allow up to five retry attempts with progressively lenient checks. If all attempts fail, the fallback is to wrap the original comment directly in <EMOTION> tags.

**Example:**

> **Original emotional comment (gratitude):** "I didn't know that, thank you for teaching me something today!"
>
> **Augmented output:** "A comment explained the process behind recycling plastics and how it affects the environment. ¡EMOTION¿I didn't know that, thank you for teaching me something today!¡/EMOTION¿"

### C.5 CONCEPTS USED IN EXPERIMENTS

For the MS-COCO, GoEmotions, and Broden datasets, we filter concepts using minimum sample thresholds (100–300 samples, depending on the dataset) to ensure sufficient data for reliable concept construction, though future work could examine SuperActivators in underfit settings. The semantics concepts used in our experiments is listed here:

- **CLEVR:** blue, green, red, cube, cylinder, sphere

- **COCO:** accessory, animal, appliance, bench, book, bottle, bowl, bus, car, chair, couch, cup, dining table, electronic, food, furniture, indoor, kitchen, motorcycle, outdoor, person, pizza, potted plant, sports, train, truck, tv, umbrella, vehicle

- **Broden–OpenSurfaces:** brick, cardboard, carpet, ceramic, concrete, fabric, food, fur, glass, granite, hair, laminate, leather, metal, mirror, painted, paper, plastic-clear, plastic-opaque, rock, rubber, skin, tile, wallpaper, wicker, wood

- **Broden–Pascal:** airplane, bicycle, bird, boat, body, book, building, bus, cap, car, cat, cup, dog, door, ear, engine, grass, hair, horse, leg, mirror, motorbike, mountain, painting, person, pottedplant, saddle, screen, sky, sofa, table, track, train, tvmonitor, wheel, wood, arm, bag, beak, bottle, box, cabinet, ceiling, chain wheel, chair, coach, curtain, eye, eyebrow, fabric, fence, floor, foot, ground, hand, handle bar, head, headlight, light, mouth, muzzle, neck, nose, paw, plant, plate, plaything, pole, pot, road, rock, rope, shelves, sidewalk, signboard, stern, tail, torso, tree, wall, water, windowpane, wing

- **Sarcasm:** sarcasm.

- **iSarcasm:** sarcastic.

- **GoEmotions:** confusion, joy, sadness, anger, love, caring, optimism, amusement, curiosity, disapproval, approval, annoyance, gratitude, admiration

## D CONCEPT FORMALISMS IN MORE DETAIL

We provide a detailed formalization of concept detection and activation aggregation strategies. We limit our analysis to transformer models given their demonstrated effectiveness across modalities.

**Model Representations.** Let $f$ be a trained transformer model that processes an input $x \in \mathcal{X}$ (an image or a text sequence) into a set of hidden representations. At a given layer $\ell$, we extract token-level embeddings

$$f_\ell(x) = \{\, z_1^{\text{tok}}(x), \ldots, z_{n(x)}^{\text{tok}}(x), z^{\text{cls}}(x) \,\}, \quad z_i^{\text{tok}}(x), z^{\text{cls}}(x) \in \mathbb{R}^d.$$

Here $z_i^{\text{tok}}(x)$ denotes the representation of the $i$-th token (or image patch), and $z^{\text{cls}}(x)$ denotes the [CLS]-style representation summarizing the full input.

**Concept Vectors and Activation Scores.** For any semantic concept $c$, we define a **concept vector** $v_c \in \mathbb{R}^d$, extracted via one of the techniques in Appendix C.2. Intuitively, $v_c$ represents a direction in embedding space along which the concept $c$ is encoded. The **activation score** of an embedding $z$ with respect to concept $c$ is defined as

$$s_c(z) = \langle z, v_c \rangle.$$

If $v_c$ is derived as a cluster centroid, this corresponds to cosine similarity (for normalized embeddings). If $v_c$ is derived from a linear separator, it corresponds to the signed distance from the separating hyperplane. Interpretively, $s_c(z)$ measures the alignment of $z$ with concept $c$: large positive values indicate that $z$ strongly encodes features associated with $c$, while negative values suggest opposition or absence.

We are further interested in characterizing these activation scores globally across many samples. For each concept $c$, we define the *in-concept distribution* $D_c^{\text{in}}$ as the collection of activation scores from tokens labeled concept-positive for $c$, and the *out-of-concept distribution* $D_c^{\text{out}}$ as those from tokens labeled concept-negative. Formally, let $Z$ denote the set of tokens across samples and $S_c = \{ s_c(z) : z \in Z \}$ the corresponding collection of activation scores. If $Z_c^{\text{in}} \subseteq Z$ are the tokens containing $c$ and $Z_c^{\text{out}} = Z \setminus Z_c^{\text{in}}$, then

$$D_c^{\text{in}} = \{ s_c(z) : z \in Z_c^{\text{in}} \}, \qquad D_c^{\text{out}} = \{ s_c(z) : z \in Z_c^{\text{out}} \}.$$

Note that $Z_c^{\text{out}}$ excludes *all* tokens from samples containing $c$, even those not labeled with the concept, in order to prevent self-attention from leaking concept information into the out-of-concept distribution.

The *support* of a distribution is the set of values where it assigns nonzero probability, and the *tail* refers to its extreme regions with small probability mass. To quantify how much $D_c^{\text{in}}$ and $D_c^{\text{out}}$ overlap, we use the *overlap coefficient (OVL)*, defined as the shared probability mass between the two distributions:

$$\text{OVL}(D_c^{\text{in}}, D_c^{\text{out}}) = \int_{-\infty}^{\infty} \min\big(p^{\text{in}}(s),\, p^{\text{out}}(s)\big)\, ds,$$

where $p^{\text{in}}$ and $p^{\text{out}}$ are their densities. Large values of OVL indicate that most in-concept activations lie within the overlapping support $\text{supp}(D_c^{\text{in}}) \cap \text{supp}(D_c^{\text{out}})$ and are thus statistically indistinguishable from out-of-concept activations, whereas small values arise when only the high-activation tail of $D_c^{\text{in}}$ extends beyond $D_c^{\text{out}}$, yielding clearer separation.

**Concept Detection.** The goal of concept detection is to determine whether a sample $x$ contains a concept $c$ (Wu et al., 2025). Transformer models produce a collection of activation scores at the token level, but for detection we require a single score per sample. This necessitates an **aggregation operator** that interprets the set of token-level activations as a sample-level score.

Let $S_c(x) = \{s_{c,1}(x), \ldots, s_{c,n(x)}(x), s_{c,\text{cls}}(x)\}$ denote the set of activation scores for concept $c$ on input $x$, where $s_{c,i}(x)$ is the score for the $i$-th token and $s_{c,\text{cls}}(x)$ is the score for the [CLS] token. An aggregation operator is any function

$$G : \mathbb{R}^{n(x)+1} \to \mathbb{R}, \quad s_c^{\text{agg}}(x) = G(S_c(x)).$$

Given a calibrated threshold $\tau_c$, detection is performed by

$$\hat{y}_c(x) = \mathbf{1}\big[\, s_c^{\text{agg}}(x) \geq \tau_c \,\big].$$

Because prior work has shown that different concepts may emerge at different layers of a transformer (Saglam et al., 2025; Yu et al., 2024; Dalvi et al., 2022), we calibrate the layer separately for each concept to avoid enforcing a strict shared choice. This calibration is also performed independently for each aggregation strategy, ensuring that no operator is unfairly advantaged or disadvantaged due to layer-specific biases.

**Standard Aggregation Strategies.** Prior work has considered several choices of $G$, each operating on the same token-level activations (with the exception of [CLS], which uses separately trained concept vectors since sample-level and input token-level representations occupy different spaces):

- **[CLS]-only ($G_{\text{cls}}$):**
$$G_{\text{cls}}(S_c(x)) = s_{c,\text{cls}}(x).$$
Uses only the [CLS] token score. Since CLS tokens are trained to attend to all inputs, they are natural candidates for summarizing sample-level concepts, and this strategy has been widely adopted (Nejadgholi et al., 2022; Yu et al., 2024; Behrendt et al., 2025).

- **Mean pooling ($G_{\text{mean}}$):**
$$G_{\text{mean}}(S_c(x)) = \tfrac{1}{n(x)} \sum_{i=1}^{n(x)} s_{c,i}(x).$$
Averages over all tokens. This ensures that no part of the input is ignored and can capture distributed concept signals, a technique used in multiple studies (Benou & Riklin-Raviv, 2025; Suresh et al., 2025; Siddique et al., 2025).

- **Max pooling ($G_{\text{max}}$):**
$$G_{\text{max}}(S_c(x)) = \max\{s_{c,1}(x), \ldots, s_{c,n(x)}(x), s_{c,\text{cls}}(x)\}.$$
Takes the strongest activation across input tokens. This is effective for isolating the most distinct concept signals (Tillman & Mossing, 2025; Wu et al., 2025; Lim et al., 2025; Xie et al., 2025).

- **Last token ($G_{\text{last}}$):**
$$G_{\text{last}}(S_c(x)) = s_{c,n(x)}(x).$$
Uses the last input token activation. For autoregressive models, the final token often encodes sequence-level information, making it a plausible summary for concept detection (Chen et al., 2025; Tillman & Mossing, 2025; Tang & Yang, 2024).

- **Random token ($G_{\text{rand}}$):**
$$G_{\text{rand}}(S_c(x)) = s_{c,j}(x), \quad j \sim \text{Unif}\{1, \ldots, n(x)\}.$$
Selects an input token activation uniformly at random. While a weak baseline, self-attention mechanisms distribute information broadly, so even a randomly chosen token may retain meaningful concept cues.

These operators differ only in how they interpret activations; they do not alter how concept vectors are trained. Thresholds $\tau_c$ are determined using a validation set (e.g., from a fixed grid of percentiles), and detection at test time is performed by applying the same $G$ to the sample activations and comparing against $\tau_c$.

**SuperActivator Aggregation.** We develop an aggregation strategy that takes advantage of the SuperActivators mechanism we identified, using the highest-activation tokens in the global true-concept distribution as the basis for concept detection.

Formally, let
$$\mathcal{S}_{\text{val},c}^{+} = \big\{ s_{c,i}(x) \mid x \in \mathcal{X}_{\text{val},c}^{+}, \ i \in \{1, \ldots, n(x)\} \big\}$$
be the set of all token-level activations for $c$ from validation samples where $c$ is present. For a chosen percentile $N$ (selected from a fixed grid), we define the *SuperActivator threshold* as
$$\tau_c^{\text{super}} = Q_{1-N}\big(\mathcal{S}_{\text{val},c}^{+}\big),$$
so that only the top $N\%$ of in-concept activations exceed $\tau_c^{\text{super}}$. Unlike traditional max pooling approaches, which calibrate thresholds based on the single maximum activation per sample, our approach looks at the highest activations generally in the in-concept distribution, allowing us to consider multiple high-fidelity token activations per sample where calibrating.

At test time, we aggregate using a max operator,
$$G_{\text{super}}(S_c(x)) = \max S_c(x),$$
and predict presence if this maximum exceeds the calibrated SuperActivator threshold:
$$\hat{y}_c^{\text{super}}(x) = \mathbf{1}\big[ G_{\text{super}}(S_c(x)) \geq \tau_c^{\text{super}} \big].$$

$N$ is calibrated per concept on the validation set to maximize detection $F_1$. Beyond providing thresholds for reporting overall detection scores, this calibration also allows us to analyze how varying the sparsity level of the SuperActivators mechanism impacts performance.

# E    COMPREHENSIVE DETECTION RESULTS

The following tables show the average $F_1$ detection scores (weighted across concepts) for all models, sample type (SuperActivators vs CLS), and concept extracton method (mean prototype, linear separator, K-Means, linear separator on K-Means clusters) across datasets. Table 3 provides random and constant-predictor detection performances for all dataset–model combinations for reference. In each table, the top-performing concept detection method for each dataset is in bold and the second best-performing is underlined.

On the image datasets (i.e., *CLEVR*, *MS-Coco*, *OpenSurfaces*, and *Pascal*), our SuperActivator method consistently outperforms all other concept detection methods, except for a couple instances in the very simple *CLEVR* dataset, where prompting achieves the highest performance by a small margin. Though sometimes the CLS-based achieves near-equivalent performance, zero-shot prompting is most consistently the next best detection method. For the text datasets, (i.e., *Sarcasm*, *Augmented iSarcasm*, and *Augmented GoEmotions*), our SuperActivator also achieves consistently high detection performance across configurations. However, particularly for the *Augmented iSarcasm* dataset, CLS-based methods are able to outperform our SuperActivator, though usually by a very small amount that falls within the margin of error.

Overall, these results confirm that across image and text modalities, model families, and concept types, SuperActivator tokens provide a highly reliable signal of concept presence.

Table 3: Baseline concept detection $F_1$ scores: Constant Positive and Random Predictors.

| Dataset | Model | Concept Detection Baselines | |
| --- | --- | --- | --- |
| | | Constant Positive | Random |
| CLEVR | Llama | 0.502 ± 0.077 | 0.414 ± 0.102 |
| | CLIP | 0.502 ± 0.077 | 0.397 ± 0.101 |
| COCO | CLIP | 0.317 ± 0.029 | 0.262 ± 0.039 |
| | Llama | 0.316 ± 0.036 | 0.262 ± 0.048 |
| OpenSurfaces | Llama | 0.341 ± 0.035 | 0.282 ± 0.046 |
| | CLIP | 0.341 ± 0.035 | 0.285 ± 0.045 |
| Pascal | Llama | 0.380 ± 0.032 | 0.310 ± 0.041 |
| | CLIP | 0.380 ± 0.032 | 0.308 ± 0.041 |
| Sarcasm | Llama | 0.658 ± 0.052 | 0.519 ± 0.070 |
| | Gemma | 0.658 ± 0.052 | 0.514 ± 0.072 |
| | Qwen | 0.658 ± 0.052 | 0.496 ± 0.071 |
| iSarcasm | Llama | 0.676 ± 0.044 | 0.507 ± 0.062 |
| | Gemma | 0.676 ± 0.044 | 0.487 ± 0.062 |
| | Qwen | 0.676 ± 0.044 | 0.515 ± 0.062 |
| GoEmotions | Gemma | 0.102 ± 0.024 | 0.104 ± 0.035 |
| | Llama | 0.102 ± 0.024 | 0.095 ± 0.034 |
| | Qwen | 0.102 ± 0.024 | 0.098 ± 0.034 |

*Concept detection $F_1$ for the **CLEVR** dataset.*

| Model | Concept Type | Concept Detection Methods | | | | | |
|---|---|---|---|---|---|---|---|
| | | RandTok | LastTok | MeanTok | CLS | Prompt | SuperAct (Ours) |
| CLIP | Avg | 0.526 ± 0.028 | 0.542 ± 0.027 | 0.684 ± 0.020 | 0.957 ± 0.017 | **0.987 ± 0.009** | 0.986 ± 0.009 |
| | Linsep | 0.745 ± 0.009 | 0.706 ± 0.008 | 0.840 ± 0.009 | 0.963 ± 0.015 | 0.987 ± 0.009 | **0.991 ± 0.007** |
| | K-Means | 0.727 ± 0.013 | 0.878 ± 0.016 | 0.976 ± 0.013 | 0.959 ± 0.016 | 0.987 ± 0.009 | **0.991 ± 0.007** |
| | K-Linsep | 0.737 ± 0.017 | 0.848 ± 0.017 | 0.907 ± 0.019 | 0.965 ± 0.015 | **0.987 ± 0.009** | 0.950 ± 0.015 |
| Llama | Avg | 0.645 ± 0.018 | 0.591 ± 0.019 | 0.660 ± 0.018 | 0.955 ± 0.017 | 0.987 ± 0.009 | **0.998 ± 0.003** |
| | Linsep | 0.967 ± 0.090 | 0.879 ± 0.004 | 0.920 ± 0.004 | 0.961 ± 0.015 | 0.987 ± 0.009 | **0.997 ± 0.004** |
| | K-Means | 0.775 ± 0.089 | 0.946 ± 0.090 | 0.955 ± 0.013 | 0.928 ± 0.021 | **0.987 ± 0.009** | 0.959 ± 0.013 |
| | K-Linsep | 0.717 ± 0.024 | 0.910 ± 0.016 | 0.910 ± 0.015 | 0.962 ± 0.015 | 0.987 ± 0.009 | **0.989 ± 0.008** |

*Concept detection $F_1$ for the **COCO** dataset.*

| Model | Concept Type | Concept Detection Methods | | | | | |
|---|---|---|---|---|---|---|---|
| | | RandTok | LastTok | MeanTok | CLS | Prompt | SuperAct (Ours) |
| CLIP | Avg | 0.575 ± 0.012 | 0.503 ± 0.012 | 0.494 ± 0.013 | 0.685 ± 0.012 | 0.686 ± 0.050 | **0.721 ± 0.012** |
| | Linsep | 0.606 ± 0.011 | 0.687 ± 0.011 | 0.592 ± 0.011 | 0.702 ± 0.011 | 0.686 ± 0.050 | **0.787 ± 0.011** |
| | K-Means | 0.525 ± 0.013 | 0.517 ± 0.013 | 0.337 ± 0.012 | 0.583 ± 0.012 | 0.686 ± 0.050 | **0.694 ± 0.012** |
| | K-Linsep | 0.486 ± 0.012 | 0.523 ± 0.012 | 0.333 ± 0.011 | 0.571 ± 0.013 | 0.686 ± 0.050 | **0.696 ± 0.012** |
| Llama | Avg | 0.485 ± 0.011 | 0.457 ± 0.012 | 0.378 ± 0.012 | 0.534 ± 0.013 | 0.686 ± 0.050 | **0.746 ± 0.012** |
| | Linsep | 0.606 ± 0.011 | 0.680 ± 0.011 | 0.551 ± 0.011 | 0.566 ± 0.013 | 0.686 ± 0.050 | **0.829 ± 0.010** |
| | K-Means | 0.510 ± 0.012 | 0.491 ± 0.012 | 0.373 ± 0.011 | 0.447 ± 0.013 | 0.686 ± 0.050 | **0.747 ± 0.011** |
| | K-Linsep | 0.493 ± 0.011 | 0.477 ± 0.012 | 0.363 ± 0.011 | 0.430 ± 0.013 | 0.686 ± 0.050 | **0.716 ± 0.011** |

*Concept detection $F_1$ for the **OpenSurfaces** dataset.*

| Model | Concept Type | Concept Detection Methods | | | | | |
|---|---|---|---|---|---|---|---|
| | | RandTok | LastTok | MeanTok | CLS | Prompt | SuperAct (Ours) |
| CLIP | Avg | 0.438 ± 0.014 | 0.419 ± 0.013 | 0.403 ± 0.014 | 0.484 ± 0.014 | 0.491 ± 0.063 | **0.538 ± 0.014** |
| | Linsep | 0.470 ± 0.014 | 0.470 ± 0.014 | 0.427 ± 0.014 | 0.492 ± 0.014 | 0.491 ± 0.063 | **0.551 ± 0.014** |
| | K-Means | 0.443 ± 0.015 | 0.441 ± 0.015 | 0.373 ± 0.013 | 0.444 ± 0.010 | 0.491 ± 0.063 | **0.544 ± 0.014** |
| | K-Linsep | 0.432 ± 0.013 | 0.454 ± 0.012 | 0.365 ± 0.011 | 0.443 ± 0.009 | 0.491 ± 0.063 | **0.543 ± 0.012** |
| Llama | Avg | 0.404 ± 0.012 | 0.375 ± 0.012 | 0.361 ± 0.012 | 0.446 ± 0.014 | 0.491 ± 0.063 | **0.534 ± 0.014** |
| | Linsep | 0.438 ± 0.014 | 0.410 ± 0.014 | 0.390 ± 0.014 | 0.456 ± 0.013 | 0.491 ± 0.063 | **0.558 ± 0.015** |
| | K-Means | 0.443 ± 0.010 | 0.431 ± 0.011 | 0.360 ± 0.010 | 0.423 ± 0.005 | 0.491 ± 0.063 | **0.545 ± 0.009** |
| | K-Linsep | 0.439 ± 0.010 | 0.416 ± 0.011 | 0.360 ± 0.010 | 0.409 ± 0.011 | 0.491 ± 0.063 | **0.545 ± 0.008** |

*Concept detection $F_1$ for the **Pascal** dataset.*

| Model | Concept Type | Concept Detection Methods | | | | | |
|---|---|---|---|---|---|---|---|
| | | RandTok | LastTok | MeanTok | CLS | Prompt | SuperAct (Ours) |
| CLIP | Avg | 0.612 ± 0.006 | 0.546 ± 0.006 | 0.594 ± 0.006 | 0.721 ± 0.006 | 0.680 ± 0.048 | **0.788 ± 0.006** |
| | Linsep | 0.723 ± 0.005 | 0.674 ± 0.005 | 0.678 ± 0.005 | 0.740 ± 0.006 | 0.680 ± 0.048 | **0.826 ± 0.005** |
| | K-Means | 0.533 ± 0.005 | 0.623 ± 0.002 | 0.490 ± 0.005 | 0.652 ± 0.003 | 0.680 ± 0.048 | **0.770 ± 0.001** |
| | K-Linsep | 0.574 ± 0.005 | 0.577 ± 0.004 | 0.466 ± 0.005 | 0.633 ± 0.004 | 0.680 ± 0.048 | **0.756 ± 0.002** |
| Llama | Avg | 0.536 ± 0.006 | 0.510 ± 0.006 | 0.502 ± 0.006 | 0.619 ± 0.007 | 0.680 ± 0.048 | **0.786 ± 0.006** |
| | Linsep | 0.659 ± 0.006 | 0.602 ± 0.006 | 0.590 ± 0.006 | 0.645 ± 0.006 | 0.680 ± 0.048 | **0.822 ± 0.005** |
| | K-Means | 0.507 ± 0.006 | 0.601 ± 0.006 | 0.481 ± 0.006 | 0.568 ± 0.007 | 0.680 ± 0.048 | **0.792 ± 0.005** |
| | K-Linsep | 0.499 ± 0.006 | 0.550 ± 0.006 | 0.443 ± 0.006 | 0.558 ± 0.007 | 0.680 ± 0.048 | **0.784 ± 0.006** |

*Concept detection $F_1$ for the **Sarcasm** dataset.*

| Model | Concept Type | Concept Detection Methods | | | | | |
|---|---|---|---|---|---|---|---|
| | | RandTok | LastTok | MeanTok | CLS | Prompt | SuperAct (Ours) |
| Llama | Avg | 0.659 ± 0.052 | 0.706 ± 0.051 | 0.659 ± 0.052 | 0.694 ± 0.060 | 0.679 ± 0.074 | **0.818 ± 0.051** |
| | Linsep | 0.659 ± 0.060 | 0.683 ± 0.048 | 0.659 ± 0.060 | 0.737 ± 0.055 | 0.679 ± 0.074 | **0.870 ± 0.039** |
| | K-Means | 0.659 ± 0.061 | 0.659 ± 0.061 | 0.659 ± 0.061 | 0.665 ± 0.053 | 0.679 ± 0.074 | **0.818 ± 0.049** |
| | K-Linsep | 0.659 ± 0.054 | 0.670 ± 0.050 | 0.659 ± 0.052 | 0.658 ± 0.053 | 0.679 ± 0.074 | **0.826 ± 0.048** |
| Qwen | Avg | 0.662 ± 0.055 | 0.659 ± 0.066 | 0.659 ± 0.066 | **0.687 ± 0.055** | 0.679 ± 0.074 | 0.679 ± 0.060 |
| | Linsep | 0.659 ± 0.055 | 0.662 ± 0.051 | 0.659 ± 0.055 | 0.750 ± 0.054 | 0.679 ± 0.074 | **0.857 ± 0.046** |
| | K-Means | 0.659 ± 0.054 | 0.659 ± 0.054 | 0.659 ± 0.054 | 0.640 ± 0.059 | 0.679 ± 0.074 | **0.717 ± 0.062** |
| | K-Linsep | 0.659 ± 0.054 | 0.716 ± 0.057 | 0.659 ± 0.054 | 0.675 ± 0.053 | 0.679 ± 0.074 | **0.769 ± 0.057** |
| Gemma | Avg | 0.659 ± 0.058 | 0.659 ± 0.058 | 0.659 ± 0.058 | 0.665 ± 0.059 | 0.679 ± 0.074 | **0.727 ± 0.056** |
| | Linsep | 0.659 ± 0.059 | 0.668 ± 0.051 | 0.670 ± 0.051 | 0.686 ± 0.057 | 0.679 ± 0.074 | **0.810 ± 0.051** |
| | K-Means | 0.659 ± 0.053 | 0.659 ± 0.053 | 0.659 ± 0.053 | 0.658 ± 0.053 | **0.679 ± 0.074** | 0.659 ± 0.052 |
| | K-Linsep | 0.659 ± 0.054 | **0.682 ± 0.054** | 0.659 ± 0.054 | 0.670 ± 0.053 | 0.679 ± 0.074 | 0.659 ± 0.052 |

*Concept detection $F_1$ for the **Augmented iSarcasm** dataset.*

| Model | Concept Type | Concept Detection Methods | | | | | |
|---|---|---|---|---|---|---|---|
| | | RandTok | LastTok | MeanTok | CLS | Prompt | SuperAct (Ours) |
| Llama | Avg | 0.677 ± 0.043 | 0.676 ± 0.043 | 0.676 ± 0.043 | **0.867 ± 0.038** | 0.789 ± 0.047 | 0.818 ± 0.043 |
| | Linsep | 0.885 ± 0.035 | 0.717 ± 0.029 | 0.791 ± 0.029 | 0.912 ± 0.031 | 0.789 ± 0.047 | **0.924 ± 0.029** |
| | K-Means | 0.737 ± 0.048 | 0.677 ± 0.055 | 0.677 ± 0.055 | **0.809 ± 0.041** | 0.789 ± 0.047 | 0.787 ± 0.044 |
| | K-Linsep | 0.811 ± 0.038 | 0.828 ± 0.040 | 0.708 ± 0.045 | 0.802 ± 0.041 | 0.789 ± 0.047 | **0.866 ± 0.038** |
| Qwen | Avg | 0.676 ± 0.041 | 0.679 ± 0.041 | 0.678 ± 0.041 | **0.890 ± 0.034** | 0.789 ± 0.047 | 0.757 ± 0.041 |
| | Linsep | 0.814 ± 0.041 | 0.711 ± 0.038 | 0.739 ± 0.041 | **0.917 ± 0.030** | 0.789 ± 0.047 | 0.895 ± 0.034 |
| | K-Means | 0.676 ± 0.076 | 0.676 ± 0.076 | 0.676 ± 0.076 | **0.856 ± 0.038** | 0.789 ± 0.047 | 0.788 ± 0.046 |
| | K-Linsep | 0.749 ± 0.044 | 0.676 ± 0.043 | 0.676 ± 0.043 | **0.878 ± 0.036** | 0.789 ± 0.047 | 0.832 ± 0.042 |
| Gemma | Avg | 0.735 ± 0.045 | 0.686 ± 0.039 | 0.702 ± 0.045 | **0.899 ± 0.032** | 0.789 ± 0.047 | 0.839 ± 0.038 |
| | Linsep | 0.853 ± 0.031 | 0.789 ± 0.035 | 0.789 ± 0.035 | **0.904 ± 0.033** | 0.789 ± 0.047 | 0.892 ± 0.034 |
| | K-Means | 0.676 ± 0.073 | 0.676 ± 0.073 | 0.676 ± 0.044 | **0.827 ± 0.040** | 0.789 ± 0.047 | 0.810 ± 0.045 |
| | K-Linsep | 0.676 ± 0.043 | 0.679 ± 0.046 | 0.754 ± 0.043 | **0.864 ± 0.038** | 0.789 ± 0.047 | 0.825 ± 0.044 |

*Concept detection $F_1$ for the **Augmented GoEmotions** dataset.*

| Model | Concept Type | Concept Detection Methods | | | | | |
|---|---|---|---|---|---|---|---|
| | | RandTok | LastTok | MeanTok | CLS | Prompt | SuperAct (Ours) |
| Llama | Avg | 0.293 ± 0.027 | 0.216 ± 0.027 | 0.216 ± 0.026 | 0.277 ± 0.028 | 0.252 ± 0.100 | **0.383 ± 0.028** |
| | Linsep | 0.372 ± 0.028 | 0.307 ± 0.027 | 0.193 ± 0.029 | 0.320 ± 0.029 | 0.252 ± 0.100 | **0.459 ± 0.029** |
| | K-Means | 0.305 ± 0.028 | 0.281 ± 0.029 | 0.117 ± 0.028 | 0.192 ± 0.022 | 0.252 ± 0.100 | **0.417 ± 0.028** |
| | K-Linsep | 0.426 ± 0.027 | 0.365 ± 0.027 | 0.327 ± 0.028 | 0.213 ± 0.022 | 0.252 ± 0.100 | **0.448 ± 0.028** |
| Qwen | Avg | 0.277 ± 0.026 | 0.214 ± 0.026 | 0.151 ± 0.026 | 0.347 ± 0.028 | 0.252 ± 0.100 | **0.431 ± 0.027** |
| | Linsep | 0.305 ± 0.028 | 0.248 ± 0.025 | 0.199 ± 0.026 | 0.357 ± 0.028 | 0.252 ± 0.100 | **0.458 ± 0.027** |
| | K-Means | 0.341 ± 0.028 | 0.284 ± 0.027 | 0.111 ± 0.026 | 0.192 ± 0.021 | 0.252 ± 0.100 | **0.451 ± 0.027** |
| | K-Linsep | 0.390 ± 0.026 | 0.373 ± 0.027 | 0.365 ± 0.026 | 0.191 ± 0.022 | 0.252 ± 0.100 | **0.453 ± 0.028** |
| Gemma | Avg | 0.336 ± 0.024 | 0.313 ± 0.023 | 0.151 ± 0.022 | 0.366 ± 0.029 | 0.252 ± 0.100 | **0.394 ± 0.026** |
| | Linsep | 0.352 ± 0.026 | 0.301 ± 0.026 | 0.190 ± 0.027 | 0.361 ± 0.029 | 0.252 ± 0.100 | **0.420 ± 0.028** |
| | K-Means | 0.294 ± 0.028 | 0.213 ± 0.025 | 0.132 ± 0.025 | 0.218 ± 0.020 | 0.252 ± 0.100 | **0.422 ± 0.026** |
| | K-Linsep | 0.339 ± 0.028 | 0.315 ± 0.024 | 0.360 ± 0.025 | 0.205 ± 0.019 | 0.252 ± 0.100 | **0.414 ± 0.028** |

# F ABLATION: HOW DOES CONCEPT DETECTION PERFORMANCE VARY WITH DEPTH?

In this section, we investigate how average concept detection performance evolves throughout model depth. Figures 15 and 16 visualize heatmaps of the average detection $F_1$ scores as a function of transformer layer depth for image and text datasets, respectively. Each heatmap reports the mean $F_1$ score across all datasets for each model, concept type, and detection scheme, computed over a grid of model depths. These heatmaps help illustrate how concept signals emerge and strengthen at different stages within the network.

In the vision domain, detection performance generally increases with depth, plateauing around the middle layers and declining slightly at the final layer. This behavior aligns with findings from prior work (Saglam et al., 2025; Yu et al., 2024; Dalvi et al., 2022), which report that mid-level and late-level layers often capture the richest and most separable semantic information. A similar trend can be observed in text-based models, though with greater variability across datasets and concept types. These results highlight that the most reliable concept signals tend to emerge most clearly past intermediate layers, and that SuperActivator-based detection consistently distinguishes concept presence better than baselines.

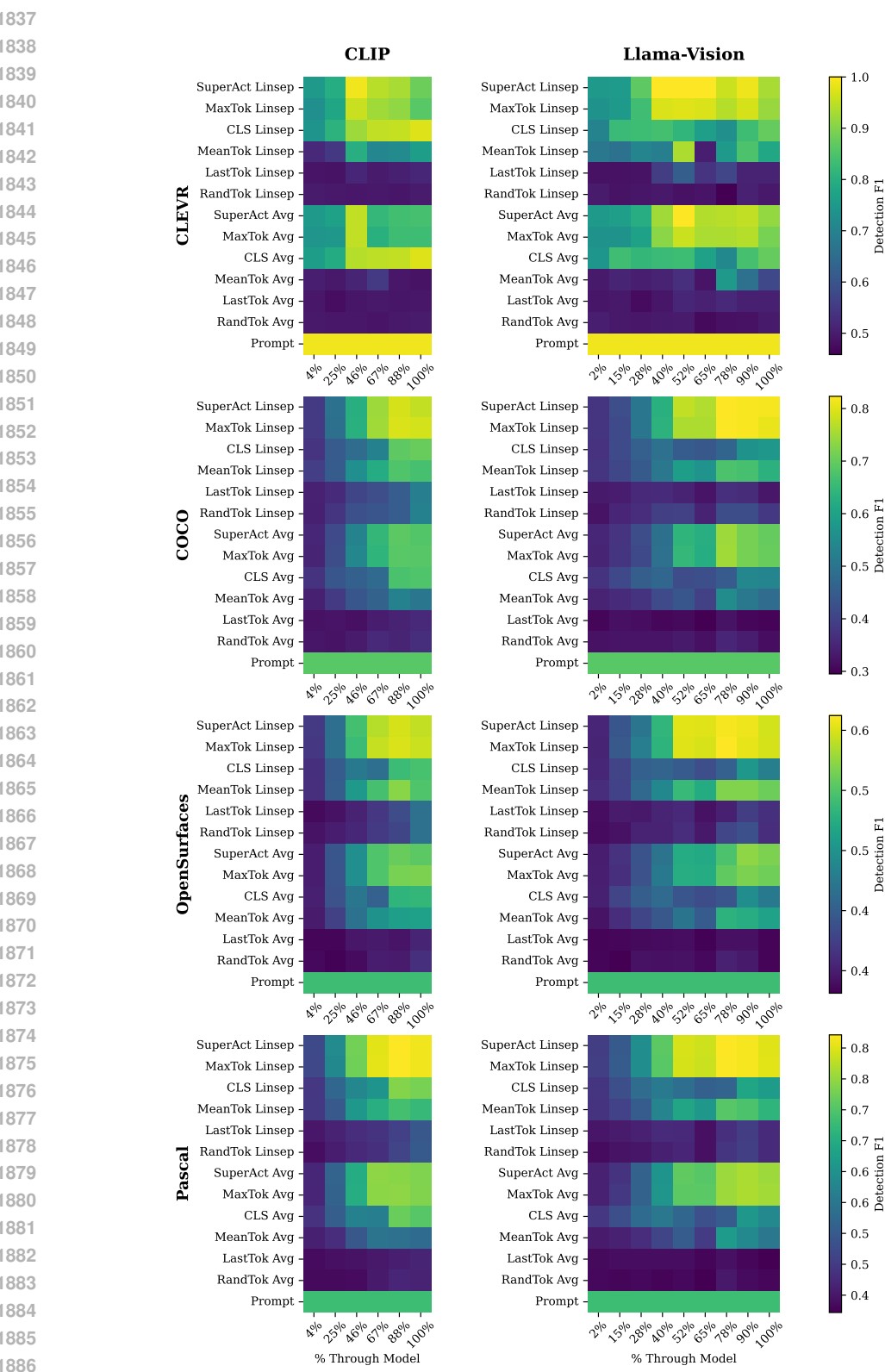

Figure 15: SuperActivator detection across image datasets.

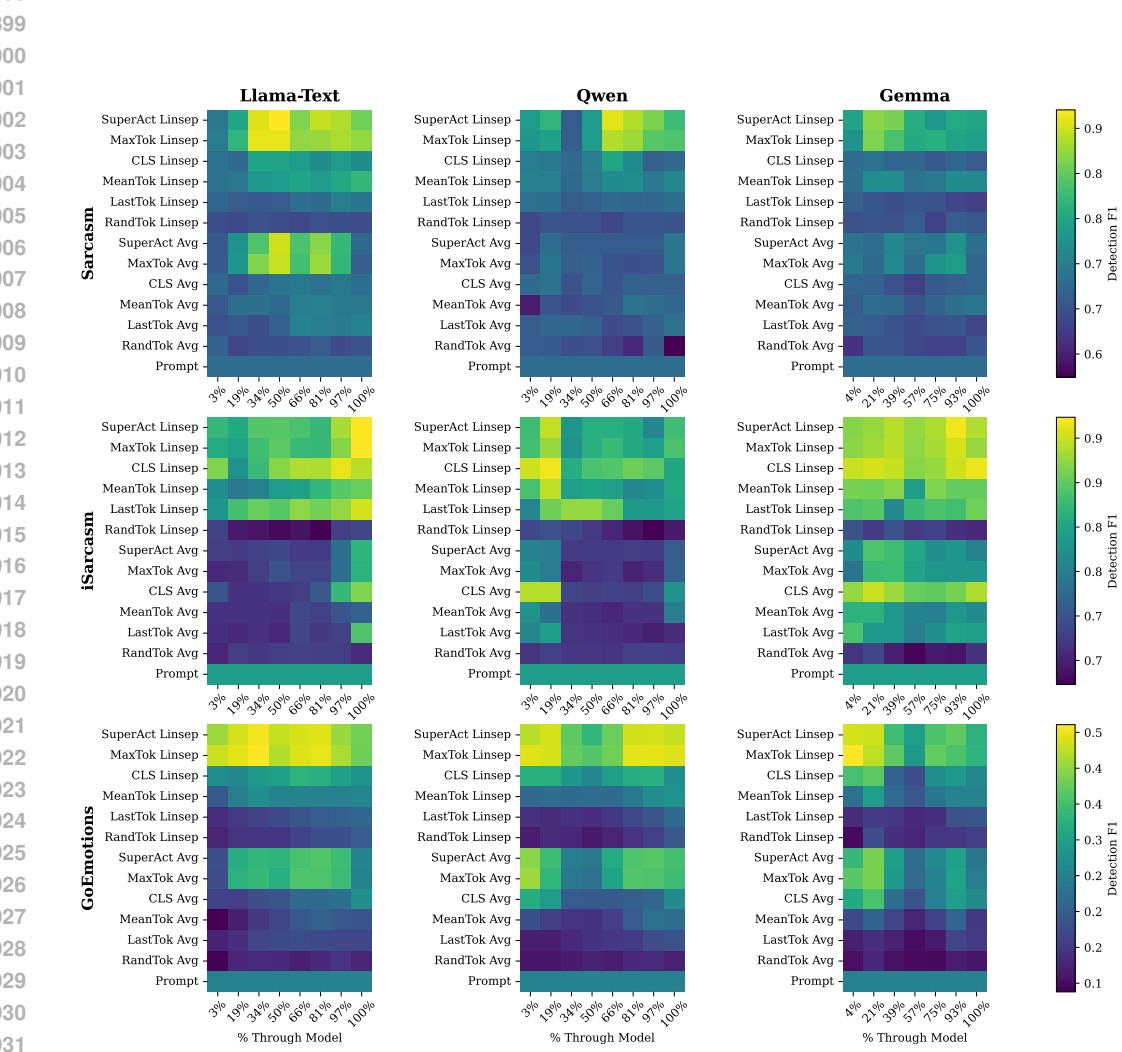

Figure 16: SuperActivator detection across text datasets.

# G    Ablation: How Does Sparsity Affect Average SuperActivator Detection Performance?

In this section, we evaluate SuperActivator-based concept detection performance across varying sparsity levels. The sparsity level $N$ corresponds to the $N$ in the SuperActivator definition—thresholds are calibrated using the top $N$ percent of in-concept token activations. Reported $F_1$ values represent the average of the per-concept detection $F_1$, each computed using the corresponding $N$, weighted by concept frequency and evaluated at each concept's best-performing layer on the validation set.

Across all model–dataset combinations, we observe that concepts generally achieve their strongest detection performance at low sparsity levels. This supports our broader finding that concept signals are highly concentrated: incorporating additional tokens beyond this sparse subset tends to degrade detection performance.

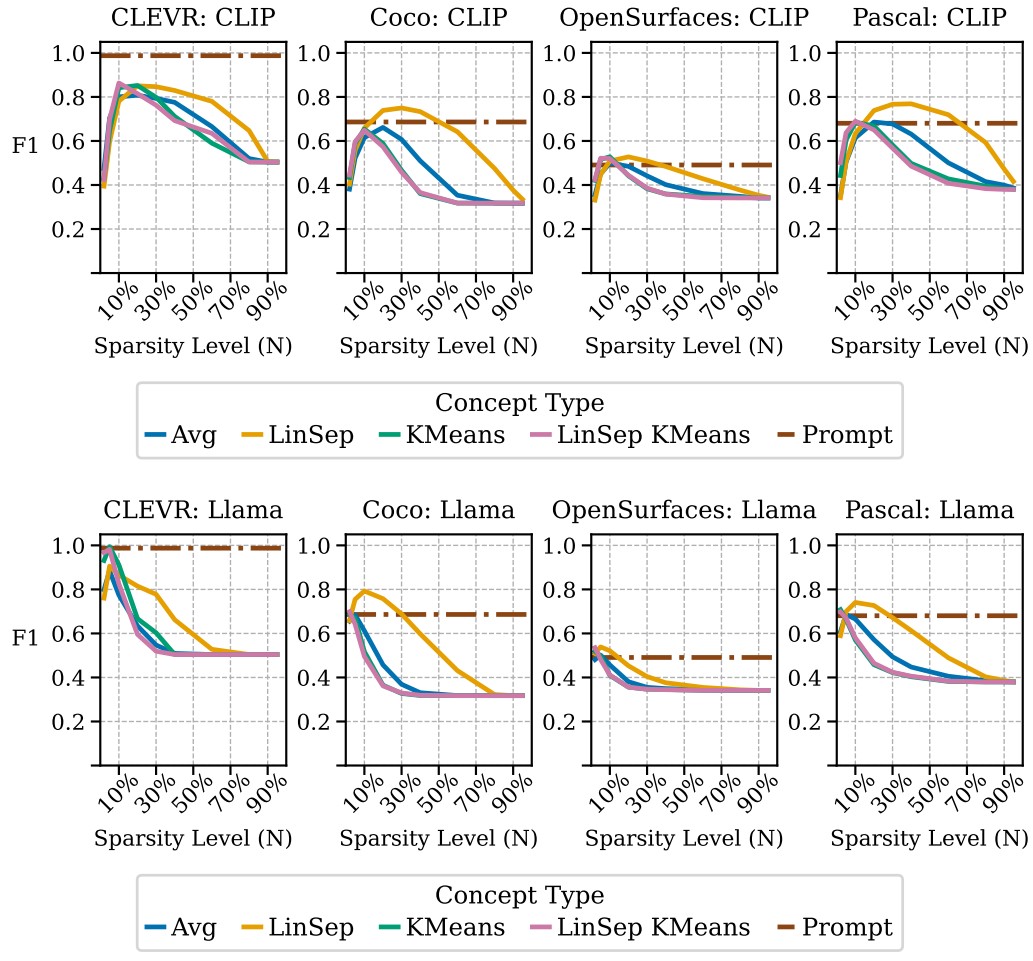

Figure 17: Image Domain – Detection $F_1$ over Sparsity Level $\delta$

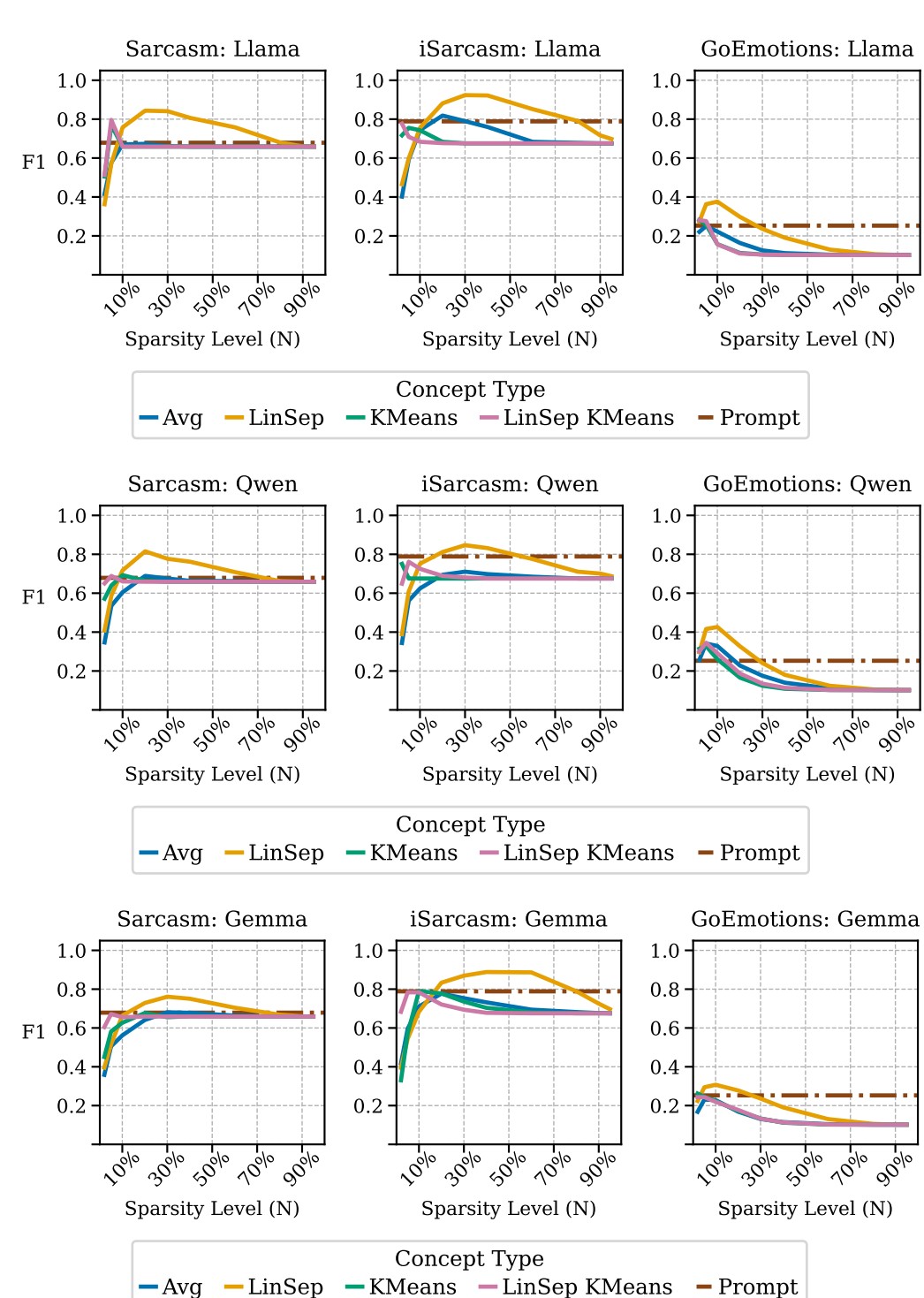

Figure 18: Text Domain – Detection $F_1$ over Sparsity Level $\delta$

# H    ABLATION: HOW DOES OPTIMAL SPARSITY FOR SUPERACTIVATOR DETECTION VARY ACROSS MODEL LAYERS?

Next, we analyze how the optimal sparsity levels, $N$s, for SuperActivator-based concept detection varies across layers in the model. Figures **??** and **??** visualize these results across layers for each model: at every layer, we report the frequency of concepts whose optimal detection occurs at each sparsity level $N$, with different colors demarcating the datasets the concepts came from.

Early in the model, the best concept detection via SuperActivators occurs at extremely high sparsity levels ($N \approx 0.02$–$0.05$) for most concepts. However, as shown in Appendix F, these early-layer activations are not yet reliable indicators of concept presence. As we move deeper through the transformer, the best-performing SuperActivators tend to occur at higher $N$, meaning that more tokens contribute to concept detection. Even so, the activations remain far from dense, typically involving fewer than half of the true in-concept tokens. Our main takeaway is that the concept signals are expressed most reliably by a small set of activations, no matter the depth that the concepts were extracted from.

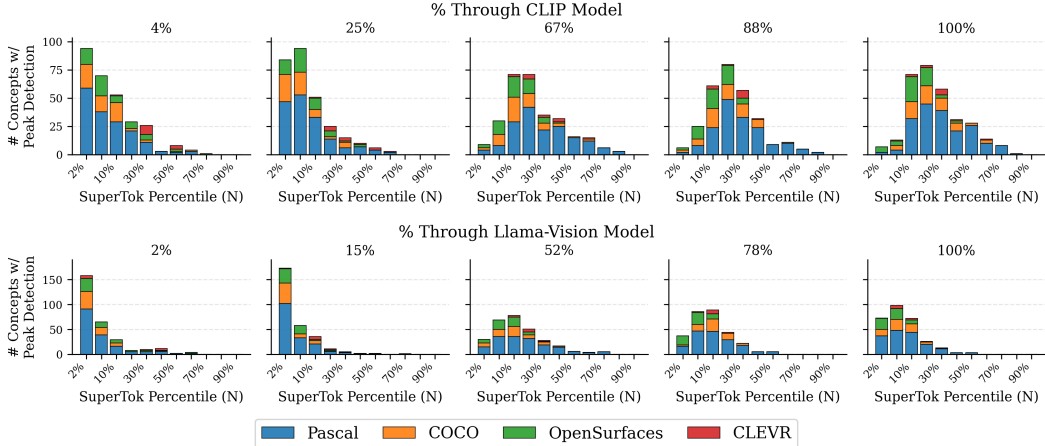

Figure 19: Image Domain – Optimal Sparsity over Layers

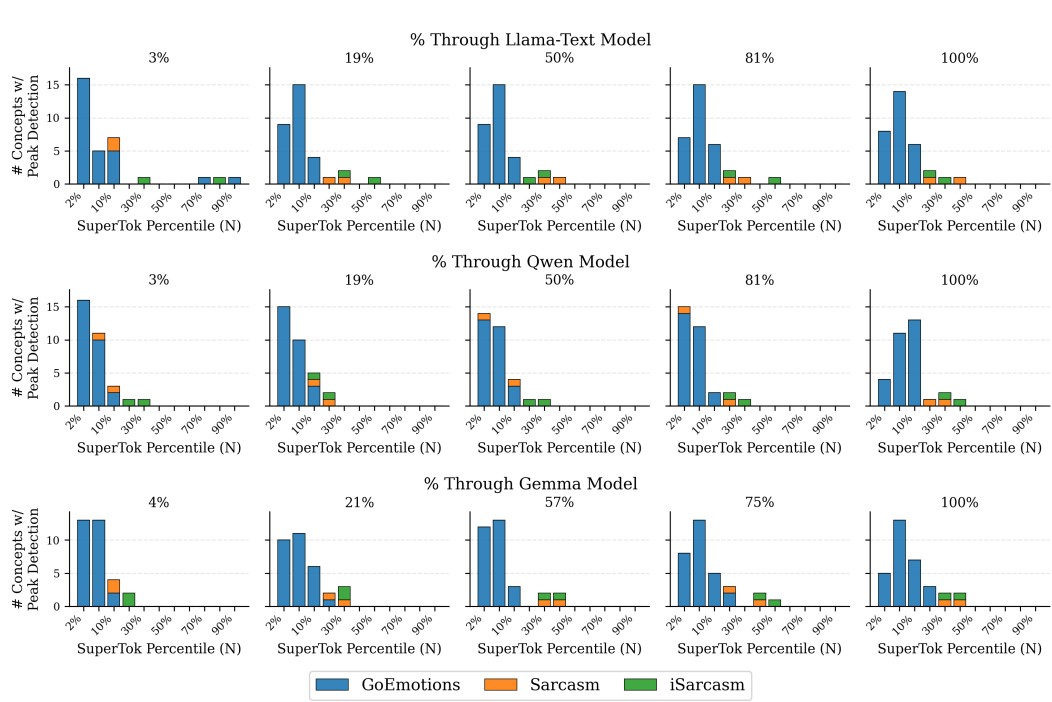

Figure 20: Text Domain – Optimal Sparsity over Layers

# I ABLATION: WHICH MODEL LAYERS YIELD THE MOST SEPARABLE CONCEPTS?

In this section, we seek to identify where in the model concepts are most separable, that is, at which layers concept vectors achieve their highest detection performance. For each dataset, we plot the frequency of concept vectors that achieve their best $F_1$ detection scores at each model layer. These trends are shown for the SuperActivator detection scheme as well as for [CLS]-, mean-, and last-token–based detection methods. All results in this analysis use linear separator concept vectors derived from the *LLaMA-3.2-11B-Vision-Instruct* model.

For image datasets with primarily high-level object concepts, such as *COCO* and *Pascal*, the best-performing concept vectors tend to appear in later layers. A similar but less pronounced pattern is observed in *OpenSurfaces*, which contains both high-level objects and lower-level texture concepts. In contrast, *CLEVR*—whose concepts include lower-level properties like color and slightly higher-level ones like shape—shows strong detection performance from both early and late layers, suggesting that different types of concepts emerge at different depths. For the text datasets *Sarcasm*, *iSarcasm*, and *GoEmotions*, a comparable pattern arises: the best-detecting concept vectors most often originate from later layers, though earlier layers also capture meaningful signals for certain concepts.

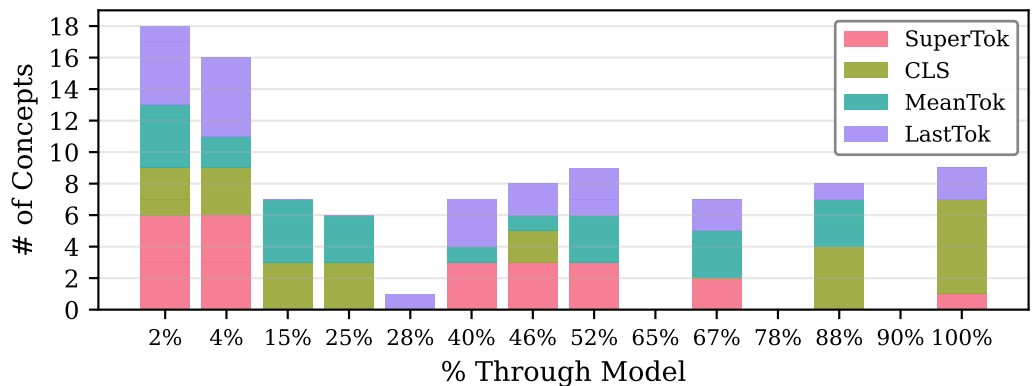

Figure 21: What Layers in Model Concepts are Best Detected in CLEVR via SuperActivator, CLS, Mean, and Last tokens. Concepts extracted from CLIP and Llama-Vision-Instruct models using average and linear separator concepts.

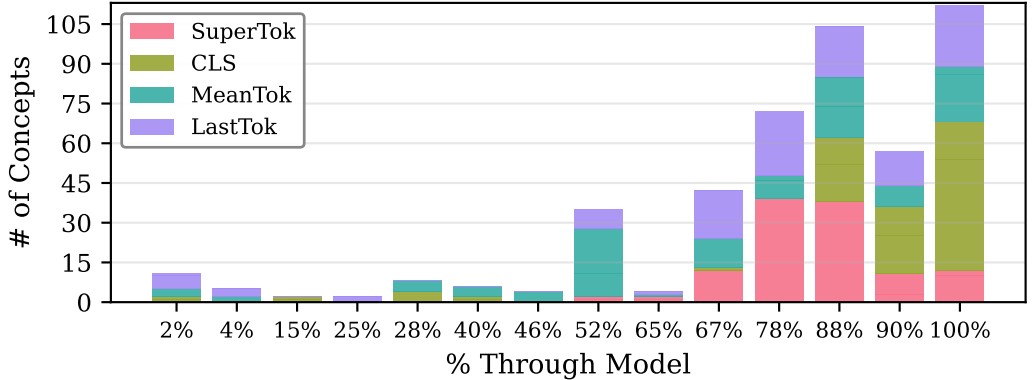

Figure 22: What Layers in Model Concepts are Best Detected in Coco via SuperActivator, CLS, Mean, and Last tokens. Concepts extracted from CLIP and Llama-Vision-Instruct models using average and linear separator concepts.

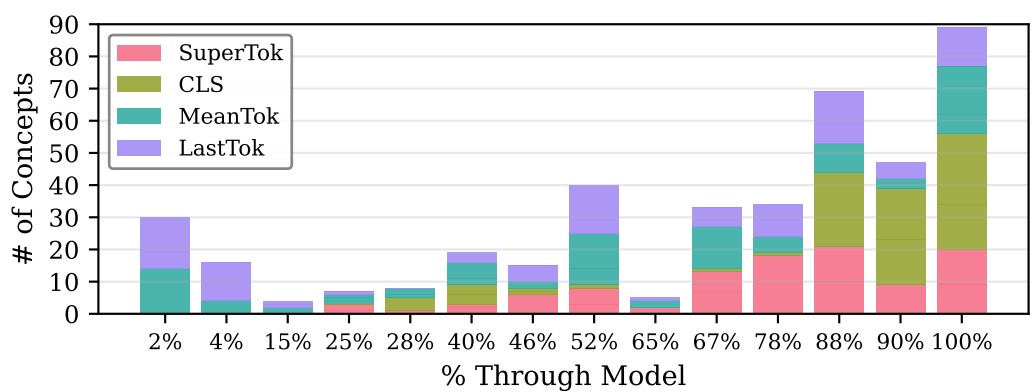

Figure 23: What Layers in Model Concepts are Best Detected in OpenSurfaces via SuperActivator, CLS, Mean, and Last tokens. Concepts extracted from CLIP and Llama-Vision-Instruct models using average and linear separator concepts.

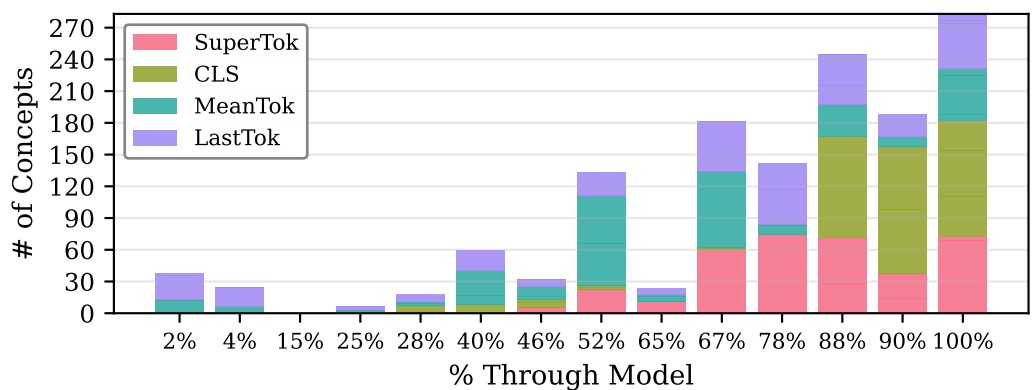

Figure 24: What Layers in Model Concepts are Best Detected in Pascal via SuperActivator, CLS, Mean, and Last tokens. Concepts extracted from CLIP and Llama-Vision-Instruct models using average and linear separator concepts.

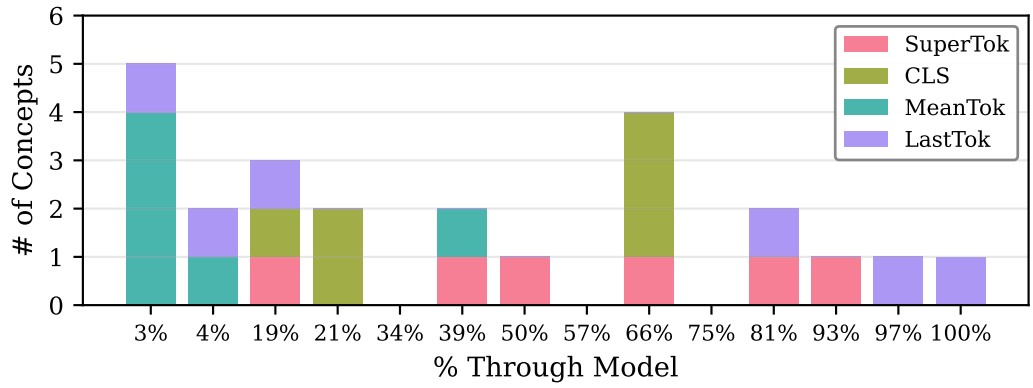

Figure 25: What Layers in Model Concepts are Best Detected in Sarcasm via SuperActivator, CLS, Mean, and Last tokens. Concepts extracted from Llama-Vision-Instruct, Qwen, and Gemma models using average and linear separator concepts.

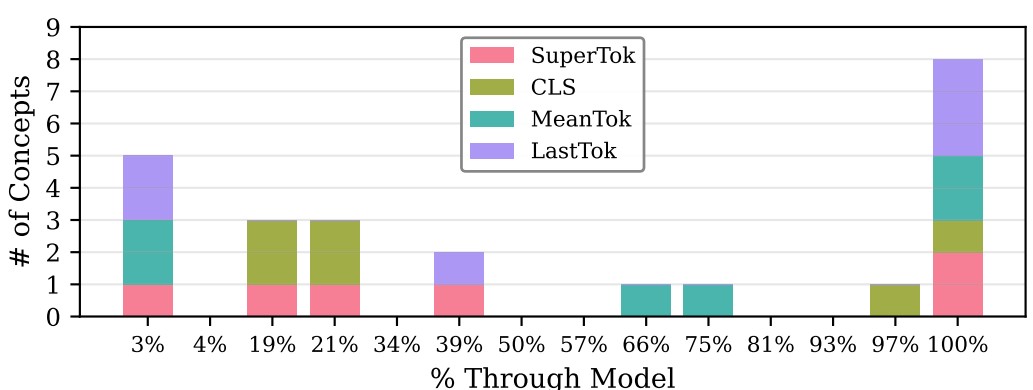

Figure 26: What Layers in Model Concepts are Best Detected in iSarcasm via SuperActivator, CLS, Mean, and Last tokens. Concepts extracted from Llama-Vision-Instruct, Qwen, and Gemma models using average and linear separator concepts.

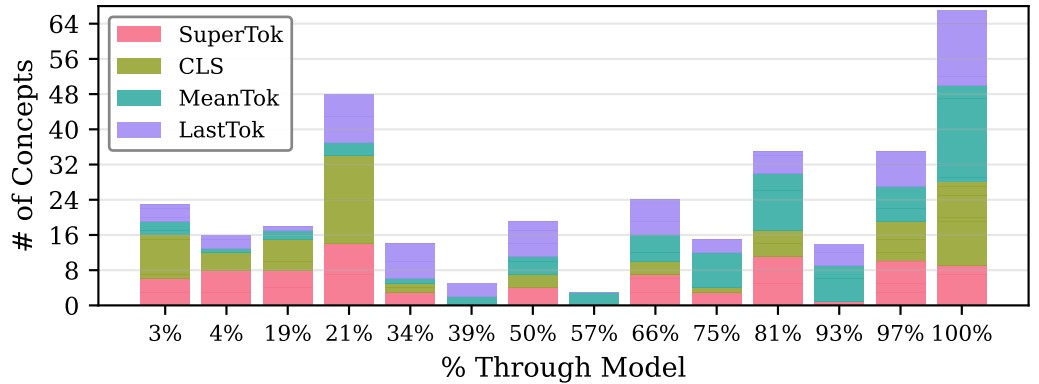

Figure 27: What Layers in Model Concepts are Best Detected in GoEmotions via SuperActivator, CLS, Mean, and Last tokens. Concepts extracted from Llama-Vision-Instruct, Qwen, and Gemma models using average and linear separator concepts.

## J   ABLATION: HOW MANY SUPERACTIVATORS DO MOST SAMPLES HAVE?

Figure 28 shows cumulative distribution functions for the *LLaMA-3.2-11B-Vision-Instruct* linear-separator concepts, using each concept's validation-selected model layer and optimal sparsity level $\delta$ on the test set. For each in-concept sample, we plot the ratio of SuperActivators in the sample to the number of in-concept tokens, which normalizes for varying concept-span lengths and allows SuperActivators to appear anywhere in the sequence.

In *COCO*, *OpenSurfaces*, *Pascal*, and *iSarcasm*, more than half of in-concept samples have a ratio below 0.2—that is, fewer than one SuperActivator for every five in-concept tokens. For *CLEVR*, *Sarcasm*, and *iSarcasm*, the ratios are roughly twice as high, but still represent only a minority of the in-concept tokens present in each sample. Overall, these plots indicate that most in-concept samples only have a small set of reliable concept signals, relative to the amount of in-concept tokens.

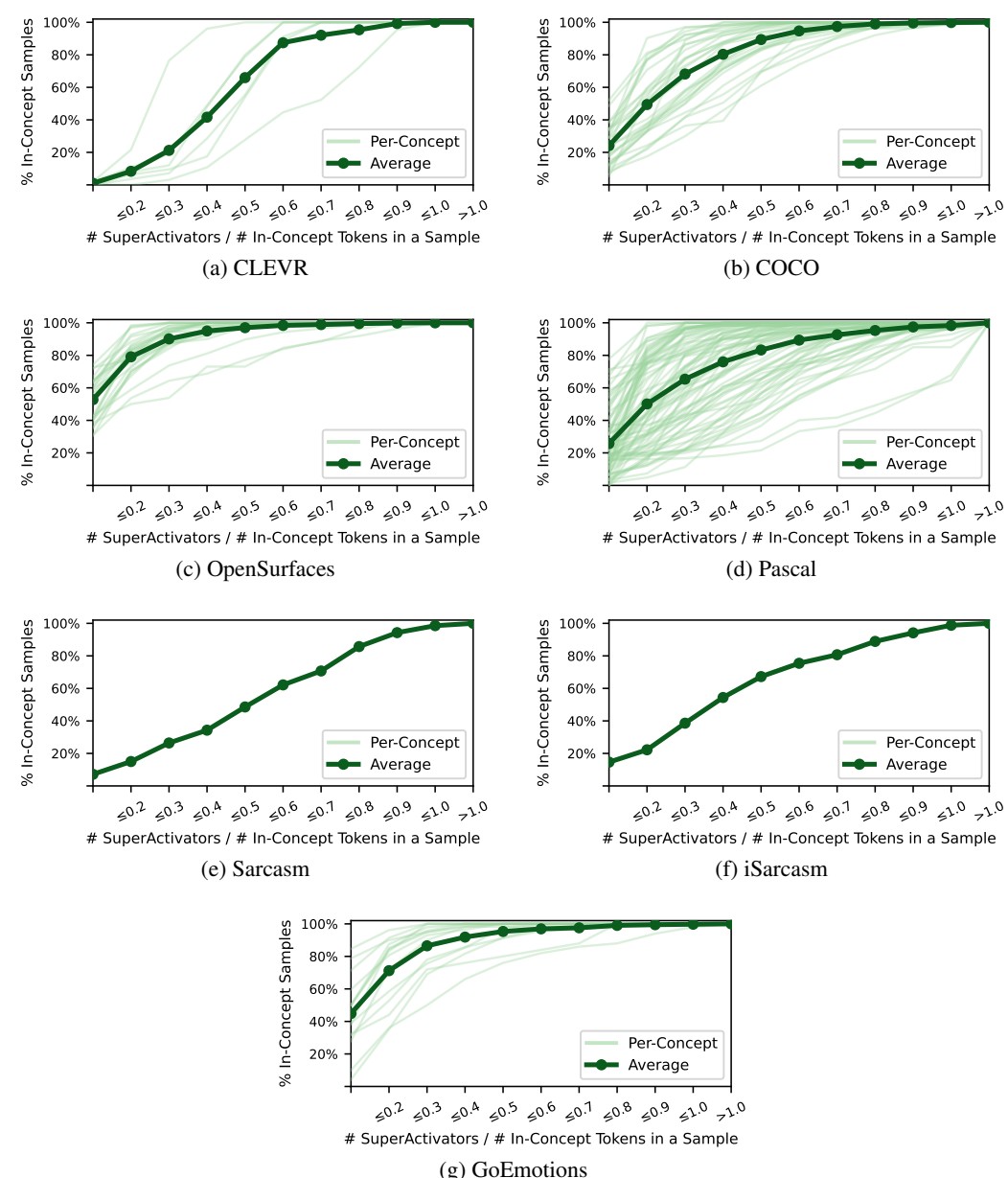

Figure 28: Cumulative distribution functions showing, for each concept and on average across a dataset, the ratio of SuperActivators to in-concept tokens in each test sample.

## K    ABLATION: DO SUPERACTIVATORS HAVE A POSITIONAL DEPENDENCE?

To check whether the SuperActivator Mechanism is driven by positional dependencies rather than genuine concept sensitivity, we plot the distribution of SuperActivators across image (Figure 29) and text (Figure 30) test splits. For each dataset, we use *Llama-3.2-11B-Vision-Instruct* linear separator concept SuperActivators, defined at the concept-specific model depth and sparsity level $\delta$ that yield the best detection performance on the validation set. The left panels show absolute SuperActivator token positions, while the right panels show relative positions normalized to the length of each sample.

In general, we observe no significant evidence of positional bias. The SuperActivators are not uniformly distributed, but there is no particular index or position where SuperActivators are much more common.

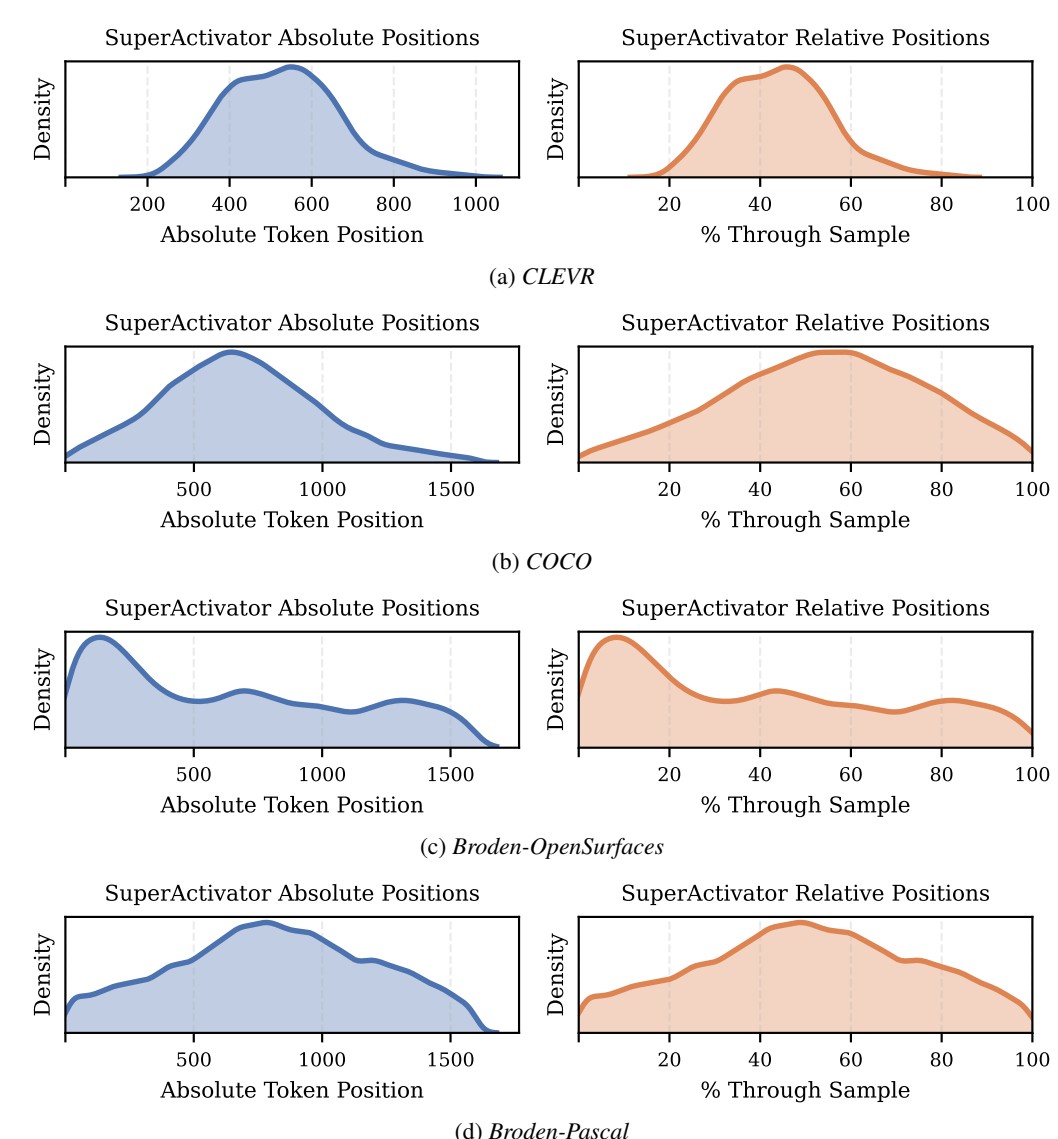

(a) *CLEVR*

(b) *COCO*

(c) *Broden-OpenSurfaces*

(d) *Broden-Pascal*

Figure 29: Image Domain – SuperActivator Position Distribution

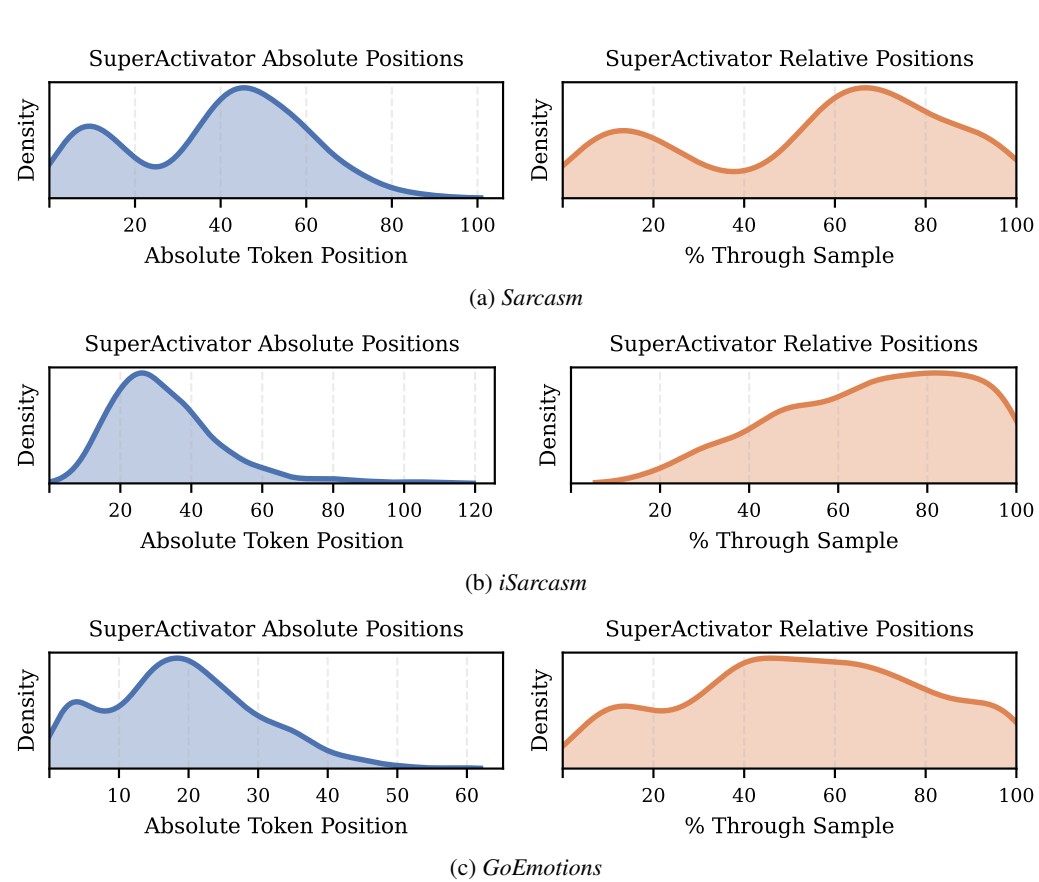

(a) *Sarcasm*

(b) *iSarcasm*

(c) *GoEmotions*

Figure 30: Text Domain – SuperActivator Position Distribution

## L  TAIL-FOCUSED SUPERACTIVATOR-BASED CONCEPT DETECTION

Section 4.2 showed that the most reliable indicators of concept presence consistently lie in the extreme high tail of the in-concept activation distribution. Building on this observation, we evaluate a simplified and practical variant of our SuperActivator detection method that operationalizes sparsity directly and requires only sample-level labels.

Leveraging the experimental results of our sparsity ablation in Appendix G, which found that a fixed sparsity of 10% performs well across datasets for *Llama-3.2-11B-Vision-Instruct* linear-separator concepts, we adopt this value for all datasets. Using the validation set, we estimate that this corresponds to selecting approximately 0.75% of all tokens per sample in image datasets and approximately 2% of tokens per sample in text datasets.

Then, for each sample, we retain only the top 0.75% of the highest image token activations (roughly 12 per sample) and 2% of the highest text token activations (roughly 2 per sample). Using only sample-level labels, we then learn a single threshold per concept that best separates selected tokens from positive versus negative samples. Tokens above this threshold are treated as SuperActivators, and a sample is predicted positive if it contains at least one. This procedure requires no segmentation masks and avoids any tuning of N.

The results for this method, which we denote as *N@Tail* are presented in Table 4, alongside the original baseline detection results and the previous SuperActivator-based results that employed N tuning. In this table we focus specifically on *Llama-3.2-11B-Vision-Instruct* linear separator concepts across all datasets. In terms of performance, the *N@Tail* variant achieves results that closely match the fully tuned version and still decisively outperforms all other baseline detection methods across datasets. This demonstrates that the practical value of the SuperActivator mechanism does not rely on extensive tuning; simply isolating the extreme tail of activations and learning a single weakly supervised threshold already captures most of the benefit.

Table 4: Weakly Supervised Tail-Only SuperActivator Detection Nearly Matches Tuned Performance

| | Concept Detection Methods | | | | | | |
| | RandTok | LastTok | MeanTok | CLS | Prompt | SuperAct (N tuned) | **SuperAct (N@tail)** |
|---|---|---|---|---|---|---|---|
| CLEVR | 0.967 ± 0.090 | 0.879 ± 0.004 | 0.920 ± 0.004 | 0.961 ± 0.015 | 0.987 ± 0.009 | **0.997 ± 0.004** | 0.995 ± 0.005 |
| COCO | 0.606 ± 0.011 | 0.680 ± 0.011 | 0.551 ± 0.011 | 0.566 ± 0.013 | 0.686 ± 0.050 | **0.829 ± 0.010** | 0.751 ± 0.069 |
| Surfaces | 0.438 ± 0.014 | 0.410 ± 0.014 | 0.390 ± 0.014 | 0.456 ± 0.013 | 0.491 ± 0.063 | **0.507 ± 0.079** | 0.495 ± 0.077 |
| Pascal | 0.659 ± 0.006 | 0.601 ± 0.006 | 0.594 ± 0.006 | 0.648 ± 0.006 | 0.680 ± 0.048 | **0.822 ± 0.005** | 0.735 ± 0.058 |
| Sarcasm | 0.659 ± 0.060 | 0.683 ± 0.048 | 0.659 ± 0.060 | 0.737 ± 0.055 | 0.679 ± 0.074 | **0.870 ± 0.039** | 0.869 ± 0.039 |
| iSarcasm | 0.885 ± 0.035 | 0.717 ± 0.029 | 0.791 ± 0.029 | 0.912 ± 0.031 | 0.789 ± 0.047 | **0.924 ± 0.029** | 0.918 ± 0.030 |
| GoEmot | 0.372 ± 0.028 | 0.307 ± 0.027 | 0.193 ± 0.029 | 0.320 ± 0.029 | 0.252 ± 0.100 | **0.459 ± 0.029** | 0.446 ± 0.102 |

# M  CONCEPT ATTRIBUTION

## M.1  ATTRIBUTION METHODS

This section provides a brief overview of several attribution methods in which the objective is defined either by a global concept vector $v_c$ or by the average embedding of local SuperActivators.

- **LIME (Local Interpretable Model-agnostic Explanations)** (Ribeiro et al., 2016) explains an individual prediction by approximating the complex model with a simpler, interpretable model (e.g., a linear model) in the local vicinity of the prediction. It achieves this by generating a new dataset of perturbed samples around the instance being explained and learning the simpler model on this new dataset, weighted by proximity to the original instance.

- **SHAP (SHapley Additive exPlanations)** (Lundberg & Lee, 2017) assigns an importance value to each feature for a particular prediction. Based on cooperative game theory, this value represents the feature's marginal contribution to the model's output, ensuring the sum of all values explains the difference between the model's prediction and a baseline.

- **RISE (Randomized Input Sampling for Explanation)** (Petsiuk et al., 2018) generates a visual explanation by probing the model with numerous randomly masked versions of an input image. The final importance map is a weighted average of these random masks, where weights are determined by the model's output confidence for each corresponding masked image.

- **SHAP IQ (SHAP Interaction-aware exPlanations for Quantifying feature importance)** (Fel et al., 2023) extends the SHAP framework to quantify the effects of feature interactions. Beyond calculating the main effect of each feature, it also computes interaction indices to provide a more complete picture of how combinations of features jointly influence a prediction.

- **IntGrad (Integrated Gradients)** (Sundararajan et al., 2017) calculates the importance of each input feature by integrating the gradients of the model's output with respect to the feature's inputs. This integration is performed along a straight-line path from a baseline input (e.g., a black image) to the actual input, satisfying key axioms like sensitivity.

- **Grad-CAM (Gradient-weighted Class Activation Mapping)** (Selvaraju et al., 2017) produces a coarse localization map for CNNs by using the gradients of the target class score with respect to the feature maps of the final convolutional layer. These gradients are used to compute a weighted combination of the activation maps, highlighting important image regions.

- **FullGrad** (Srinivas & Fleuret, 2019) enhances gradient-based explanations by aggregating gradient information from all layers of a neural network. It combines the input gradients with bias gradients from all intermediate feature maps to capture more comprehensive feature representations, resulting in more detailed saliency maps.

- **CALM (Class Activation Latent Mapping)** (Mahajan et al., 2021) improves on Class Activation Mapping (CAM) by introducing a probabilistic latent variable that directly represents the location of the most important visual cue for a model's prediction. Trained with the Expectation-Maximization (EM) algorithm, the method outputs a probability map showing the likelihood that each pixel is the critical cue for the decision.

- **MFABA (More Faithful and Accelerated Boundary-based Attribution)** (Zhu et al., 2024) is a boundary-based attribution method that constructs a path from an input toward the decision boundary. Along this path, it uses a second-order Taylor expansion of the loss function to better approximate how the model's output or loss changes. The resulting attribution scores reflect how much each feature contributes to pushing the input toward or away from the boundary.

## M.2  ADDITIONAL RESULTS FOR CONCEPT ATTRIBUTION

This section presents the full results for concept attribution across all experimental configurations, which were summarized in Table 2 in the main text. These detailed tables are provided to demonstrate that our main findings are consistent across all individual concepts and experimental settings.

As these results confirm, using the average embedding of SuperActivators as the explanation objective consistently leads to better performance than using the concept vector directly. Moreover, linear separators generally outperform simple clustering for concept representation.

We present our results across fourteen tables, evaluating two concept representations (clustering-based vs. linear separators) and two attribution objectives (global concept vector vs. average local SuperActivatorspatch embedding). Each table reports the average $F_1$ score across all concepts, weighted by concept frequency in the test set (Appendix C.5). The tables are organized as follows:

- **Supervised Setting**: We provide results across seven tables. Four tables correspond to image tasks (Tables 5, 6, 7, and 8), and three correspond to text tasks (Tables 9, 10, and 11). The concept types for this setting are detailed in Appendix C.2.
- **Unsupervised Setting**: We provide results across seven tables. Four tables correspond to image tasks (Tables 12, 13, 14, and 15), and three correspond to text tasks (Tables 16, 17, and 18). Here, concepts are derived from k-means clusters, and for each concept, we evaluate the best-performing cluster out of 1000 candidates. The concept types are detailed in Appendix C.2.

Table 5: Average F1 for the CLEVR Dataset (Supervised).

| Attribution Method | Concept Type | CLIP | | Llama | |
|---|---|---|---|---|---|
| | | Concept | SuperActivators | Concept | SuperActivators |
| CosSim | Clustering | **0.60 ± 0.02** | **0.60 ± 0.01** | **0.78 ± 0.01** | 0.55 ± 0.03 |
| | LinSep | **0.65 ± 0.01** | 0.61 ± 0.03 | **0.85 ± 0.02** | 0.54 ± 0.01 |
| LIME | Clustering | 0.49 ± 0.02 | **0.55 ± 0.04** | 0.76 ± 0.03 | **0.81 ± 0.02** |
| | LinSep | 0.49 ± 0.00 | **0.68 ± 0.01** | 0.70 ± 0.01 | **0.85 ± 0.01** |
| SHAP | Clustering | 0.51 ± 0.01 | **0.53 ± 0.02** | 0.75 ± 0.02 | **0.80 ± 0.03** |
| | LinSep | 0.52 ± 0.03 | **0.58 ± 0.01** | 0.75 ± 0.01 | **0.80 ± 0.01** |
| RISE | Clustering | **0.53 ± 0.02** | **0.53 ± 0.03** | 0.55 ± 0.03 | **0.56 ± 0.02** |
| | LinSep | 0.58 ± 0.01 | **0.59 ± 0.02** | 0.60 ± 0.02 | **0.63 ± 0.01** |
| SHAP IQ | Clustering | 0.52 ± 0.04 | **0.53 ± 0.01** | 0.55 ± 0.01 | **0.58 ± 0.02** |
| | LinSep | **0.58 ± 0.02** | **0.58 ± 0.03** | 0.60 ± 0.03 | **0.61 ± 0.01** |
| IntGrad | Clustering | 0.46 ± 0.01 | **0.53 ± 0.03** | 0.77 ± 0.02 | **0.80 ± 0.02** |
| | LinSep | 0.49 ± 0.03 | **0.55 ± 0.01** | 0.72 ± 0.01 | **0.78 ± 0.03** |
| GradCAM | Clustering | 0.45 ± 0.02 | **0.48 ± 0.01** | 0.50 ± 0.03 | **0.52 ± 0.01** |
| | LinSep | **0.48 ± 0.01** | **0.48 ± 0.02** | 0.50 ± 0.02 | **0.52 ± 0.02** |
| FullGrad | Clustering | **0.46 ± 0.02** | **0.46 ± 0.03** | **0.47 ± 0.01** | 0.49 ± 0.02 |
| | LinSep | 0.50 ± 0.01 | **0.52 ± 0.02** | 0.51 ± 0.02 | **0.55 ± 0.01** |
| CALM | Clustering | 0.48 ± 0.03 | **0.52 ± 0.01** | 0.49 ± 0.03 | **0.53 ± 0.02** |
| | LinSep | 0.55 ± 0.02 | **0.56 ± 0.02** | **0.57 ± 0.01** | **0.57 ± 0.03** |
| MFABA | Clustering | 0.50 ± 0.01 | **0.51 ± 0.01** | 0.51 ± 0.02 | **0.53 ± 0.01** |
| | LinSep | **0.55 ± 0.03** | **0.55 ± 0.02** | 0.56 ± 0.01 | **0.58 ± 0.03** |

### M.3 QUALITATIVE EXAMPLE SHOWING SUPERACTIVATORS FOR IMPROVED CONCEPT ATTRIBUTION

Figure 6 further illustrates the advantage: attribution using SuperActivators for the concept *person* provides better coverage for the full target object while avoiding irrelevant regions such as tables, which the global vector incorrectly highlights.

Table 6: Average F1 for the COCO Dataset (Supervised).

| Attribution Method | Concept Type | CLIP | | Llama | |
|---|---|---|---|---|---|
| | | Concept | SuperActivators | Concept | SuperActivators |
| CosSim | Clustering | **0.43 ± 0.03** | 0.40 ± 0.02 | 0.36 ± 0.02 | **0.37 ± 0.01** |
| | LinSep | **0.52 ± 0.02** | 0.45 ± 0.00 | **0.46 ± 0.03** | 0.44 ± 0.02 |
| LIME | Clustering | 0.32 ± 0.01 | **0.38 ± 0.02** | 0.47 ± 0.01 | **0.51 ± 0.02** |
| | LinSep | 0.29 ± 0.02 | **0.40 ± 0.03** | 0.49 ± 0.02 | **0.50 ± 0.03** |
| SHAP | Clustering | 0.34 ± 0.03 | **0.38 ± 0.01** | 0.48 ± 0.03 | **0.51 ± 0.01** |
| | LinSep | 0.35 ± 0.01 | **0.37 ± 0.02** | 0.49 ± 0.02 | **0.55 ± 0.04** |
| RISE | Clustering | **0.34 ± 0.01** | **0.34 ± 0.02** | 0.36 ± 0.01 | **0.38 ± 0.01** |
| | LinSep | 0.35 ± 0.02 | **0.38 ± 0.03** | 0.35 ± 0.03 | **0.40 ± 0.02** |
| SHAP IQ | Clustering | 0.33 ± 0.03 | **0.35 ± 0.02** | 0.35 ± 0.02 | **0.36 ± 0.01** |
| | LinSep | 0.34 ± 0.01 | **0.37 ± 0.01** | 0.36 ± 0.01 | **0.38 ± 0.03** |
| IntGrad | Clustering | 0.30 ± 0.02 | **0.33 ± 0.02** | 0.42 ± 0.03 | **0.45 ± 0.01** |
| | LinSep | 0.28 ± 0.00 | **0.35 ± 0.04** | 0.43 ± 0.02 | **0.48 ± 0.01** |
| GradCAM | Clustering | **0.31 ± 0.03** | **0.31 ± 0.01** | 0.32 ± 0.02 | **0.35 ± 0.03** |
| | LinSep | 0.37 ± 0.01 | **0.38 ± 0.02** | **0.37 ± 0.01** | **0.37 ± 0.02** |
| FullGrad | Clustering | **0.33 ± 0.02** | 0.32 ± 0.01 | 0.35 ± 0.03 | **0.38 ± 0.01** |
| | LinSep | **0.43 ± 0.01** | **0.43 ± 0.00** | **0.39 ± 0.01** | **0.39 ± 0.03** |
| CALM | Clustering | **0.32 ± 0.02** | **0.32 ± 0.03** | 0.30 ± 0.01 | **0.29 ± 0.02** |
| | LinSep | **0.42 ± 0.01** | **0.42 ± 0.01** | 0.38 ± 0.02 | **0.41 ± 0.01** |
| MFABA | Clustering | 0.31 ± 0.04 | **0.37 ± 0.02** | 0.33 ± 0.03 | **0.34 ± 0.02** |
| | LinSep | 0.33 ± 0.01 | **0.39 ± 0.03** | 0.35 ± 0.02 | **0.39 ± 0.01** |

Table 7: Average F1 for the OpenSurfaces Dataset (Supervised).

| Attribution Method | Concept Type | CLIP | | Llama | |
|---|---|---|---|---|---|
| | | Concept | SuperActivators | Concept | SuperActivators |
| CosSim | Clustering | **0.22 ± 0.01** | 0.18 ± 0.04 | **0.19 ± 0.03** | 0.15 ± 0.02 |
| | LinSep | **0.28 ± 0.03** | 0.22 ± 0.02 | **0.23 ± 0.01** | 0.17 ± 0.01 |
| LIME | Clustering | 0.42 ± 0.03 | **0.50 ± 0.01** | 0.55 ± 0.03 | **0.62 ± 0.01** |
| | LinSep | 0.46 ± 0.01 | **0.50 ± 0.03** | 0.60 ± 0.01 | **0.68 ± 0.02** |
| SHAP | Clustering | 0.40 ± 0.02 | **0.42 ± 0.04** | 0.53 ± 0.02 | **0.57 ± 0.03** |
| | LinSep | 0.42 ± 0.02 | **0.44 ± 0.01** | 0.55 ± 0.03 | **0.56 ± 0.01** |
| RISE | Clustering | 0.40 ± 0.04 | **0.42 ± 0.01** | 0.51 ± 0.02 | **0.52 ± 0.03** |
| | LinSep | 0.43 ± 0.01 | **0.45 ± 0.02** | 0.53 ± 0.01 | **0.55 ± 0.02** |
| SHAP IQ | Clustering | 0.40 ± 0.02 | **0.43 ± 0.01** | 0.51 ± 0.03 | **0.53 ± 0.02** |
| | LinSep | 0.42 ± 0.03 | **0.45 ± 0.02** | **0.52 ± 0.01** | **0.52 ± 0.02** |
| IntGrad | Clustering | 0.43 ± 0.01 | **0.51 ± 0.02** | 0.46 ± 0.02 | **0.47 ± 0.03** |
| | LinSep | 0.44 ± 0.02 | **0.49 ± 0.02** | 0.56 ± 0.01 | **0.62 ± 0.02** |
| GradCAM | Clustering | 0.41 ± 0.02 | **0.43 ± 0.03** | 0.45 ± 0.01 | **0.46 ± 0.02** |
| | LinSep | 0.44 ± 0.01 | **0.46 ± 0.01** | 0.45 ± 0.03 | **0.51 ± 0.01** |
| FullGrad | Clustering | 0.38 ± 0.03 | **0.41 ± 0.02** | 0.40 ± 0.02 | **0.41 ± 0.01** |
| | LinSep | 0.42 ± 0.04 | **0.45 ± 0.01** | 0.43 ± 0.01 | **0.47 ± 0.02** |
| CALM | Clustering | 0.33 ± 0.01 | **0.35 ± 0.01** | 0.35 ± 0.02 | **0.37 ± 0.01** |
| | LinSep | 0.35 ± 0.02 | **0.38 ± 0.03** | 0.36 ± 0.01 | **0.41 ± 0.03** |
| MFABA | Clustering | 0.42 ± 0.02 | **0.44 ± 0.03** | **0.44 ± 0.01** | **0.44 ± 0.02** |
| | LinSep | 0.45 ± 0.01 | **0.48 ± 0.01** | 0.44 ± 0.02 | **0.47 ± 0.03** |

Table 8: Average F1 for the Pascal Dataset (Supervised).

| Attribution Method | Concept Type | CLIP | | Llama | |
|---|---|---|---|---|---|
| | | Concept | SuperActivators | Concept | SuperActivators |
| CosSim | Clustering | **0.42 ± 0.02** | 0.35 ± 0.01 | **0.40 ± 0.01** | 0.29 ± 0.04 |
| | LinSep | **0.54 ± 0.01** | 0.42 ± 0.03 | **0.46 ± 0.02** | 0.33 ± 0.03 |
| LIME | Clustering | 0.50 ± 0.02 | **0.52 ± 0.02** | 0.69 ± 0.02 | **0.71 ± 0.03** |
| | LinSep | 0.51 ± 0.03 | **0.55 ± 0.01** | 0.71 ± 0.03 | **0.72 ± 0.01** |
| SHAP | Clustering | 0.48 ± 0.01 | **0.52 ± 0.03** | 0.65 ± 0.01 | **0.70 ± 0.02** |
| | LinSep | 0.50 ± 0.00 | **0.52 ± 0.02** | 0.69 ± 0.02 | **0.72 ± 0.01** |
| RISE | Clustering | 0.50 ± 0.03 | **0.51 ± 0.01** | 0.52 ± 0.01 | **0.55 ± 0.03** |
| | LinSep | **0.54 ± 0.03** | **0.54 ± 0.02** | 0.55 ± 0.02 | **0.58 ± 0.01** |
| SHAP IQ | Clustering | 0.50 ± 0.01 | **0.51 ± 0.03** | 0.52 ± 0.01 | **0.55 ± 0.04** |
| | LinSep | 0.52 ± 0.02 | **0.53 ± 0.04** | 0.53 ± 0.03 | **0.54 ± 0.01** |
| IntGrad | Clustering | 0.48 ± 0.03 | **0.51 ± 0.01** | 0.69 ± 0.01 | **0.71 ± 0.02** |
| | LinSep | 0.49 ± 0.01 | **0.52 ± 0.03** | 0.67 ± 0.03 | **0.71 ± 0.01** |
| GradCAM | Clustering | 0.43 ± 0.04 | **0.45 ± 0.02** | 0.45 ± 0.02 | **0.45 ± 0.03** |
| | LinSep | 0.44 ± 0.03 | **0.47 ± 0.01** | 0.47 ± 0.02 | **0.50 ± 0.01** |
| FullGrad | Clustering | 0.41 ± 0.01 | **0.44 ± 0.03** | 0.40 ± 0.01 | **0.42 ± 0.03** |
| | LinSep | 0.44 ± 0.02 | **0.45 ± 0.01** | **0.44 ± 0.02** | 0.44 ± 0.02 |
| CALM | Clustering | **0.42 ± 0.03** | 0.42 ± 0.02 | 0.44 ± 0.03 | **0.45 ± 0.01** |
| | LinSep | 0.46 ± 0.01 | **0.48 ± 0.01** | 0.48 ± 0.02 | **0.52 ± 0.01** |
| MFABA | Clustering | 0.50 ± 0.02 | **0.52 ± 0.02** | 0.50 ± 0.03 | **0.51 ± 0.01** |
| | LinSep | 0.53 ± 0.02 | **0.55 ± 0.03** | 0.51 ± 0.01 | **0.52 ± 0.02** |

Table 9: Average F1 for the Sarcasm Dataset (Supervised).

| Attribution Method | Concept Type | Llama | | Qwen | | Gemma | |
|---|---|---|---|---|---|---|---|
| | | Concept | Super Activators | Concept | Super Activators | Concept | Super Activators |
| CosSim | Cluster | **0.39 ± 0.01** | 0.25 ± 0.03 | **0.38 ± 0.02** | 0.26 ± 0.03 | **0.42 ± 0.03** | 0.25 ± 0.02 |
| | LinSep | **0.63 ± 0.02** | 0.37 ± 0.01 | **0.58 ± 0.01** | 0.37 ± 0.02 | **0.57 ± 0.01** | 0.40 ± 0.03 |
| LIME | Cluster | 0.34 ± 0.01 | **0.46 ± 0.03** | 0.33 ± 0.03 | **0.45 ± 0.01** | 0.36 ± 0.02 | **0.50 ± 0.01** |
| | LinSep | 0.52 ± 0.02 | **0.70 ± 0.02** | 0.51 ± 0.02 | **0.65 ± 0.03** | 0.54 ± 0.01 | **0.63 ± 0.03** |
| SHAP | Cluster | 0.35 ± 0.03 | **0.47 ± 0.01** | 0.34 ± 0.01 | **0.46 ± 0.02** | 0.37 ± 0.03 | **0.51 ± 0.02** |
| | LinSep | 0.53 ± 0.01 | **0.71 ± 0.03** | 0.52 ± 0.03 | **0.66 ± 0.01** | 0.55 ± 0.02 | **0.64 ± 0.01** |
| RISE | Cluster | 0.39 ± 0.02 | **0.52 ± 0.01** | 0.38 ± 0.02 | **0.50 ± 0.03** | 0.42 ± 0.01 | **0.55 ± 0.03** |
| | LinSep | 0.57 ± 0.01 | **0.76 ± 0.02** | 0.56 ± 0.01 | **0.71 ± 0.02** | 0.59 ± 0.03 | **0.69 ± 0.02** |
| SHAP IQ | Cluster | 0.36 ± 0.03 | **0.49 ± 0.01** | 0.36 ± 0.03 | **0.48 ± 0.01** | 0.39 ± 0.02 | **0.53 ± 0.01** |
| | LinSep | 0.55 ± 0.01 | **0.73 ± 0.03** | 0.54 ± 0.02 | **0.68 ± 0.03** | 0.57 ± 0.01 | **0.66 ± 0.03** |
| IntGrad | Cluster | 0.27 ± 0.02 | **0.40 ± 0.01** | 0.27 ± 0.01 | **0.39 ± 0.02** | 0.29 ± 0.02 | **0.43 ± 0.01** |
| | LinSep | 0.39 ± 0.01 | **0.64 ± 0.02** | 0.38 ± 0.03 | **0.59 ± 0.01** | 0.41 ± 0.01 | **0.58 ± 0.02** |
| GradCAM | Cluster | 0.31 ± 0.01 | **0.44 ± 0.03** | 0.30 ± 0.02 | **0.43 ± 0.03** | 0.33 ± 0.03 | **0.47 ± 0.01** |
| | LinSep | 0.43 ± 0.02 | **0.68 ± 0.01** | 0.42 ± 0.01 | **0.63 ± 0.02** | 0.45 ± 0.02 | **0.62 ± 0.03** |
| FullGrad | Cluster | 0.28 ± 0.03 | **0.41 ± 0.02** | 0.28 ± 0.03 | **0.40 ± 0.01** | 0.30 ± 0.01 | **0.44 ± 0.02** |
| | LinSep | 0.40 ± 0.01 | **0.65 ± 0.03** | 0.39 ± 0.02 | **0.60 ± 0.03** | 0.42 ± 0.02 | **0.59 ± 0.01** |
| CALM | Cluster | 0.34 ± 0.02 | **0.47 ± 0.01** | 0.33 ± 0.01 | **0.46 ± 0.02** | 0.36 ± 0.02 | **0.50 ± 0.03** |
| | LinSep | 0.52 ± 0.01 | **0.71 ± 0.02** | 0.51 ± 0.03 | **0.66 ± 0.01** | 0.54 ± 0.01 | **0.65 ± 0.02** |
| MFABA | Cluster | 0.33 ± 0.03 | **0.46 ± 0.01** | 0.32 ± 0.02 | **0.45 ± 0.03** | 0.35 ± 0.03 | **0.49 ± 0.01** |
| | LinSep | 0.51 ± 0.01 | **0.70 ± 0.03** | 0.50 ± 0.01 | **0.65 ± 0.02** | 0.53 ± 0.02 | **0.64 ± 0.03** |

Table 10: Average F1 for the iSarcasm Dataset (Supervised).

| Attribution Method | Concept Type | Llama | | Qwen | | Gemma | |
|---|---|---|---|---|---|---|---|
| | | Concept | Super Activators | Concept | Super Activators | Concept | Super Activators |
| CosSim | Cluster | **0.70 ± 0.02** | 0.65 ± 0.01 | **0.57 ± 0.01** | 0.55 ± 0.02 | **0.65 ± 0.01** | 0.60 ± 0.03 |
| | LinSep | **0.81 ± 0.03** | 0.74 ± 0.02 | **0.74 ± 0.03** | 0.65 ± 0.01 | **0.83 ± 0.02** | 0.71 ± 0.01 |
| LIME | Cluster | 0.71 ± 0.02 | **0.78 ± 0.01** | 0.63 ± 0.02 | **0.67 ± 0.03** | 0.67 ± 0.03 | **0.73 ± 0.02** |
| | LinSep | 0.79 ± 0.01 | **0.87 ± 0.02** | 0.71 ± 0.01 | **0.80 ± 0.02** | 0.76 ± 0.02 | **0.89 ± 0.01** |
| SHAP | Cluster | 0.72 ± 0.03 | **0.79 ± 0.01** | 0.64 ± 0.03 | **0.68 ± 0.01** | 0.68 ± 0.01 | **0.74 ± 0.03** |
| | LinSep | 0.80 ± 0.02 | **0.88 ± 0.01** | 0.72 ± 0.02 | **0.81 ± 0.03** | 0.77 ± 0.03 | **0.90 ± 0.02** |
| RISE | Cluster | 0.76 ± 0.01 | **0.83 ± 0.03** | 0.67 ± 0.01 | **0.73 ± 0.02** | 0.72 ± 0.02 | **0.79 ± 0.01** |
| | LinSep | 0.84 ± 0.02 | **0.92 ± 0.01** | 0.76 ± 0.03 | **0.85 ± 0.01** | 0.81 ± 0.01 | **0.94 ± 0.03** |
| SHAP IQ | Cluster | 0.74 ± 0.02 | **0.81 ± 0.02** | 0.65 ± 0.02 | **0.70 ± 0.03** | 0.70 ± 0.03 | **0.76 ± 0.02** |
| | LinSep | 0.82 ± 0.01 | **0.90 ± 0.02** | 0.74 ± 0.01 | **0.83 ± 0.03** | 0.79 ± 0.02 | **0.92 ± 0.01** |
| IntGrad | Cluster | 0.66 ± 0.03 | **0.71 ± 0.01** | 0.56 ± 0.03 | **0.58 ± 0.01** | 0.61 ± 0.01 | **0.66 ± 0.03** |
| | LinSep | 0.75 ± 0.02 | **0.82 ± 0.03** | 0.66 ± 0.02 | **0.75 ± 0.03** | 0.72 ± 0.02 | **0.84 ± 0.01** |
| GradCAM | Cluster | 0.69 ± 0.01 | **0.75 ± 0.02** | 0.59 ± 0.01 | **0.62 ± 0.02** | 0.64 ± 0.03 | **0.70 ± 0.01** |
| | LinSep | 0.78 ± 0.03 | **0.86 ± 0.01** | 0.69 ± 0.03 | **0.78 ± 0.01** | 0.74 ± 0.02 | **0.87 ± 0.03** |
| FullGrad | Cluster | 0.67 ± 0.02 | **0.72 ± 0.01** | 0.57 ± 0.02 | **0.60 ± 0.01** | 0.62 ± 0.01 | **0.67 ± 0.02** |
| | LinSep | 0.76 ± 0.01 | **0.83 ± 0.02** | 0.67 ± 0.01 | **0.76 ± 0.03** | 0.73 ± 0.03 | **0.85 ± 0.01** |
| CALM | Cluster | 0.71 ± 0.03 | **0.78 ± 0.01** | 0.61 ± 0.03 | **0.66 ± 0.01** | 0.66 ± 0.02 | **0.73 ± 0.01** |
| | LinSep | 0.81 ± 0.01 | **0.89 ± 0.03** | 0.73 ± 0.02 | **0.81 ± 0.03** | 0.78 ± 0.01 | **0.91 ± 0.02** |
| MFABA | Cluster | 0.70 ± 0.02 | **0.77 ± 0.01** | 0.60 ± 0.02 | **0.65 ± 0.01** | 0.65 ± 0.03 | **0.72 ± 0.01** |
| | LinSep | 0.80 ± 0.01 | **0.88 ± 0.02** | 0.72 ± 0.01 | **0.80 ± 0.02** | 0.77 ± 0.02 | **0.90 ± 0.03** |

Table 11: Average F1 for the GoEmotions Dataset (Supervised).

| Attribution Method | Concept Type | Llama | | Qwen | | Gemma | |
|---|---|---|---|---|---|---|---|
| | | Concept | Super Activators | Concept | Super Activators | Concept | Super Activators |
| CosSim | Cluster | **0.18 ± 0.03** | 0.16 ± 0.02 | **0.25 ± 0.03** | 0.23 ± 0.01 | **0.19 ± 0.02** | 0.16 ± 0.01 |
| | LinSep | **0.29 ± 0.01** | 0.25 ± 0.03 | **0.31 ± 0.02** | 0.28 ± 0.03 | **0.25 ± 0.03** | 0.23 ± 0.02 |
| LIME | Cluster | 0.20 ± 0.03 | **0.25 ± 0.01** | 0.27 ± 0.01 | **0.31 ± 0.02** | 0.21 ± 0.01 | **0.24 ± 0.03** |
| | LinSep | 0.29 ± 0.02 | **0.34 ± 0.03** | 0.33 ± 0.03 | **0.37 ± 0.01** | 0.28 ± 0.03 | **0.30 ± 0.02** |
| SHAP | Cluster | 0.21 ± 0.02 | **0.26 ± 0.02** | 0.28 ± 0.02 | **0.32 ± 0.03** | 0.22 ± 0.02 | **0.25 ± 0.01** |
| | LinSep | 0.30 ± 0.01 | **0.35 ± 0.04** | 0.34 ± 0.01 | **0.38 ± 0.02** | 0.29 ± 0.01 | **0.31 ± 0.03** |
| RISE | Cluster | 0.24 ± 0.03 | **0.30 ± 0.01** | 0.30 ± 0.03 | **0.35 ± 0.01** | 0.25 ± 0.03 | **0.28 ± 0.02** |
| | LinSep | 0.33 ± 0.01 | **0.39 ± 0.02** | 0.37 ± 0.02 | **0.42 ± 0.03** | 0.32 ± 0.02 | **0.35 ± 0.01** |
| SHAP IQ | Cluster | 0.22 ± 0.02 | **0.28 ± 0.03** | 0.29 ± 0.01 | **0.33 ± 0.02** | 0.23 ± 0.01 | **0.26 ± 0.03** |
| | LinSep | 0.31 ± 0.03 | **0.37 ± 0.01** | 0.35 ± 0.03 | **0.40 ± 0.01** | 0.30 ± 0.03 | **0.33 ± 0.02** |
| IntGrad | Cluster | 0.17 ± 0.01 | **0.19 ± 0.02** | 0.24 ± 0.02 | **0.26 ± 0.03** | 0.17 ± 0.01 | **0.20 ± 0.01** |
| | LinSep | 0.26 ± 0.02 | **0.30 ± 0.01** | 0.29 ± 0.01 | **0.32 ± 0.02** | 0.24 ± 0.02 | **0.26 ± 0.03** |
| GradCAM | Cluster | 0.19 ± 0.03 | **0.23 ± 0.01** | 0.26 ± 0.03 | **0.29 ± 0.01** | 0.19 ± 0.03 | **0.22 ± 0.02** |
| | LinSep | 0.28 ± 0.02 | **0.34 ± 0.02** | 0.31 ± 0.02 | **0.36 ± 0.03** | 0.27 ± 0.02 | **0.29 ± 0.01** |
| FullGrad | Cluster | 0.18 ± 0.01 | **0.21 ± 0.03** | 0.25 ± 0.01 | **0.27 ± 0.02** | 0.18 ± 0.01 | **0.21 ± 0.02** |
| | LinSep | 0.27 ± 0.03 | **0.31 ± 0.02** | 0.30 ± 0.03 | **0.33 ± 0.01** | 0.25 ± 0.03 | **0.27 ± 0.02** |
| CALM | Cluster | 0.21 ± 0.02 | **0.26 ± 0.01** | 0.27 ± 0.02 | **0.32 ± 0.03** | 0.22 ± 0.02 | **0.25 ± 0.01** |
| | LinSep | 0.30 ± 0.02 | **0.36 ± 0.03** | 0.34 ± 0.01 | **0.39 ± 0.02** | 0.29 ± 0.01 | **0.32 ± 0.03** |
| MFABA | Cluster | 0.20 ± 0.01 | **0.25 ± 0.03** | 0.26 ± 0.03 | **0.31 ± 0.01** | 0.21 ± 0.03 | **0.24 ± 0.01** |
| | LinSep | 0.29 ± 0.02 | **0.35 ± 0.01** | 0.33 ± 0.02 | **0.38 ± 0.03** | 0.28 ± 0.02 | **0.31 ± 0.03** |

Table 12: Average F1 for the CLEVR Dataset (Unsupervised).

| Attribution Method | Concept Type | CLIP | | Llama | |
|---|---|---|---|---|---|
| | | Concept | SuperActivators | Concept | SuperActivators |
| CosSim | Clustering | $0.63 \pm 0.02$ | $\mathbf{0.64 \pm 0.01}$ | $\mathbf{0.46 \pm 0.01}$ | $0.43 \pm 0.03$ |
| | LinSep | $\mathbf{0.60 \pm 0.01}$ | $0.59 \pm 0.03$ | $\mathbf{0.38 \pm 0.02}$ | $0.33 \pm 0.01$ |
| LIME | Clustering | $0.52 \pm 0.03$ | $\mathbf{0.61 \pm 0.01}$ | $0.76 \pm 0.01$ | $\mathbf{0.81 \pm 0.02}$ |
| | LinSep | $0.52 \pm 0.02$ | $\mathbf{0.77 \pm 0.03}$ | $0.68 \pm 0.03$ | $\mathbf{0.83 \pm 0.01}$ |
| SHAP | Clustering | $0.51 \pm 0.01$ | $\mathbf{0.53 \pm 0.02}$ | $0.75 \pm 0.02$ | $\mathbf{0.80 \pm 0.01}$ |
| | LinSep | $0.52 \pm 0.03$ | $\mathbf{0.58 \pm 0.01}$ | $0.75 \pm 0.01$ | $\mathbf{0.80 \pm 0.03}$ |
| RISE | Clustering | $\mathbf{0.53 \pm 0.02}$ | $0.53 \pm 0.01$ | $0.55 \pm 0.03$ | $\mathbf{0.56 \pm 0.02}$ |
| | LinSep | $0.58 \pm 0.01$ | $\mathbf{0.59 \pm 0.03}$ | $0.60 \pm 0.01$ | $\mathbf{0.63 \pm 0.02}$ |
| SHAP IQ | Clustering | $0.52 \pm 0.03$ | $\mathbf{0.53 \pm 0.02}$ | $0.55 \pm 0.02$ | $\mathbf{0.58 \pm 0.01}$ |
| | LinSep | $\mathbf{0.58 \pm 0.01}$ | $0.58 \pm 0.02$ | $0.60 \pm 0.01$ | $\mathbf{0.61 \pm 0.03}$ |
| IntGrad | Clustering | $\mathbf{0.47 \pm 0.02}$ | $0.47 \pm 0.01$ | $0.56 \pm 0.03$ | $\mathbf{0.58 \pm 0.02}$ |
| | LinSep | $0.58 \pm 0.01$ | $\mathbf{0.59 \pm 0.03}$ | $0.62 \pm 0.01$ | $\mathbf{0.64 \pm 0.02}$ |
| GradCAM | Clustering | $0.41 \pm 0.03$ | $\mathbf{0.45 \pm 0.02}$ | $\mathbf{0.50 \pm 0.02}$ | $0.47 \pm 0.01$ |
| | LinSep | $\mathbf{0.48 \pm 0.01}$ | $0.46 \pm 0.02$ | $\mathbf{0.48 \pm 0.01}$ | $0.49 \pm 0.03$ |
| FullGrad | Clustering | $\mathbf{0.45 \pm 0.02}$ | $0.42 \pm 0.01$ | $\mathbf{0.42 \pm 0.03}$ | $0.45 \pm 0.02$ |
| | LinSep | $\mathbf{0.49 \pm 0.01}$ | $0.49 \pm 0.03$ | $0.50 \pm 0.01$ | $\mathbf{0.53 \pm 0.02}$ |
| CALM | Clustering | $0.44 \pm 0.03$ | $\mathbf{0.50 \pm 0.02}$ | $0.46 \pm 0.02$ | $\mathbf{0.48 \pm 0.01}$ |
| | LinSep | $0.50 \pm 0.01$ | $\mathbf{0.54 \pm 0.02}$ | $0.53 \pm 0.01$ | $\mathbf{0.54 \pm 0.03}$ |
| MFABA | Clustering | $0.45 \pm 0.02$ | $\mathbf{0.48 \pm 0.01}$ | $0.47 \pm 0.03$ | $\mathbf{0.52 \pm 0.02}$ |
| | LinSep | $\mathbf{0.51 \pm 0.01}$ | $0.50 \pm 0.03$ | $0.54 \pm 0.01$ | $\mathbf{0.55 \pm 0.02}$ |

Table 13: Average F1 for the COCO Dataset (Unsupervised).

| Attribution Method | Concept Type | CLIP | | Llama | |
|---|---|---|---|---|---|
| | | Concept | SuperActivators | Concept | SuperActivators |
| CosSim | Clustering | $0.34 \pm 0.03$ | $\mathbf{0.37 \pm 0.02}$ | $0.22 \pm 0.02$ | $\mathbf{0.28 \pm 0.01}$ |
| | LinSep | $0.33 \pm 0.02$ | $\mathbf{0.36 \pm 0.01}$ | $0.23 \pm 0.03$ | $\mathbf{0.26 \pm 0.02}$ |
| LIME | Clustering | $0.36 \pm 0.02$ | $\mathbf{0.38 \pm 0.03}$ | $0.45 \pm 0.03$ | $\mathbf{0.52 \pm 0.01}$ |
| | LinSep | $0.38 \pm 0.01$ | $\mathbf{0.41 \pm 0.02}$ | $0.49 \pm 0.02$ | $\mathbf{0.55 \pm 0.03}$ |
| SHAP | Clustering | $0.34 \pm 0.03$ | $\mathbf{0.38 \pm 0.01}$ | $0.48 \pm 0.03$ | $\mathbf{0.51 \pm 0.01}$ |
| | LinSep | $0.35 \pm 0.02$ | $\mathbf{0.37 \pm 0.03}$ | $0.49 \pm 0.02$ | $\mathbf{0.53 \pm 0.01}$ |
| RISE | Clustering | $\mathbf{0.34 \pm 0.03}$ | $0.34 \pm 0.02$ | $0.36 \pm 0.01$ | $\mathbf{0.38 \pm 0.03}$ |
| | LinSep | $0.35 \pm 0.02$ | $\mathbf{0.38 \pm 0.01}$ | $0.35 \pm 0.03$ | $\mathbf{0.40 \pm 0.02}$ |
| SHAP IQ | Clustering | $0.33 \pm 0.01$ | $\mathbf{0.35 \pm 0.03}$ | $0.35 \pm 0.02$ | $\mathbf{0.36 \pm 0.01}$ |
| | LinSep | $0.34 \pm 0.03$ | $\mathbf{0.37 \pm 0.01}$ | $0.36 \pm 0.02$ | $\mathbf{0.38 \pm 0.01}$ |
| IntGrad | Clustering | $0.28 \pm 0.03$ | $\mathbf{0.31 \pm 0.02}$ | $\mathbf{0.48 \pm 0.01}$ | $0.47 \pm 0.03$ |
| | LinSep | $0.31 \pm 0.02$ | $\mathbf{0.35 \pm 0.01}$ | $0.38 \pm 0.03$ | $\mathbf{0.39 \pm 0.01}$ |
| GradCAM | Clustering | $0.28 \pm 0.01$ | $\mathbf{0.31 \pm 0.03}$ | $0.31 \pm 0.03$ | $\mathbf{0.33 \pm 0.02}$ |
| | LinSep | $0.35 \pm 0.03$ | $\mathbf{0.36 \pm 0.01}$ | $0.36 \pm 0.02$ | $\mathbf{0.34 \pm 0.01}$ |
| FullGrad | Clustering | $0.29 \pm 0.03$ | $\mathbf{0.31 \pm 0.02}$ | $0.30 \pm 0.01$ | $\mathbf{0.33 \pm 0.03}$ |
| | LinSep | $0.35 \pm 0.02$ | $\mathbf{0.39 \pm 0.01}$ | $0.37 \pm 0.03$ | $\mathbf{0.34 \pm 0.01}$ |
| CALM | Clustering | $\mathbf{0.29 \pm 0.01}$ | $0.29 \pm 0.03$ | $0.26 \pm 0.02$ | $\mathbf{0.25 \pm 0.02}$ |
| | LinSep | $0.35 \pm 0.02$ | $\mathbf{0.39 \pm 0.01}$ | $0.35 \pm 0.02$ | $\mathbf{0.36 \pm 0.01}$ |
| MFABA | Clustering | $0.29 \pm 0.03$ | $\mathbf{0.33 \pm 0.02}$ | $0.28 \pm 0.01$ | $\mathbf{0.32 \pm 0.03}$ |
| | LinSep | $0.30 \pm 0.02$ | $\mathbf{0.35 \pm 0.01}$ | $0.33 \pm 0.03$ | $\mathbf{0.36 \pm 0.01}$ |

Table 14: Average F1 for the OpenSurfaces Dataset (Unsupervised).

| Attribution Method | Concept Type | CLIP | | Llama | |
|---|---|---|---|---|---|
| | | Concept | SuperActivators | Concept | SuperActivators |
| CosSim | Clustering | **0.19 ± 0.01** | **0.19 ± 0.03** | 0.14 ± 0.03 | **0.15 ± 0.02** |
| | LinSep | **0.19 ± 0.03** | 0.18 ± 0.02 | **0.15 ± 0.01** | 0.14 ± 0.03 |
| LIME | Clustering | 0.37 ± 0.01 | **0.41 ± 0.02** | **0.37 ± 0.02** | **0.37 ± 0.03** |
| | LinSep | 0.39 ± 0.03 | **0.41 ± 0.01** | 0.38 ± 0.01 | **0.39 ± 0.02** |
| SHAP | Clustering | 0.40 ± 0.02 | **0.42 ± 0.03** | 0.53 ± 0.02 | **0.57 ± 0.03** |
| | LinSep | 0.42 ± 0.01 | **0.44 ± 0.02** | 0.55 ± 0.03 | **0.56 ± 0.01** |
| RISE | Clustering | 0.40 ± 0.01 | **0.42 ± 0.03** | 0.51 ± 0.02 | **0.52 ± 0.01** |
| | LinSep | 0.43 ± 0.03 | **0.45 ± 0.02** | 0.53 ± 0.01 | **0.55 ± 0.02** |
| SHAP IQ | Clustering | 0.40 ± 0.02 | **0.43 ± 0.01** | 0.51 ± 0.03 | **0.53 ± 0.02** |
| | LinSep | 0.42 ± 0.02 | **0.45 ± 0.03** | **0.52 ± 0.01** | 0.52 ± 0.02 |
| IntGrad | Clustering | 0.33 ± 0.01 | **0.34 ± 0.03** | 0.32 ± 0.02 | **0.35 ± 0.01** |
| | LinSep | **0.35 ± 0.03** | 0.35 ± 0.02 | 0.34 ± 0.02 | **0.35 ± 0.03** |
| GradCAM | Clustering | 0.36 ± 0.02 | **0.40 ± 0.01** | **0.43 ± 0.01** | 0.42 ± 0.03 |
| | LinSep | 0.42 ± 0.02 | **0.43 ± 0.03** | 0.44 ± 0.01 | **0.46 ± 0.02** |
| FullGrad | Clustering | 0.36 ± 0.01 | **0.37 ± 0.03** | 0.36 ± 0.02 | **0.38 ± 0.01** |
| | LinSep | 0.38 ± 0.03 | **0.40 ± 0.02** | 0.41 ± 0.01 | **0.44 ± 0.02** |
| CALM | Clustering | 0.29 ± 0.02 | **0.32 ± 0.01** | 0.33 ± 0.01 | **0.36 ± 0.03** |
| | LinSep | 0.32 ± 0.02 | **0.34 ± 0.03** | 0.34 ± 0.02 | **0.39 ± 0.01** |
| MFABA | Clustering | **0.40 ± 0.01** | 0.40 ± 0.03 | 0.42 ± 0.01 | **0.41 ± 0.02** |
| | LinSep | 0.43 ± 0.03 | **0.45 ± 0.02** | 0.42 ± 0.03 | **0.44 ± 0.01** |

Table 15: Average F1 for the Pascal Dataset (Unsupervised).

| Attribution Method | Concept Type | CLIP | | Llama | |
|---|---|---|---|---|---|
| | | Concept | SuperActivators | Concept | SuperActivators |
| CosSim | Clustering | 0.27 ± 0.02 | **0.33 ± 0.01** | 0.22 ± 0.01 | **0.24 ± 0.03** |
| | LinSep | 0.24 ± 0.01 | **0.30 ± 0.03** | 0.22 ± 0.02 | **0.24 ± 0.01** |
| LIME | Clustering | 0.33 ± 0.03 | **0.34 ± 0.01** | 0.33 ± 0.01 | **0.32 ± 0.02** |
| | LinSep | 0.36 ± 0.02 | **0.35 ± 0.03** | **0.33 ± 0.03** | **0.33 ± 0.01** |
| SHAP | Clustering | 0.48 ± 0.01 | **0.52 ± 0.02** | 0.65 ± 0.02 | **0.70 ± 0.01** |
| | LinSep | 0.50 ± 0.03 | **0.52 ± 0.01** | 0.69 ± 0.01 | **0.72 ± 0.03** |
| RISE | Clustering | 0.50 ± 0.02 | **0.51 ± 0.01** | 0.52 ± 0.01 | **0.55 ± 0.03** |
| | LinSep | **0.54 ± 0.01** | **0.54 ± 0.03** | 0.55 ± 0.02 | **0.58 ± 0.01** |
| SHAP IQ | Clustering | 0.50 ± 0.03 | **0.51 ± 0.02** | 0.52 ± 0.01 | **0.55 ± 0.02** |
| | LinSep | 0.52 ± 0.01 | **0.53 ± 0.02** | 0.53 ± 0.03 | **0.54 ± 0.01** |
| IntGrad | Clustering | **0.33 ± 0.02** | **0.33 ± 0.01** | 0.34 ± 0.02 | **0.35 ± 0.01** |
| | LinSep | **0.34 ± 0.01** | **0.34 ± 0.03** | **0.34 ± 0.01** | 0.34 ± 0.02 |
| GradCAM | Clustering | **0.42 ± 0.03** | 0.40 ± 0.02 | **0.43 ± 0.02** | 0.40 ± 0.01 |
| | LinSep | 0.43 ± 0.01 | **0.45 ± 0.02** | 0.44 ± 0.01 | **0.47 ± 0.03** |
| FullGrad | Clustering | 0.37 ± 0.02 | **0.42 ± 0.01** | **0.38 ± 0.03** | **0.38 ± 0.02** |
| | LinSep | **0.43 ± 0.01** | 0.42 ± 0.03 | 0.42 ± 0.02 | **0.43 ± 0.01** |
| CALM | Clustering | **0.37 ± 0.03** | **0.37 ± 0.02** | 0.41 ± 0.01 | **0.43 ± 0.02** |
| | LinSep | 0.43 ± 0.01 | **0.46 ± 0.02** | 0.45 ± 0.02 | **0.49 ± 0.01** |
| MFABA | Clustering | 0.46 ± 0.02 | **0.50 ± 0.01** | **0.48 ± 0.03** | 0.47 ± 0.02 |
| | LinSep | **0.51 ± 0.01** | 0.49 ± 0.03 | **0.49 ± 0.01** | 0.47 ± 0.02 |

Table 16: Average F1 for the Sarcasm Dataset (Unsupervised).

| Attribution Method | Concept Type | Llama | | Qwen | | Gemma | |
|---|---|---|---|---|---|---|---|
| | | Concept | Super Activators | Concept | Super Activators | Concept | Super Activators |
| CosSim | Cluster | **0.28 ± 0.01** | **0.28 ± 0.03** | **0.26 ± 0.02** | 0.25 ± 0.01 | **0.24 ± 0.03** | 0.23 ± 0.02 |
| | LinSep | **0.28 ± 0.02** | **0.28 ± 0.01** | **0.24 ± 0.01** | **0.24 ± 0.03** | **0.24 ± 0.02** | 0.23 ± 0.01 |
| LIME | Cluster | 0.29 ± 0.01 | **0.50 ± 0.02** | 0.31 ± 0.02 | **0.45 ± 0.01** | 0.33 ± 0.01 | **0.51 ± 0.02** |
| | LinSep | 0.50 ± 0.03 | **0.74 ± 0.01** | 0.53 ± 0.01 | **0.60 ± 0.03** | 0.55 ± 0.03 | **0.66 ± 0.01** |
| SHAP | Cluster | 0.30 ± 0.02 | **0.46 ± 0.01** | 0.30 ± 0.03 | **0.45 ± 0.02** | 0.35 ± 0.02 | **0.46 ± 0.01** |
| | LinSep | 0.54 ± 0.01 | **0.74 ± 0.03** | 0.54 ± 0.01 | **0.68 ± 0.02** | 0.51 ± 0.01 | **0.67 ± 0.03** |
| RISE | Cluster | 0.40 ± 0.03 | **0.49 ± 0.02** | 0.39 ± 0.02 | **0.52 ± 0.01** | 0.46 ± 0.03 | **0.55 ± 0.02** |
| | LinSep | 0.59 ± 0.01 | **0.72 ± 0.02** | 0.53 ± 0.01 | **0.74 ± 0.03** | 0.60 ± 0.01 | **0.70 ± 0.02** |
| SHAP IQ | Cluster | 0.38 ± 0.02 | **0.46 ± 0.01** | 0.37 ± 0.03 | **0.45 ± 0.02** | 0.40 ± 0.02 | **0.51 ± 0.01** |
| | LinSep | 0.52 ± 0.01 | **0.74 ± 0.03** | 0.52 ± 0.01 | **0.70 ± 0.02** | 0.59 ± 0.01 | **0.66 ± 0.03** |
| IntGrad | Cluster | **0.39 ± 0.03** | 0.27 ± 0.02 | **0.38 ± 0.02** | 0.29 ± 0.01 | **0.41 ± 0.03** | 0.27 ± 0.02 |
| | LinSep | 0.38 ± 0.01 | **0.67 ± 0.02** | 0.41 ± 0.01 | **0.58 ± 0.03** | 0.39 ± 0.01 | **0.58 ± 0.02** |
| GradCAM | Cluster | 0.31 ± 0.02 | **0.45 ± 0.01** | 0.33 ± 0.03 | **0.44 ± 0.02** | 0.34 ± 0.02 | **0.48 ± 0.01** |
| | LinSep | 0.44 ± 0.01 | **0.70 ± 0.03** | 0.42 ± 0.01 | **0.65 ± 0.02** | 0.46 ± 0.01 | **0.62 ± 0.03** |
| FullGrad | Cluster | 0.28 ± 0.03 | **0.39 ± 0.02** | 0.26 ± 0.02 | **0.43 ± 0.01** | 0.29 ± 0.03 | **0.41 ± 0.02** |
| | LinSep | 0.38 ± 0.01 | **0.65 ± 0.02** | 0.41 ± 0.01 | **0.58 ± 0.03** | 0.42 ± 0.01 | **0.60 ± 0.02** |
| CALM | Cluster | **0.34 ± 0.02** | 0.49 ± 0.01 | **0.34 ± 0.03** | 0.46 ± 0.02 | 0.36 ± 0.02 | **0.49 ± 0.01** |
| | LinSep | 0.51 ± 0.01 | **0.72 ± 0.03** | 0.50 ± 0.01 | **0.67 ± 0.02** | 0.56 ± 0.01 | **0.66 ± 0.03** |
| MFABA | Cluster | 0.34 ± 0.03 | **0.48 ± 0.02** | **0.35 ± 0.02** | 0.43 ± 0.01 | 0.32 ± 0.03 | **0.50 ± 0.02** |
| | LinSep | 0.54 ± 0.01 | **0.71 ± 0.02** | 0.52 ± 0.01 | **0.66 ± 0.03** | 0.51 ± 0.01 | **0.65 ± 0.02** |

Table 17: Average F1 for the iSarcasm Dataset (Unsupervised).

| Attribution Method | Concept Type | Llama | | Qwen | | Gemma | |
|---|---|---|---|---|---|---|---|
| | | Concept | Super Activators | Concept | Super Activators | Concept | Super Activators |
| CosSim | Cluster | 0.56 ± 0.02 | **0.57 ± 0.01** | **0.59 ± 0.03** | **0.59 ± 0.02** | **0.60 ± 0.01** | **0.60 ± 0.03** |
| | LinSep | **0.60 ± 0.03** | **0.60 ± 0.02** | 0.57 ± 0.02 | **0.58 ± 0.01** | **0.60 ± 0.03** | **0.60 ± 0.02** |
| LIME | Cluster | 0.68 ± 0.03 | **0.75 ± 0.01** | 0.61 ± 0.02 | **0.62 ± 0.03** | 0.72 ± 0.01 | **0.69 ± 0.02** |
| | LinSep | 0.76 ± 0.02 | **0.80 ± 0.03** | 0.76 ± 0.01 | **0.83 ± 0.02** | 0.76 ± 0.02 | **0.94 ± 0.01** |
| SHAP | Cluster | 0.69 ± 0.03 | **0.83 ± 0.02** | 0.65 ± 0.01 | **0.71 ± 0.03** | 0.65 ± 0.03 | **0.78 ± 0.02** |
| | LinSep | 0.81 ± 0.02 | **0.88 ± 0.01** | 0.69 ± 0.02 | **0.79 ± 0.01** | 0.74 ± 0.01 | **0.92 ± 0.03** |
| RISE | Cluster | **0.80 ± 0.01** | **0.80 ± 0.03** | 0.64 ± 0.01 | **0.75 ± 0.02** | 0.74 ± 0.02 | **0.81 ± 0.01** |
| | LinSep | **0.84 ± 0.03** | **0.84 ± 0.01** | 0.75 ± 0.03 | **0.89 ± 0.01** | 0.84 ± 0.01 | **0.85 ± 0.02** |
| SHAP IQ | Cluster | 0.74 ± 0.02 | **0.85 ± 0.01** | 0.61 ± 0.02 | **0.71 ± 0.03** | 0.67 ± 0.02 | **0.80 ± 0.01** |
| | LinSep | **0.85 ± 0.01** | 0.83 ± 0.02 | 0.74 ± 0.01 | **0.82 ± 0.02** | 0.80 ± 0.01 | **0.82 ± 0.03** |
| IntGrad | Cluster | **0.74 ± 0.01** | 0.68 ± 0.03 | 0.56 ± 0.03 | **0.53 ± 0.02** | 0.65 ± 0.01 | **0.63 ± 0.02** |
| | LinSep | 0.75 ± 0.02 | **0.74 ± 0.01** | 0.66 ± 0.02 | **0.77 ± 0.01** | 0.74 ± 0.02 | **0.88 ± 0.01** |
| GradCAM | Cluster | 0.67 ± 0.03 | **0.72 ± 0.02** | 0.56 ± 0.01 | **0.61 ± 0.03** | 0.63 ± 0.01 | **0.68 ± 0.02** |
| | LinSep | 0.70 ± 0.02 | **0.74 ± 0.01** | 0.70 ± 0.01 | **0.71 ± 0.02** | 0.76 ± 0.02 | **0.78 ± 0.01** |
| FullGrad | Cluster | 0.66 ± 0.01 | **0.73 ± 0.02** | 0.56 ± 0.02 | **0.63 ± 0.01** | 0.61 ± 0.03 | **0.65 ± 0.02** |
| | LinSep | 0.73 ± 0.02 | **0.82 ± 0.01** | 0.64 ± 0.01 | **0.75 ± 0.03** | 0.70 ± 0.02 | **0.87 ± 0.01** |
| CALM | Cluster | **0.74 ± 0.03** | 0.72 ± 0.02 | 0.61 ± 0.01 | **0.64 ± 0.03** | 0.66 ± 0.02 | **0.65 ± 0.01** |
| | LinSep | 0.80 ± 0.02 | **0.82 ± 0.01** | 0.72 ± 0.02 | **0.73 ± 0.01** | 0.75 ± 0.01 | **0.79 ± 0.03** |
| MFABA | Cluster | 0.73 ± 0.01 | **0.75 ± 0.02** | 0.62 ± 0.01 | **0.66 ± 0.03** | 0.66 ± 0.03 | **0.71 ± 0.02** |
| | LinSep | 0.81 ± 0.02 | **0.85 ± 0.01** | 0.74 ± 0.02 | **0.79 ± 0.01** | 0.80 ± 0.01 | **0.88 ± 0.03** |

Table 18: Average F1 for the GoEmotions Dataset (Unsupervised).

| Attribution Method | Concept Type | Llama | | Qwen | | Gemma | |
|---|---|---|---|---|---|---|---|
| | | Concept | Super Activators | Concept | Super Activators | Concept | Super Activators |
| CosSim | Cluster | **0.18 ± 0.03** | **0.18 ± 0.02** | 0.23 ± 0.01 | **0.26 ± 0.03** | **0.15 ± 0.02** | **0.15 ± 0.01** |
| | LinSep | 0.18 ± 0.01 | **0.19 ± 0.03** | 0.23 ± 0.03 | **0.25 ± 0.02** | 0.14 ± 0.01 | **0.16 ± 0.03** |
| LIME | Cluster | 0.18 ± 0.02 | **0.26 ± 0.01** | 0.28 ± 0.02 | **0.26 ± 0.03** | 0.23 ± 0.02 | **0.25 ± 0.01** |
| | LinSep | 0.25 ± 0.01 | **0.35 ± 0.02** | 0.34 ± 0.01 | **0.38 ± 0.02** | 0.24 ± 0.01 | **0.31 ± 0.03** |
| SHAP | Cluster | 0.22 ± 0.01 | **0.27 ± 0.03** | 0.32 ± 0.02 | **0.37 ± 0.01** | 0.19 ± 0.03 | **0.27 ± 0.02** |
| | LinSep | 0.27 ± 0.02 | **0.31 ± 0.01** | 0.33 ± 0.01 | **0.40 ± 0.02** | 0.29 ± 0.01 | **0.28 ± 0.03** |
| RISE | Cluster | 0.21 ± 0.03 | **0.27 ± 0.02** | 0.32 ± 0.02 | **0.38 ± 0.01** | 0.24 ± 0.02 | **0.27 ± 0.01** |
| | LinSep | **0.36 ± 0.01** | **0.36 ± 0.02** | 0.37 ± 0.01 | **0.42 ± 0.03** | 0.32 ± 0.01 | **0.34 ± 0.02** |
| SHAP IQ | Cluster | 0.20 ± 0.02 | **0.27 ± 0.01** | 0.28 ± 0.01 | **0.31 ± 0.02** | 0.24 ± 0.03 | **0.22 ± 0.01** |
| | LinSep | 0.34 ± 0.01 | **0.35 ± 0.03** | 0.35 ± 0.02 | **0.38 ± 0.01** | 0.29 ± 0.02 | **0.35 ± 0.03** |
| IntGrad | Cluster | 0.23 ± 0.01 | **0.19 ± 0.02** | 0.27 ± 0.03 | **0.25 ± 0.01** | 0.18 ± 0.01 | **0.19 ± 0.02** |
| | LinSep | 0.28 ± 0.02 | **0.29 ± 0.01** | 0.27 ± 0.02 | **0.32 ± 0.03** | 0.24 ± 0.02 | **0.23 ± 0.01** |
| GradCAM | Cluster | 0.20 ± 0.01 | **0.21 ± 0.03** | 0.25 ± 0.02 | **0.31 ± 0.01** | 0.20 ± 0.03 | **0.21 ± 0.02** |
| | LinSep | 0.27 ± 0.02 | **0.34 ± 0.01** | 0.33 ± 0.01 | **0.35 ± 0.02** | 0.25 ± 0.01 | **0.26 ± 0.03** |
| FullGrad | Cluster | 0.18 ± 0.03 | **0.19 ± 0.02** | 0.23 ± 0.01 | **0.26 ± 0.03** | 0.16 ± 0.02 | **0.22 ± 0.01** |
| | LinSep | 0.26 ± 0.02 | **0.30 ± 0.01** | 0.29 ± 0.02 | **0.32 ± 0.01** | 0.27 ± 0.01 | **0.25 ± 0.03** |
| CALM | Cluster | 0.23 ± 0.01 | **0.24 ± 0.02** | 0.28 ± 0.01 | **0.30 ± 0.02** | 0.22 ± 0.02 | **0.25 ± 0.01** |
| | LinSep | 0.29 ± 0.02 | **0.35 ± 0.01** | 0.33 ± 0.02 | **0.37 ± 0.01** | 0.27 ± 0.01 | **0.30 ± 0.03** |
| MFABA | Cluster | 0.19 ± 0.01 | **0.26 ± 0.03** | 0.27 ± 0.02 | **0.34 ± 0.01** | 0.23 ± 0.02 | **0.26 ± 0.01** |
| | LinSep | 0.28 ± 0.02 | **0.36 ± 0.01** | 0.32 ± 0.01 | **0.36 ± 0.03** | 0.29 ± 0.03 | **0.34 ± 0.02** |

Table 19: Detection F1 (avg. across concepts) from SAE concepts: 92% through CLIP for image datasets and 81% through Gemma for text datasets.

| | Concept Detection Methods | | | | |
|---|---|---|---|---|---|
| | CLS | RandTok | LastTok | MeanTok | SuperTok (Ours) |
| CLEVR | 0.898 ± 0.135 | 0.504 ± 0.077 | 0.504 ± 0.077 | 0.609 ± 0.083 | **0.992 ± 0.090** |
| COCO | 0.462 ± 0.064 | 0.335 ± 0.049 | 0.339 ± 0.049 | **0.591 ± 0.069** | 0.582 ± 0.000 |
| Surfaces | 0.419 ± 0.062 | 0.345 ± 0.042 | 0.344 ± 0.042 | 0.479 ± 0.074 | **0.501 ± 0.085** |
| Pascal | 0.570 ± 0.063 | 0.398 ± 0.049 | 0.404 ± 0.053 | 0.601 ± 0.060 | **0.662 ± 0.000** |
| Sarcasm | **0.662 ± 0.075** | 0.659 ± 0.052 | 0.659 ± 0.052 | 0.659 ± 0.052 | 0.659 ± 0.052 |
| iSarcasm | 0.706 ± 0.069 | 0.676 ± 0.044 | 0.676 ± 0.044 | 0.703 ± 0.051 | **0.777 ± 0.054** |
| GoEmotions | 0.159 ± 0.067 | 0.124 ± 0.062 | 0.124 ± 0.062 | 0.350 ± 0.106 | **0.395 ± 0.093** |

# N  SPARSE AUTOENCODERS

## N.1  SAEs FOR CONCEPT DETECTION

Sparse autoencoders (Goh et al., 2021) (SAEs) have recently been proposed as a mechanism for uncovering latent concepts in large models. By training an encoder–decoder architecture with sparsity constraints, SAEs aim to discover a set of basis features that are both interpretable and disentangled. This approach is attractive for concept analysis because sparsity encourages individual hidden units to capture relatively specific and semantically meaningful directions in representation space. In principle, such units could act as natural "concept detectors" without additional supervision.

Despite these benefits, SAEs come with notable limitations. Training them at scale is extremely resource-intensive, and thus only a small number of pretrained SAEs have been made publicly available. These models are typically trained on very specific layers of particular architectures and cannot be easily transferred to other checkpoints or layers. For this reason, we restrict our comparisons to what is currently feasible: an SAE trained on the penultimate residual stream of CLIP (Radford et al., 2021; Schuhmann et al., 2022; Ilharco et al., 2021) (covering 92% of the model depth for images) and SAEs trained on intermediate layers of Gemma (Team et al., 2024; Lieberum et al., 2024) (covering 81% of the depth for text). A second practical issue is that SAEs output thousands of candidate units, which makes automatic labeling more difficult. To address this, we filtered out units that activated on nearly all samples or no samples (Cywiński & Deja, 2025), or with insufficient activation strength (Gao et al., 2024b).

After filtering, we evaluated the retained SAE units as potential unsupervised concept detectors. We apply the same SuperActivator paradigm for detection, treating CLS and token-alignment with the retined SAE units as concept activation scores.

Table 19 shows the $F_1$ concept detection performance for the best-perfoming SAE units for each ground truth concept. Our SuperActivators method performs quite well across all datasets. However, we note in Figure 31 that our method achieved peak performance by just using a much larger subset of the most activated tokens (larger N%). We suspect this is due to the sparsity constraint in SAE training objectives. By penalizing high activations, SAEs eliminate weak and noisy responses and shrink the scale of the surviving ones. With less contrast between the strongest and moderate responses, concept evidence becomes spread across more activated tokens and less concentrated in the tail.

## N.2  SAEs FOR CONCEPT ATTRIBUTION

Having established that SAEs can act as competitive unsupervised detectors, we next evaluate whether they can also support concept attribution. Tables 20 and 21 report average attribution $F_1$ across both image (using CLIP model) and text (using Gemma model) datasets.

Across all methods, we observe a consistent pattern: SuperActivators pooling yields stronger attribution performance than CLS pooling. On image datasets, SuperActivators improves scores in nearly

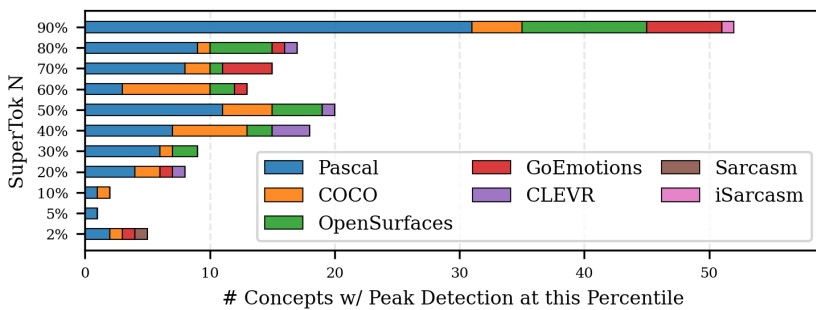

Figure 31: For SAEs The strongest globally applicable concept signals are not concentrated in a very sparse set of signals.

every setting, often by non-trivial margins. Similar trends appear in text, where SuperActivators again provides the strongest performance in most cases.

While the average $F_1$ across all concepts remains modest relative to supervised baselines, the results highlight a consistent trend: even for SAEs, SuperActivators consistently provides a more accurate signal for both concept detection and attribution than global CLS-based pooling. This suggests that fine-grained, token-level alignment is crucial for extracting interpretable signals from unsupervised representations.

Table 20: Average Attribution F1 for SAEs on Image Datasets with CLIP model.

(a) CLEVR and COCO Dataset

| Attribution Method | CLEVR | | COCO | |
|---|---|---|---|---|
| | CLS | SuperActivators | CLS | SuperActivators |
| LIME | $0.45 \pm 0.04$ | $\mathbf{0.49 \pm 0.01}$ | $0.32 \pm 0.03$ | $\mathbf{0.33 \pm 0.04}$ |
| SHAP | $0.47 \pm 0.05$ | $\mathbf{0.51 \pm 0.03}$ | $0.31 \pm 0.03$ | $\mathbf{0.34 \pm 0.02}$ |
| RISE | $0.44 \pm 0.03$ | $\mathbf{0.48 \pm 0.03}$ | $0.30 \pm 0.02$ | $\mathbf{0.33 \pm 0.01}$ |
| SHAP IQ | $\mathbf{0.46 \pm 0.04}$ | $\mathbf{0.46 \pm 0.02}$ | $0.28 \pm 0.05$ | $\mathbf{0.33 \pm 0.04}$ |
| IntGrad | $0.40 \pm 0.05$ | $\mathbf{0.44 \pm 0.04}$ | $0.27 \pm 0.04$ | $\mathbf{0.31 \pm 0.03}$ |
| GradCAM | $0.36 \pm 0.05$ | $\mathbf{0.40 \pm 0.05}$ | $0.26 \pm 0.05$ | $\mathbf{0.30 \pm 0.04}$ |
| FullGrad | $0.37 \pm 0.04$ | $\mathbf{0.41 \pm 0.02}$ | $\mathbf{0.32 \pm 0.03}$ | $0.31 \pm 0.04$ |
| CALM | $0.44 \pm 0.02$ | $\mathbf{0.49 \pm 0.04}$ | $0.27 \pm 0.05$ | $\mathbf{0.32 \pm 0.03}$ |
| MFABA | $0.44 \pm 0.03$ | $\mathbf{0.49 \pm 0.02}$ | $0.28 \pm 0.04$ | $\mathbf{0.30 \pm 0.03}$ |

(b) OpenSurfaces and Pascal Dataset

| Attribution Method | OpenSurfaces | | Pascal | |
|---|---|---|---|---|
| | CLS | SuperActivators | CLS | SuperActivators |
| LIME | $0.41 \pm 0.04$ | $\mathbf{0.43 \pm 0.04}$ | $0.40 \pm 0.05$ | $\mathbf{0.44 \pm 0.04}$ |
| SHAP | $0.31 \pm 0.03$ | $\mathbf{0.35 \pm 0.02}$ | $0.41 \pm 0.04$ | $\mathbf{0.45 \pm 0.03}$ |
| RISE | $0.36 \pm 0.05$ | $\mathbf{0.40 \pm 0.02}$ | $0.40 \pm 0.05$ | $\mathbf{0.44 \pm 0.05}$ |
| SHAP IQ | $0.37 \pm 0.04$ | $\mathbf{0.41 \pm 0.05}$ | $0.41 \pm 0.05$ | $\mathbf{0.45 \pm 0.01}$ |
| IntGrad | $0.39 \pm 0.02$ | $\mathbf{0.43 \pm 0.02}$ | $0.46 \pm 0.05$ | $\mathbf{0.50 \pm 0.02}$ |
| GradCAM | $0.32 \pm 0.05$ | $\mathbf{0.36 \pm 0.02}$ | $0.34 \pm 0.03$ | $\mathbf{0.38 \pm 0.04}$ |
| FullGrad | $0.34 \pm 0.03$ | $\mathbf{0.38 \pm 0.03}$ | $0.36 \pm 0.05$ | $\mathbf{0.40 \pm 0.02}$ |
| CALM | $0.26 \pm 0.05$ | $\mathbf{0.30 \pm 0.02}$ | $0.35 \pm 0.04$ | $\mathbf{0.39 \pm 0.03}$ |
| MFABA | $\mathbf{0.39 \pm 0.04}$ | $\mathbf{0.39 \pm 0.02}$ | $0.41 \pm 0.03$ | $\mathbf{0.46 \pm 0.02}$ |

Table 21: Average Attribution F1 for SAEs on Text Datasets with Gemma Model.

| Attribution Method | Sarcasm | | iSarcasm | | GoEmotions | |
|---|---|---|---|---|---|---|
| | CLS | Super Activators | CLS | Super Activators | CLS | Super Activators |
| LIME | **0.37 ± 0.05** | 0.36 ± 0.02 | 0.62 ± 0.03 | **0.65 ± 0.04** | 0.16 ± 0.04 | **0.20 ± 0.04** |
| SHAP | 0.33 ± 0.04 | **0.37 ± 0.04** | 0.59 ± 0.05 | **0.64 ± 0.01** | 0.18 ± 0.03 | **0.23 ± 0.02** |
| RISE | 0.37 ± 0.05 | **0.42 ± 0.03** | 0.68 ± 0.04 | **0.72 ± 0.04** | 0.20 ± 0.05 | **0.22 ± 0.02** |
| SHAP IQ | **0.40 ± 0.05** | **0.40 ± 0.02** | 0.68 ± 0.05 | **0.69 ± 0.02** | 0.18 ± 0.04 | **0.23 ± 0.02** |
| IntGrad | 0.31 ± 0.05 | **0.35 ± 0.04** | 0.52 ± 0.05 | **0.57 ± 0.04** | 0.10 ± 0.04 | **0.15 ± 0.05** |
| GradCAM | 0.34 ± 0.04 | **0.39 ± 0.03** | 0.53 ± 0.03 | **0.58 ± 0.01** | 0.16 ± 0.05 | **0.20 ± 0.02** |
| FullGrad | 0.28 ± 0.05 | **0.33 ± 0.03** | **0.59 ± 0.04** | **0.59 ± 0.03** | 0.14 ± 0.03 | **0.18 ± 0.04** |
| CALM | 0.37 ± 0.04 | **0.39 ± 0.04** | 0.56 ± 0.05 | **0.60 ± 0.04** | 0.16 ± 0.03 | **0.21 ± 0.02** |
| MFABA | 0.33 ± 0.03 | **0.38 ± 0.03** | 0.55 ± 0.04 | **0.60 ± 0.02** | 0.18 ± 0.03 | **0.23 ± 0.02** |

