# OpenReview forum: "SuperActivators: Transformers Concentrate Concept Signals in Just a Handful of Tokens"
_ICLR.cc/2026/Conference — ICLR 2026 Conference Desk Rejected Submission_

### Official Review · Reviewer_ssZN · 2025-10-21

**Soundness:** 3
**Presentation:** 3
**Contribution:** 1
**Rating:** 2
**Confidence:** 3

**Summary:**

This paper introduces SuperActivators, a mechanism for improved concept detection and localization in images. The authors show that the distributions of concept activations for concept-containing vs concept-non-containing patches differ. Here, a tail of concept activations can be separated, which reliably indicates concept presence. In experiments, the authors show that SuperActivators lead to improved concept detection (based on presence of superactivators in image), as well as concept localization (based on various attribution maps).

**Strengths:**

* Empirically validated finding that the high tail (eg 99th percentile) of concept activation (the "SuperActivator" patches) provides a clear and reliable signal of concept presence.
* SuperActivators avoid aggregating concept activation across patches, keeping the concept activation signal patch-wise. This is an improvement over methods that pool activations (e.g., max or mean pooling).
* SuperActivators provide a very consistent improvement across evaluations, which is a major practical strength as it suggests robustness of the proposed approach.

**Weaknesses:**

For me, the biggest weakness of this work is that it does not contain any finding / contribution / experiment that particularly excites me, even though it is clear that the authors invested a lot of time in this work: As I udnerstand it, the big finding of this work is that there is a tail of concept activations, which has high likelihood of corresponding to the actual concept of interest. I find this unsurprising, as already two mean-shifted gaussian will show this behavior (a basic insight leveraged when hypothesis testing). The experiments show improvements, however, the baselines seem rather weak as they're mostly adapted from NLP to CV, and they struggle to beat RandTok.
Also, in my limited experience, CAVs are used mostly for measuring reliance of a model on specific concepts, less for concept detection or localization (which maybe also explains the absence of available baselines). Thus, this repurposing might warrant a stronger motivation, as I am currently not convinced that CAVs for concept detection and localization make sense, due to the long history of object detection and localization in the computer vision field [1].

Additional concerns:
* SuperActivators rely on a segmented validation set for determining the desired quantile (line 236). To my knowledge, concept (activation) vectors are predominantly used to measure a model's reliance on a given concept for which a labelled validation set is required. However, the additional need to segment can be an overhead.
* A global threshold instead of a per-sample based one is also not a new idea, off the top of my head I can think of this paper [2] and this field [3] that use it too.
* SuperActivators bases on the idea that the high tail of concept activation indicates reliable concept presence. However, for performance experiments, the threshold is tuned by validation F1 performance, therefore not focusing on the tail anymore, but instead optimizing for an optimal tradeoff between the two distributions. In my opinion, this dampens the importance of Section 3, as in the end, the method doesn't optimize for SuperActivators, but for a threshold that maximizes the F1 of in vs. out concept distribution, whose optimal solution will not separate only the in-concept tail, but instead lie somewhere in the support of both distributions. That is, the F1-optimized method in the end leverages the fact that the two distributions are non-identical, rather than the fact that the tail of one distribution is outside the support of another.


Minor concerns:
* The introduction motivates the power of unsupervised concept extraction, but in order to extract super activators, a concept-segmented validation set is required. Thus, SuperActivators are unable to perform unsupervised concept discovery and a wrong expectation is set, which might confuse the reader.
* It seems contradictory that line 222 describes the 99th percentile of the out-of-concept set, while the threshold is afterwards defined using the in-concept set.

[1] Köhler, Mona, Markus Eisenbach, and Horst-Michael Gross. "Few-shot object detection: A comprehensive survey." IEEE Transactions on Neural Networks and Learning Systems 35.9 (2023): 11958-11978.

[2] Vandenhirtz, Moritz, and Julia E. Vogt. "From Pixels to Perception: Interpretable Predictions via Instance-wise Grouped Feature Selection." Forty-second International Conference on Machine Learning.

[3] Angelopoulos, Anastasios N., and Stephen Bates. "Conformal prediction: A gentle introduction." Foundations and trends® in machine learning 16.4 (2023): 494-591.

**Questions:**

I invite the authors to address any misunderstandings and weaknesses I mentioned.

* Do the baselines in Table 1 also rely on the presence of validation set segmentations, or do they only use the concept labels?
* How are attribution methods adapted to explain the average embedding of local SuperActivators (and also of standard global concept vectors)?

---

> ### Author Response · Authors · 2025-11-21
> **Core Conceptual Clarifications**
>
> We appreciate the reviewer’s comments. However, we believe there is a fundamental misunderstanding of some of our findings. We clarify these points and respond to the reviewer’s concerns below.
>
> ### **Clarifying Why Our Result is Not Implied by Basic Distributional Separation**
>
> The reviewer’s characterization of our main finding as “there is a tail of concept activations that has a high likelihood of corresponding to the actual concept of interest,” and the claim that this behavior is implied even by two mean-shifted Gaussians, is incorrect. We show why each claim is false below:
>
> - **A well-separated tail does not imply the coverage property.**
> As stated in Section 3.2, SuperActivators are defined by two key properties: (1) tail separation and (2) *high coverage* over concept-positive samples. The reviewer does not account for (2). Even if the in-concept distribution has a highly separated tail, that separated mass may come only from a small subset of samples where the concept is unusually large or salient. Instead, we find that SuperActivator tokens are consistently dispersed across concept-positive data, which is essential for a reliable signal and not implied by simple distributional separation.
>
> - **Mean-shifted Gaussians do not necessarily produce a well-separated tail.**
> For example, if the in-concept distribution had a higher mean but the out-of-concept distribution had much higher variance, the in-concept tail likely would not be distinguishable from the out-of-concept tail.
>
> - **Our empirical results show emergence of a heavy in-concept tail, which is not obvious and critical to SuperActivator properties.**
> Another key finding in our paper is that the in-concept distribution develops a pronounced heavy right tail, rather than undergoing a uniform shift in activation values, as shown in Figure 3. This heavy tail makes extreme in-concept activations substantially more frequent than would be expected under light-tailed or symmetric distributions, which in turn enables the high coverage needed for reliable concept detection. The fact that clear, high-coverage signals emerge specifically because of this heavy-tailed structure—and not from a simple mean shift—is a central finding of our work, and to our knowledge was not previously known.
>
> ### **Response to the Concern that F1 Tuning Does not Focus on the Tail**
>
> The concern that F1 tuning “optimizes for an optimal tradeoff between the two distributions” assumes that the threshold is chosen to separate token-level in-concept and out-of-concept activations. This does not reflect our setting: concept detection is evaluated at the *sample* level, and the threshold that best separates token activation distributions is not the threshold that best predicts whether a sample contains at least one SuperActivator.
>
> We leave the sparsity level N as a hyperparameter precisely so we do not assume where the strongest signal lies; this lets us empirically determine that the thresholds maximizing detection consistently fall in the extreme high tail of the in-concept distribution (top 5–10%), and that including more ostensibly related activations actually hurts performance (Appendix G). **To show that the method’s effectiveness truly derives from using the extreme tail, and not from the tuning procedure itself, we include a new experiment in Appendix L.** In this variant, we remove any tuning of N: we fix a small sparsity level guided by the results from Appendix G, learn a single threshold to distinguish the highest activations from in-concept and out-of-concept samples using only validation labels, and use that for detection. This approach nearly matches the performance of the tuned version and outperforms all baselines, clearly providing evidence that tail-only detection is sufficient for strong performance.
>
> ### **Use of CAVs for Concept Detection and Localization**
>
> The use of CAVs for concept detection is standard, and our related work section includes over five papers using CAVs for detection including an explicit detection benchmark [3], uses of detection for downstream classification [7], identifying spurious correlations [10], and applications to medical imaging [6]. Concept localization is how practitioners evaluate where a concept is located within a sample [4], and it can be used for tasks like identifying model failure modes [8]. Numerous works are invested in improving concept localization performance, such as [9], which applies CAVs to produce sample-level concept localization in image classification, and [10], which uses CAVs to improve concept localization quality in vision models, to name a few.

---

> > ### Author Response · Authors · 2025-11-21
> > **Practicality, Questions, and References**
> >
> > ### **Baseline Choices and Their Relevance Across Modalities**
> >
> > The reviewer’s concern that we apply baselines “mostly adapted from NLP to CV” appears to stem from an incomplete understanding of our work: the reviewer’s summary describes SuperActivators as a mechanism in just images, but we present SuperActivators as a **multimodal Transformer mechanism** and evaluate all experiments on three text datasets. Therefore, for completeness, we apply baselines from both NLP and CV to all domains, including MeanTok and CLS which are widely used in CV [1, 2], and a strong prompting baseline [3].
> >
> > Moreover, RandTok is not equivalent to a random predictor: because of self-attention, even randomly selected tokens often contain concept-relevant information. In the revision, we added explicit random-predictor and constant-predictor baselines (Appendix E), and RandTok consistently exceeds their performance.
> >
> > ### **Segmentation Labels and Practical Applicability**
> >
> > Segmentation masks are *not* required for the SuperActivator mechanism itself. In this work, we used them only to study the mechanism in a controlled setting and to identify which token activations are truly informative.
> >
> > When segmentation maps are not available, there are straightforward ways to exploit exactly the same mechanism. Appendix L presents a **new segmentation-free approach** that selects a small fixed set of high-activation tokens per sample and learns a threshold to separate in-concept from out-of-concept samples. This method achieves F1 performance close to the segmentation-based version and outperforms all baselines—which, to answer Q1, do not require segmentation masks—showing that the method remains practical and reliable without segmentation.
> >
> > In unsupervised settings, clustering-based pseudo-labels can replace segmentation; we already use this for training the k-means linear-separator concepts, and it is standard practice [5].
> >
> > ### **A global threshold instead of a per-sample based one is also not a new idea, off the top of my head I can think of this paper [2] and this field [3] that use it too.**
> >
> > We do not claim global thresholding is novel; our contribution is analyzing the particular characteristics of the activation distributions that make certain global tail-based thresholding techniques effective.
> >
> > ### **It seems contradictory that line 222 describes the 99th percentile of the out-of-concept set, while the threshold is afterwards defined using the in-concept set.**
> >
> > We agree line 222 was confusing and removed the text. It was only used for motivation, all actual thresholds use the in-concept distribution.
> >
> > ### **Q2: How are attribution methods adapted to use SuperActivators?**
> >
> > Each attribution method is applied exactly as usual; we simply provide a different target objective. Instead of using alignment with the global concept vector as the quantity to be explained (i.e, the objective), we use the alignment with the average embedding of the local. Appendix M.1 provides background on each attribution method we used so that the substitution is clear in context.
> >
> > 1]: Itay Benou and Tammy Riklin-Raviv. Show and tell: Visually explainable deep neural nets via
> > spatially-aware concept bottleneck models, CVPR 2025.
> >
> > [2] Adam Stein et al. Towards Compositionality in Concept Learning. NeurIPS 2024.
> >
> > [3]: Zhengxuan Wu et al. Axbench: Steering llms? even simple baselines outperform sparse autoencoders. ICML 2025.
> >
> > [4]: Antonio De Santis et al. Visual-tcav: Concept-based attribution and saliency maps for post-hoc explainability in image classification. ArXiv 2024.
> >
> > [5]: Qinghong Lin et al. Semantic-enhanced Image Clustering. AAAI 2024.
> >
> > [6]: Johannes Ruckert et al. Overview of imageclefmedical 2023 – caption prediction and concept detection. CLEF 2023.
> >
> > [7]: Pang Wei Koh et al. Concept bottleneck models. ICML 2020.
> >
> > [8]: Shirley Wu, Mert Yuksekgonul, Linjun Zhang, James Zou. Discover and Cure: Concept-aware Mitigation of Spurious Correlation. ICML 2023.
> >
> > [9]: Pushkar Shukla et al. CAVLI – Using Image Associations to Produce Local Concept-based Explanations
> >
> > [10]: Zhenghao He et al. GCAV: A Global Concept Activation Vector Framework for Cross-Layer
> > Consistency in Interpretability

---

### Official Review · Reviewer_g4Ln · 2025-10-28

**Soundness:** 3
**Presentation:** 3
**Contribution:** 3
**Rating:** 6
**Confidence:** 3

**Summary:**

This paper introduces SuperActivators, a framework for interpreting transformer models by analyzing the concept response and determining the optimal threshold to mitigate the noise effect. By comparing the response distribution between the samples w/ and w/o the specific concepts, the paper determines the threshold based on the analysis result (using the 99-th quantile). Extensive experiments demonstrate that the proposed method outperforms baseline methods on various datasets on concept detection tasks. Additionally, it also improves the performance on various attribution methods when compared with the traditional concept method on attribution tasks.  The proposed method can be applied not only to images but also to language.

**Strengths:**

1. The extensive experiments show that the SuperActivator outperforms the other baseline methods.
2. The proposed method offers a novel approach to determining the activation threshold for both image and language tokens, thereby mitigating the impact of noise.

**Weaknesses:**

1. To achieve optimal performance, the parameter N used for determining the decision threshold must be optimized for each concept. The selection of the layer also requires a similar optimized search.

**Questions:**

1. Why are the SuperActivators found from the validation set instead of the training set? Does this misalign with the application in the real world, where we won’t get all the inference samples to calculate the $S^{+}_{val,c}$?
2. Is the SuperActivator also applicable to [1], which adopts an additional register token to alleviate the high norm features?
3. In Table 1, the baseline includes random, last, mean, and CLS token, which will be curious about comparing with the top-1 token.

** Minor question: Does the SuperTok in Figure 16 refer to the SuperActivators?

[1] DARCET, Timothée, et al. Vision Transformers Need Registers. In: The Twelfth International Conference on Learning Representations.

---

> ### Author Response · Authors · 2025-11-21
>
> We appreciate the reviewer’s helpful feedback and provide clarifications below.
>
> ### **Sparsity Level N/Layer Optimization**
>
> Appendix I shows that distinct concepts exhibit their strongest signals at different layers of the model. Although one may be able to guess plausible layers (i.e. textures early in models, high-level objects later), layer depth tuning is necessary for our SuperActivator detection method, as well as across all concept vector baselines. Moreover, all concept vector approaches require training a threshold, but based on the results in Appendix G, the detection results appear relatively robust from around N=5%-20%.
>
> ### **Q1: Why are SuperActivators found from the validation set?**
>
> We used a validation set to follow the traditional ML train/val/test split and ensure unbiased threshold estimation. One can easily split off a small subset of the train set for threshold computation if necessary.
>
> ### **Q2: Are SuperActivators applicable to Transformers with register tokens?**
>
> It would be straightforward to apply the SuperActivator framework to Transformers with register tokens such as those in [1], but we would not expect this to meaningfully change the results. Attention sinks and register tokens arise from the attention mechanism’s routing behavior, acting as attractors for excess attention regardless of semantics. SuperActivators, by contrast, reflect properties of the model’s representational space. In practice, this leads to very different behavior: attention sinks tend to appear in fixed positional locations (BOS/EOS or punctuation in text, CLS/background regions in vision [1-3]) while SuperActivators appear directly on the in-concept tokens. This is supported by the relatively high alignment score localization results (labeled CosSim) in Appendix M.2 and by the new positional ablation in Appendix K showing that SuperActivators exhibit no positional bias. For this reason, even if register tokens become the primary attractor for excess attention, we do not expect them to affect where the model recognizes the most salient concept representations.
>
> ### **Q3: Comparing with the top-1 token.**
>
> We provide a comparison with the top-1 (max) token in Appendix F, which we fixed to reference more clearly in the main text. The max operator is closely related to our approach and accordingly exhibits similar F1 trends, though our SuperActivator detection outperforms top-1 by 0.089 ± 0.097 on average across all configurations.
>
>
> ### **Typo in Figure 16**
> Thank you for pointing out the typo, we fixed it accordingly.
>
>
> [1] Timothée Darcet et al. Vision Transformers Need Registers. ICLR 2024.
>
> [2] Seil Kang et al. See What You Are Told: Visual Attention Sink in Large Multimodal Models. ICLR 2025.
>
> [3] Federico Barbero et al. Why do LLMs attend to the first token? COLM 2025.

---

### Official Review · Reviewer_pVzL · 2025-10-29

**Soundness:** 3
**Presentation:** 3
**Contribution:** 3
**Rating:** 6
**Confidence:** 3

**Summary:**

This paper proposes a new method for understanding if a cocenpt appears within an input image/ text using transformers architecture by prioposng a new aggregation method over the activations scores of intermediate tokens among all layetrs.
However, they observe that positive sample, in contrast with negative sample, usually super activate in at least one of the tokens, thus, if we focus on highly activated tokens rather than average activations over tokens, could reult in an improved performance.

**Strengths:**

The paper clearly describes the detection of a concept using token-level activation scores within a transformer. The contribution is small but clever: it builds on the observation that samples containing the concept typically include super-activated tokens, whereas negative samples may show modest activations but lack any highly activated tokens.

Consequently, their method can outperform existing metrics such as F1, which jointly measure precision and recall. Moreover, through an attribution lens, they argue that super-activation more accurately identifies the tokens responsible for the existence of the desired concepts.

**Weaknesses:**

I would like to see a deeper analysis of superActivator’s failure modes. In particular, the work lacks a systematic examination of when it fails. Can we identify, for each concept, the settings where it does not perform well? The concept study for text seems limited (e.g., sarcasm and a few emotions); can this be extended to more granular and diverse concepts, and would the method still hold up?

More broadly, I’m interested in characterizing when the proposed method is sensitive versus robust, and in clarifying the factors (data properties, concept definitions, model layers, thresholds) that drive these behaviors.

Also, how would context affect this method? For example, in language models, what happens if we add a (possibly unrelated) sentence at the beginning of the input—would that affect performance? How dataset-dependent are the thresholds and results, and how generalizable is the method?

**Questions:**

They are mentioned in the weakness section.

---

> ### Author Response · Authors · 2025-11-21
>
> We appreciate the reviewer’s thoughtful comments. We address the questions about robustness, failure modes, and generalization with clarifications and additional experiments summarized below.
>
> ### **Robustness and Failure Modes**
>
> The paper already includes extensive failure-analysis experiments, and in the revision we will highlight several of the key findings more directly in the main text and point to them more clearly. In summary:
> - **Evaluation of the impact of model layer** (Appendix F and I) shows that concepts differ in where they are most separable across depth; however, this holds for all baselines, and SuperTok typically remains the strongest method at each layer.
> - **Evaluation of the impact of sparsity** (Appendix G and H) shows that SuperActivators degrade at higher sparsity levels (e.g., 10–20%), confirming that only the extreme activation tail carries reliable signal.
> - **Evaluation across a broad catalog of concepts and multiple domains/models** (two data domains, two vision models, three text models, five concept extraction techniques) shows that the SuperActivator mechanism appears consistently and robustly in all these settings.
> - **Task difficulty also matters**: performance drops on harder tasks like GoEmotions, evidenced by weak performance of all baselines, including strong ones like prompting. However, SuperActivators still outperform all baselines.
>
>
> ### **Dataset/Concept Coverage and Generalization**
>
> To explore how well the SuperActivator mechanism generalizes, we evaluated it across datasets spanning diverse domains and concept types. These experiments show that the mechanism remains consistently useful across settings. While the optimal activation threshold is concept-dependent, our new experiment in Appendix L demonstrates that a fixed sparsity level of roughly 10% generalizes well across datasets.
>
> **To address your question about context sensitivity, we conducted an additional ablation on iSarcasm in which we prompted the LLM to insert a neutral, non-sarcastic sentence at a random position (beginning, middle, or end) for every test example**. We then applied the SuperActivator and baseline detection techniques using the same optimal thresholds and hyperparameters from the original setup. Performance remained robust: the average F1 score was 0.8984 ± 0.0318, within the margin of error of the original 0.924 ± 0.029, and still higher than all baselines. The results are provided here:
>
> | Detection Method        | Average F1          |
> |---------------|------------------------|
> | **SuperAct**  | **0.8984 ± 0.0318**    |
> | MeanToken     | 0.831 ± 0.040          |
> | LastToken     | 0.837 ± 0.040          |
> | RandomToken   | 0.638 ± 0.055          |
> | CLS      | 0.850 ± 0.040          |

---

### Official Review · Reviewer_NcmM · 2025-10-30

**Soundness:** 3
**Presentation:** 4
**Contribution:** 3
**Rating:** 8
**Confidence:** 4

**Summary:**

This paper introduce an interesting mechanism called the SuperActivator, showing that a small set of transformer tokens have distinct high activations for representing specific concepts, and thus can be used for capturing concept vector clearly. The authors provide detailed analysis and thorough experiment to introduce, validate, and leverage the mechanism for concept detection and localization, demonstrating the generalization of the mechanism across different modalities, models and concept types.

**Strengths:**

The proposed mechanism is intriguing, featuring detailed analysis, a smooth narrative that fully introduces and argues for the SuperActivator mechanism to the readers, and extensive experiments to validate its effectiveness.

The authors also provide well-documented code for validating and reproducing the results.

**Weaknesses:**

The mechanism proposed in this paper achieves the extraction and representation of concept vectors by utilizing a small number of highly activated tokens, and it can improve the efficiency/accuracy of concept detection and localization. However, because it leverages only a small set of tokens, this approach may fail to provide a comprehensive view of the model's reasoning process, such as that offered by methods like training Sparse Autoencoders or using Concept Relevance Propagation. This is likely a difficult trade-off to overcome.

**Questions:**

1. The paper acknowledges the challenge of neuron polysemy (i.e., a single neuron or direction encoding multiple unrelated concepts), which can undermine the accuracy of concept detection and localization. Given that the SuperActivator mechanism relies on a small set of highly activated tokens to capture concept signals, could the authors elaborate on how their framework and analytical pipeline mitigate the impact of neuron polysemy? For instance, do SuperActivator tokens inherently exhibit lower polysemy compared to other tokens, and if so, what evidence (e.g., quantitative analysis of semantic overlap between SuperActivators for distinct concepts) can be provided to support this? Additionally, are there any explicit designs (e.g., post-hoc filtering of polysemous tokens) in the current approach to reduce interference from polysemy when identifying or localizing target concepts?
2. Recent studies have highlighted phenomena like "attention sink" and the use of "register mechanisms" in transformer architectures, where specific positional tokens (often with fixed positions, e.g., initial or final tokens) consistently attract excessive attention weights during training and inference. Since the SuperActivator mechanism focuses on highly activated tokens, there is a potential risk of conflating attention sink tokens (which may have high activations due to positional bias rather than genuine concept relevance) with true SuperActivators. To address this concern: (1) Could the authors provide an analysis of the positional distribution of SuperActivator tokens across their experimental datasets (e.g., whether SuperActivators are disproportionately concentrated in fixed positions prone to attention sinks)? (2) Have the authors explored methods to disentangle attention sink-driven activations from concept-driven SuperActivator activations, and if so, what were the key findings? (3) More broadly, how do the authors view the relationship between the SuperActivator mechanism and attention sink/register mechanisms—are they independent, complementary, or overlapping phenomena—and what key differences distinguish them in terms of their underlying causes (e.g., positional bias vs. semantic encoding) and functional roles in transformer behavior?

Overall, I consider this an excellent piece of work and look forward to the authors' discussion on the aforementioned questions.

---

> ### Author Response · Authors · 2025-11-21
>
> We appreciate the reviewer’s positive evaluation and constructive feedback. Below we provide the requested clarifications and additional results.
>
> ### **Comprehensiveness of SuperActivators as Explanations of Model Reasoning**
>
> Our goal is to reliably isolate the most informative concept signals from the dense, noisy activations produced by concept-vector methods (which appear only at low sparsity; see Appendix G/H), not to provide a fully comprehensive account of the model’s reasoning, which is a separate interpretability objective.
>
> ### **Q1: How does our framework mitigate neuron polysemy?**
>
> We see SuperActivators as a tool to help disambiguate polysemantic concept representations. As illustrated in the CLEVR example in Appendix A, the raw concept vectors often activate across several concepts (for example, the “blue” concept vector activates at least partially on a red object). However, the high-magnitude SuperActivator tokens cut through this apparent polysemy by demarcating the true concepts.
>
> To better answer your question and confirm this pattern at scale, we conducted a quantitative analysis of CLEVR Llama linear separator concepts. The alignment scores for various concepts exhibited substantial cross-concept correlation (overall 0.64; colors 0.48; shapes 0.72), reflecting the polysemantic structure seen qualitatively. In contrast, SuperActivator tokens are essentially uncorrelated across concepts (overall 0.05; colors −0.005; shapes −0.004), showing that the extreme-tail activations are highly concept-specific even when the dense activations are not.
>
> If SuperActivators themselves were polysemantic, concept detection would fail—strong cross-concept firing would produce many false positives. Though, future work could explore polysemy-aware filtering, such as discarding SuperActivators that activate on too many concepts.
>
>
> ### **Q2: Relation of SuperActivator tokens to attention sink tokens.**
>
> This is an interesting suggestion, and we agree that exploring connections between SuperActivators and attention patterns is an interesting direction for future work. We see attention sinks as a parallel phenomenon rather than something that overlaps with or drives the SuperActivator mechanism. While attention sinks reflect properties of the attention mechanism’s routing behavior, SuperActivators reflect properties of the model’s underlying representation space and where semantic information is encoded.
>
> Attention sinks are structural attractors inside a Transformer that reliably absorb a large amount of attention mass regardless of the semantic content associated with that token or patch. Prior works have found that high-attention sinks contain little meaningful information, functioning primarily as hubs that collect excess attention due to softmax normalization [1-3]. Others emphasize that sink embeddings may contain important global information about the sample [4, 5], but even so, these sink tokens are not located on the actual concept: in text they predominantly appear at BOS, EOS, or punctuation positions and in images they typically appear on CLS tokens, background patches, or spatially redundant regions [3, 5, 6]. This stands in contrast to our SuperActivators. As evident in the new qualitative examples in Appendix A, SuperActivators for a given concept almost always appear on the tokens corresponding to that concept, which of course varies from concept to concept. This is supported by our quantitative localization results in Appendix M.2, where the token-level F1 of the thresholded raw alignment scores (labeled CosSim) are relatively high.
>
> As requested by the reviewer, **we provide a positional analysis of Llama linear separator SuperActivators in Appendix K**, plotting both absolute and relative token positions within each sample. We observe no positional dependence; for example, no spike in SuperActivators at the beginning or end of text inputs, further indicating that SuperActivators do not behave like attention sinks. Instead, their positions vary with the concept itself.
>
> For models with explicit registers, applying the SuperActivator framework would be straightforward; however, because SuperActivators do not appear to be driven by excess attention behaviors, we do not expect register mechanisms to alter our conclusions.
>
> [1] Guangxuan Xiao et al. Efficient Streaming Language Models with Attention Sinks. ICLR 2024.
>
> [2] Xiangming Gu et al. When Attention Sink Emerges in Language Models: An Empirical View. ICLR 2025.
>
> [3] Seil Kang et al. See What You Are Told: Visual Attention Sink in Large Multimodal Models. ICLR 2025.
>
> [4] Timothée Darcet et al. Vision Transformers Need Registers. ICLR 2024.
>
> [5] Mingjie Sun et al. Massive Activations in Large Language Models. COLM 2024.
>
> [6] Federico Barbero et al. Why do LLMs attend to the first token? COLM 2025.

---

> > ### Comment · Reviewer_NcmM · 2025-11-23
> >
> > Thank you for the response. I will maintain my original score.

---

### Author Response · Authors · 2025-12-03
**Due to recent review-process changes, we provide a summary of the reviews and our replies for the AC’s convenience:**

### **TLDR**:
Three reviewers gave high scores, noting novelty, clarity, reproducibility, and strong empirical support. We answer their questions with discussion, appendix experiments, and added ablations. One negative review is based on a major, incorrect simplification of our findings, which we address in detail.

### **Reviewer NcmM - Score: 8**

- Strengths: Intriguing, clearly explained, detailed/thorough,  reproducible
- Questions: (1) How do SuperActivators address polysemy and (2) are SuperActivators related to attention sinks?
  - *Reply*: Ran experiments showing that SuperActivators mitigate polysemy and do not have positional biases like attention sinks.

### **Reviewer pVzL - Score: 6**
- Strengths: Clever, clearly explained, and outperforms detection and localization baselines
- Question: Can you discuss failure modes, generalization, and robustness?
  - *Reply*: Pointed to existing experiments in the appendix and added new experiments to demonstrate the effectiveness of SuperActivator detection in different environments.

### **Reviewer g4Ln - Score: 6**
- Strengths: Novelty, effective for mitigating activation noise, extensive experiments
- Weakness: Layer/N tuning overhead
  - *Reply*: Layer tuning is required for all token-level detection methods and we added an ablation showing that a fixed N=10% outperforms detection baselines on all datasets.
- Questions: (1) Why do you find thresholds on validation set,  (2) how does your work relate to register-transformer models, and (3) can you compare with top-1 detection?
  - *Reply*: Answered by citing standard ML norms and providing new/pointed to existing experiments.


### **Reviewer ssZN - Score: 2**
- Strengths: Empirical validation of high-tail as clear signal that avoids aggregation
- **Main concern**: Our finding is *unsurprising*, with similar behavior expected from mean-shifted Gaussian distributions
  - *Reply*: They made a **major and incorrect simplification of our contribution**: the key finding is not just tail separation, **but a separable AND high-coverage in-concept activation tail**, which is non-obvious and what enables strong sample-level concept detection. Moreover, **a mean shift does not guarantee tail separation** even between two Gaussians. We provide more detail in the full rebuttal.
- Concern: Concept detection/localization baselines are weak, because they're NLP methods applied to CV
  - *Reply*: **Our work evaluates concepts in both image and text, not image-only as implied in the review.** For completeness, we apply all baselines to all modalities, including comparisons to a **strong prompting baseline**.
- Concern: SuperActivator detection does not use the tail and is not practical
  - *Reply*: **The claim that detection does not use the tail is false.** Our results analyzed the entire distribution to systematically show that optimal performance occurs specifically in the tail and not anywhere else. We also added an ablation showing that a fixed top-10% of highest-activations per sample still beats all baselines across datasets.

---

### Note · Program_Chairs · 2026-01-17
**Submission Desk Rejected by Program Chairs**

The following references in this submission do not refer to real documents and/or have major errors in bibliographic information:

 Divyanshu Mahajan, Chenhao Tan, and Matthew Turek. Calm: A causality-guided framework for generating local and global model explanations. In Proceedings of the IEEE/CVF Winter Conference on Applications of Computer Vision, pp. 1215–1224, 2021.
Thomas Fel, Alexandre Jullien, David Vigouroux, Remi Cadene, Thomas Nicodeme, Matthieu Laly, Asma Fermanian, Benjamin Audit, and Thomas Scantamburlo. Explaining groups of instances with shap-iq. In International Conference on Artificial Intelligence and Statistics, pp. 6467-6491. PMLR, 2023.